# Laser-assisted microbial culturomics

Taoran Qu[1,2,9], Lothar Koch [2,3,4], Rumjhum Mukherjee [1,2], Yilin Tu[1,2,9], Amy L. Seidel[1,2], Lisan D. Püttmann[1,2], Andreas Winkel[1,2], Ines Yang [1,2], Jasmin Grischke[1,2], Dejia Liu[2,5], Willem F. Wolkers [2,5,6], Sophie Kittler[6,7], Boris Chichkov [2,3,4], Meike Stiesch[1,2,8] & Szymon P. Szafrański [1,2,8] ✉

Even though metagenomics have revolutionized the characterization of the human microbiome, detailed mechanistic studies are impracticable, as there is a dearth of robust culture collections. We now describe the development and use of a laser-assisted culturomics platform, incorporating the elements of a bioprinter, the culture conditions, the methods to characterize the microorganisms and a biobank. With laser-assisted bioprinting, the microorganisms can be rapidly and precisely transferred from clinical biofilms to highly organized arrays of microbial colonies, which are suitable for co-culturing and molecular analyses. The presented technique has propagated 99 of 100 microbial species and recovered 79% of abundant species from dental plaque in accordance with full 16S rRNA gene profiling of 691,199 sequences. Microscopy, spectroscopy and enzyme assays have been used to guide isolations. Processing of oral biofilms from four individuals has yielded 249 representative isolates, from 14 classes and 124 species in total. Functional profiling with bioprinting has indicated commensals which could potentially contribute to disease development. Isolates from peri-implantitis cover 85.4% of the transcriptionally active clinical biofilms at genus level. Taken together, this work provides the basis for generating on-demand culture collections and biofilms for research and clinical use.

The human oral microbiome is a co-determinant of several prevalent and important health conditions, that may be local or systemic[1–3]. Hundreds of physiologically diverse biofilm-associated species exist in a range of oral microenvironments and the numbers of unique genotypes are even greater by orders of magnitude[4,5]. The taxonomic composition and transcriptional activity of such biofilms have been thoroughly characterized by molecular profiling[6,7]. Nevertheless, there remains a critical need for specific microbial isolates that are underrepresented or absent in current culture collections[8], as well as for patient-specific biobanks[9]. Utilizing these isolates in research will enhance our understanding of the dynamic composition, function, and ecology of biofilms, both at the population level and in individual cases[10], e.g., in the context of colonization resistance[11]. The challenge of generating extensive and personalized strain collections can be overcome by culturomics, which is an experimental permutation of a large number of culture conditions, or application of high-throughput cell sorting methods, combined with the rapid identification of bacteria[12]. Culturomics have been applied to human gut and skin biofilms, but similar resources for the oral cavity are lacking[9,11–15]. Isolation of oral organisms is difficult, due to their dependence on syntrophic

[1]Department of Prosthetic Dentistry and Biomedical Materials Science, Hannover Medical School, Hannover, Germany. [2]Lower Saxony Centre for Biomedical Engineering, Implant Research and Development (NIFE), Hannover, Germany. [3]Institute of Quantum Optics, Leibniz Universität Hannover, Hannover, Germany. [4]Cluster of Excellence Rebirth (EXC 62), Leibniz Universität Hannover, Hannover, Germany. [5]Unit for Reproductive Medicine, University of Veterinary Medicine Hannover, Hannover, Germany. [6]Center for Translational Studies, University of Veterinary Medicine Hannover, Hannover, Germany. [7]Institute for Food Quality and Food Safety, University of Veterinary Medicine Hannover, Hannover, Germany. [8]Cluster of Excellence RESIST (EXC 2155), Hannover Medical School, Hannover, Germany. [9]Present address: Shanghai Stomatological Hospital & School of Stomatology, Shanghai Key Laboratory of Craniomaxillofacial Development and Diseases, Fudan University, Shanghai, PR China. ✉e-mail: Szafranski.Szymon@mh-hannover.de

links as well as parasitic interactions[16–19]. Traditional techniques for microbial isolation rely on picking randomly distributed well-separated colonies from diverse cultivation media containing a range of growth factors and selective agents[20], sometimes combined with so-called 'helper' strains[17]. However, even these laborious and time-consuming methods are usually unable to reproduce interspecies synergies, as their precision may be poor (e.g., failure to reproduce desired interspecies co-localizations) or the culture conditions may be incorrect (e.g., application of rich media that favor low biodiversity)[21,22]. Moreover, the biomass of oral specimens from a specific site can be very limited.

Microfluidic platforms offer a cost-effective alternative to classical methods, due to their high scalability and throughput[23]. However, microfluidic systems usually rely on liquid cultures that are harder to characterize than microbial colonies[9]. Moreover, it is difficult to separate biofilms into individual cells using flow cytometry, and loss of medium by evaporation can make it difficult to apply microfluidics to slow growing organisms. Another isolation technique, that is optical tweezers, has been used to manipulate individual biological cells for more than two decades, but it suffers from limited throughput[24].

In order to address limitations of aforementioned methods, we have developed a laser bioprinting pipeline that can accelerate, miniaturize and spatially organize culturomics and thus facilitate the cultivation of biofilm-associated oral microorganisms. Laser bioprinting technology is well established for use in the fast and precise biofabrication of mammalian cell constructs and human tissues. This approach is being applied to the production of living materials, including printing in situ on living organisms[25–29] and there have been rapid advances. In contrast to other bioprinting methods[30], laser-assisted approaches do not require the use of nozzles that could be clogged by a biofilm[31]. Depending on the parameters used, it may produce either pure cultures or polymicrobial communities. Precise recovery of small aggregates by bioprinting may allow co-culture of fastidious taxa along with the strains that support their growth. Laser bioprinting of microorganisms has not yet been widely explored and has been mostly limited to a few model organisms and environmental microbiomes[32–34]. Bearing in mind the unique advantages of laser bioprinting, we intend to demonstrate that this approach can be effectively used to culture human oral microbiota. We now present a laser bioprinting platform that enables high-throughput cultivation, management, characterization and isolation of oral microorganisms. Our platform includes the following key elements: (i) a bioprinter that rapidly and precisely transfers biofilm-associated microorganisms from clinical samples to user-defined in vitro arrays of microbial colonies, and a bioink that is optimized for microbial printing, (ii) tailored culture conditions, including nutrients, physicochemical parameters, selective agents, five co-culture settings and membranes, (iii) a wide spectrum of complementary characterization methods for microbial colonies and isolates and (iv) a biobank of biofilms, bioprints and isolates (Fig. 1a). The tested specimens are in vitro cultures and dental plaque samples from either healthy individuals (n = 12) or patients diagnosed with severe peri-implantitis (n = 3). We demonstrate that bioprinting is able to rapidly create complex microbial patterns with high precision and generate arrays of colonies that closely reflect the composition of clinical biofilms at species and sequence variant levels. Co-culture patterns have induced or improved the growth of fastidious microorganisms. We have used microscopy and enzyme assays to guide microbial isolations. To characterize the isolates, we have established an approach for the classification of oral microorganisms that is based on Fourier Transform Infrared spectroscopy (FT-IR) empowered by machine learning, with potential to complement microscopy, 16S rRNA gene profiling and mass spectrometry. Processing of biofilms from four individuals has, so far, generated a total of 249 representative microbial isolates from 14 classes and 124 species. We have shown that diverse isolates from a healthy individual produce virulence traits

that may contribute to early dysbiosis. Peri-implantitis isolates represent genera, which, on average, are responsible for 85.4% of transcriptional activity in clinical biofilms. Overall, our technology combines high-throughput and precise bioprinting on tailored cultivation media with rapid and non-invasive analyses. Thus, due to its simple and robust nature, bioprinting promises to revolutionize future microbiome-based technology for the investigation and management of complex microbial communities in health and disease.

## Results

### Development of laser bioprinting of microorganisms, including printing on membranes and forming co-culture patterns

High-throughput isolation of diverse microorganisms from oral biofilms requires efficient technology for their processing. For this purpose, we have developed laser bioprinting using microorganisms or biofilms suspended in a bioink, usually a sol, which is a precursor to a hydrogel, in order to generate patterns on a solid medium with high precision (Fig. 1a). The bioink layer was placed on a glass donor substrate with a thin layer of material that absorbs laser energy between them. Each focused laser pulse vaporized part of the absorbed layer and directed a picoliter-size droplet of bioprinting material onto the collector substrate. Using this laser bioprinting technique, we have generated a highly organized bioprint array consisting of 441 closely spaced but non-contacting *Staphylococcus aureus* colonies on an area of roughly 4 cm² (Fig. 1b). To reach this high quality of the print and preserve viability of microorganisms, bioink components, laser energy and cell densities were adjusted (Supplementary Fig. 1a, b). Subsequently, we developed settings for rapid bioprinting on glass slides, additional solid media and in liquid media, in Petri dishes and in multi-well plates (Supplementary Fig. 1c, d). The multi-well setup spatially confines microbial microcolonies, effectively preventing the spread of motile microorganisms and diffusion of metabolites/enzymes across print arrays. Filter membranes have been widely applied in classical microbiological techniques. We developed bioprinting on these membranes to improve transfer (by creating portable medium), copying (via membrane stamping), storage (inside a membrane sandwich) and analysis (e.g., by lowering the background fluorescence) of colony arrays (Supplementary Fig. 2a–f). Syntrophic or parasitic interspecies interactions can support the growth of fastidious microorganisms in co-cultures. *Cutibacterium*, *Fusobacterium* and *Staphylococcus* species were inferred to be the most prominent 'helper' strains using custom database summarizing previous experimental studies (Fig. 1c, Supplementary Data 1). These species were consequently employed in our co-cultures. High accuracy and programmability of bioprinting enabled five co-culture settings (Fig. 1d). Two-species bacterium-bacterium and bacteriophage-bacterium relationships were reproduced (Fig. 1d, Supplementary Fig. 3).

### Bioprinting of reference strain collection

We collected detailed information about suitable conditions for the isolation or propagation of oral microorganisms (Fig. 2) and bioprinted reference ink collections comprising 9 microbial phyla, 17 classes, 24 orders, 39 families, 49 genera, and 100 species (Supplementary Fig. 3). The studied viruses, bacteria, and fungi had highly diverse cell envelope structure, tolerance to oxidative and mechanical stress, doubling time and general physiology. The majority of strains grew in anaerobic conditions on Fastidious Anaerobe Agar supplemented with blood (designated as medium MSPS_029, see Methods for all media compositions and designations) and often formed colonies with a characteristic morphology. Occasionally, special conditions, e.g., additional growth factors, presence of 'helper'/host strains, or completely different media were required. Typical hard-to-culture strains belonged to *Fretibacterium*, *Tannerella* or *Treponema* genera. Laser bioprinting on solid medium

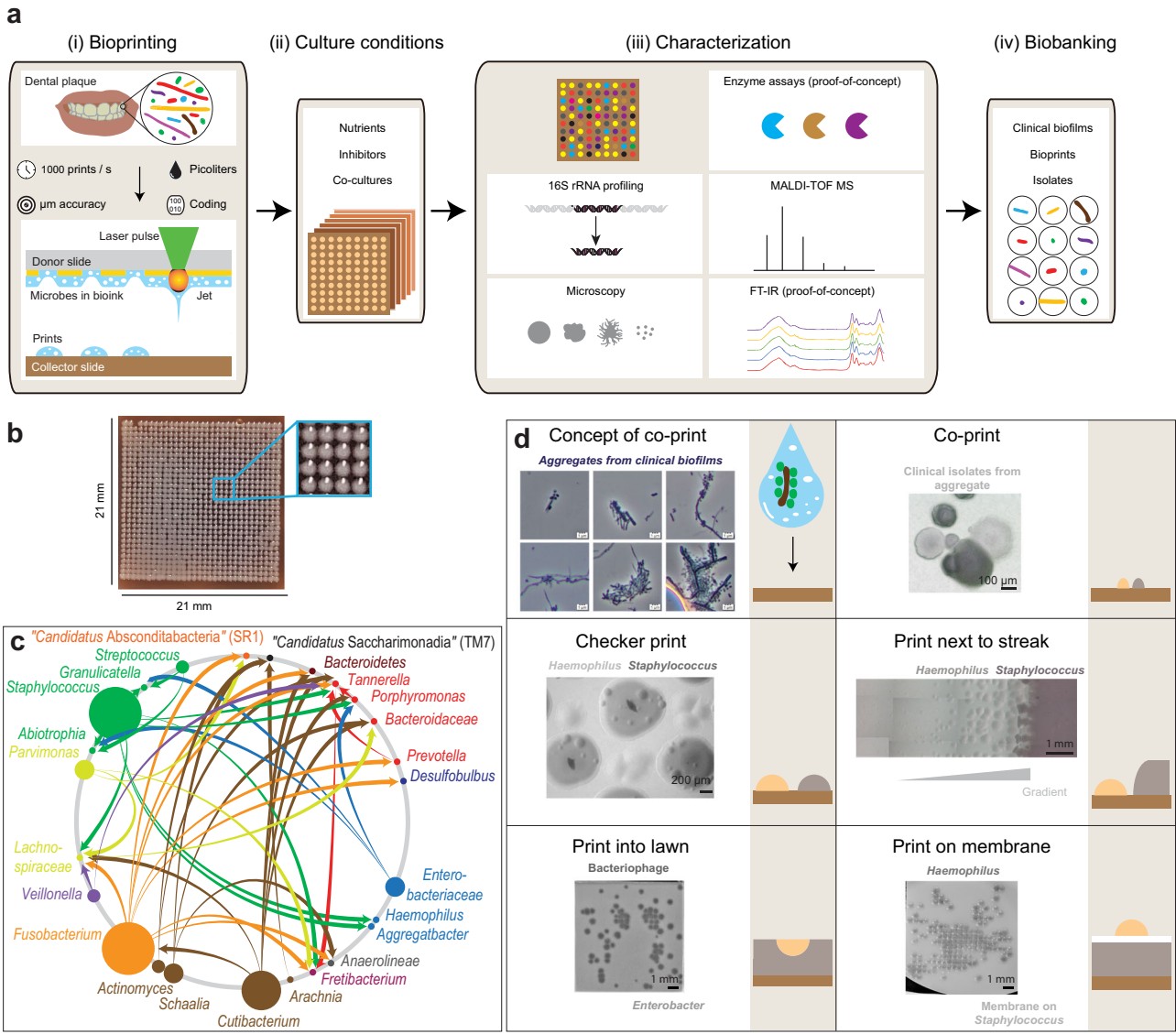

**Fig. 1 | Laser bioprinting of microbial cells. a** Framework of laser-assisted culturomics of the human oral microbiome. **b** Highly organized colony arrays generated with bioprinting. **c** Syntrophic or parasitic interactions reported for oral microorganisms. Arrows connect 'helper' strains and recipients sorted by taxonomy. Node size indicates extent of 'helper' usage, calculated as the number of genera representing the supported species. *Cutibacterium*, *Fusobacterium* and *Staphylococcus* species showed the most connections and have been employed in our co-cultures. Colors represent taxonomy at the class level. For additional details, refer to the Supplementary Data 1, and Source Data. **d** Aggregates in clinical biofilms and five variations of the co-print technique. Co-cultivation of otherwise "unculturable" recipients with "helpers". Printing of model organisms using each co-printing technique was repeated at least twice, yielding consistent results. Model print for each co-print technique was repeated at least twice with similar results.

produced viable and accurate patterns for studied strains, except motile organisms that fused into a continuous lawn (bioprinting into multi-well plates addressed this problem) and *Treponema denticola*, the only studied species that did not survive the printing procedure (due to bioink toxicity).

**Reproducible bioprinting of dental plaque on a single non-selective medium recovered most abundant species and nearly half of the original species richness**

To assess the capability and reliability of bioprinting, we processed human dental plaque, that has become a paradigm for multispecies host-associated biofilms[35]. Plaque samples from 12 healthy volunteers were bioprinted on MSPS_029, the most robust medium. Clinical samples and bioprints were subjected to full 16S rRNA gene amplicons sequencing (Fig. 3a–c, Supplementary Fig. 4a, Supplementary Data 2). Compared to short read sequencing, this method

provides a much higher taxonomic resolution and this is critical for proper evaluation of culturomics outcomes[36]. Sequences originating from non-growing cells were unlikely to influence the results, because input to output cell counts ratio was estimated to be lower than $10^{-4}$ (see Methods) and appropriate abundance cut-offs were introduced during analysis. In comparison to the original clinical biofilms, one week old bioprints were enriched with *Negativicutes* and '*Campylobacteria*' (former *Epsilonproteobacteria)* at the cost of *Actinobacteria*, *Bacilli*, *Betaproteobacteria*, *Gammaproteobacteria*, and *Spirochaetia* classes (Fig. 3a, b). On the species level, 45% of the original richness was reproduced in bioprints (Fig. 3c). Recovery was 79% in the case of abundant species that reached at least 1% of relative abundance in the inoculum. Inocula showed a 30% higher Shannon's diversity index compared to bioprints (Fig. 3c). Results of hierarchical cluster analysis of microbial 16S rRNA gene fingerprints confirmed the reproducibility of bioprints (Supplementary Fig. 4b).

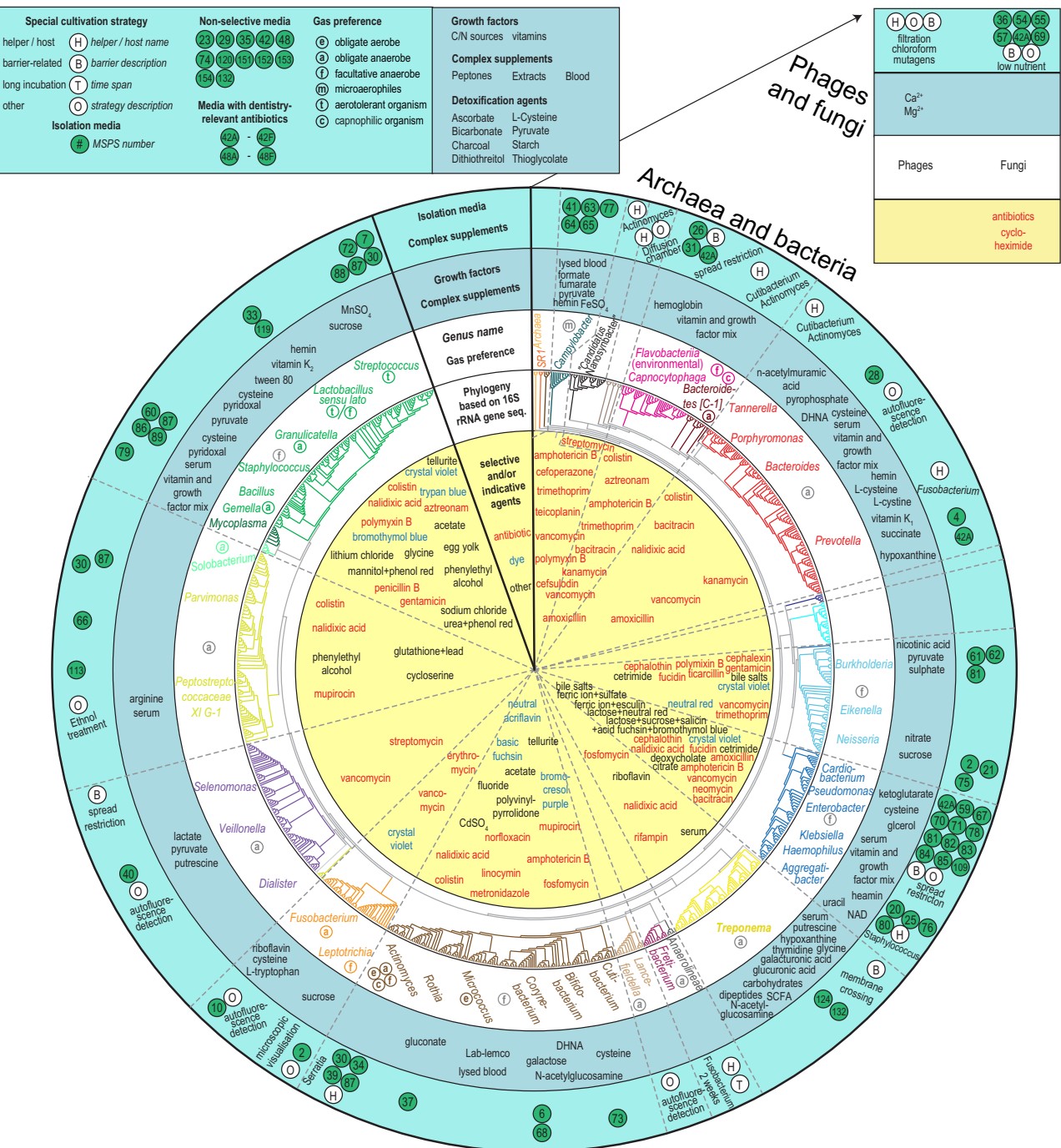

**Fig. 2 | Characteristics of culturomics.** Culture conditions applied in culturomics are summarized in relation to microbial taxonomy. In the outermost ring of a central round graph, the media used for isolation as well as special cultivation strategies are indicated. In a second ring, growth factors and complex supplements are listed. In a third ring, genus names for target species as well as gas preferences are given. Genus names are colored by taxonomy at class level. In a fourth ring, the phylogeny based on 16S rRNA gene sequence is plotted and branches are colored by class. Finally, in the fifth and innermost ring, the selective or indicative agents are listed. Antibiotics are written in red. Dyes are written in blue. Fungi and viruses are depicted on separate small graph, top-right, using the same colors for data presentation. DHNA 1,4-dihydroxy-2-naphthoic acid, NAD Nicotinamide Adenine Dinucleotide, SCFA short-chain fatty acids.

Amplicon sequence variant (ASV)-level analysis (Supplementary Data 3) revealed that bioprinting on MSPS_029 recovered 43% of ASVs (cut-off = 0.1%) present in the original inoculum, with members of 18 taxonomically diverse genera being underrepresented (Supplementary Fig. 5a). The performance of bioprinting on MSPS_029 was further evaluated using rarefaction curves (Supplementary Fig. 6a–c), and a variety of diversity indices across taxonomic levels (Supplementary Data 4), all indicating a consistent trend of approximately 50% microbial recovery. Despite not adjusting for 16S rRNA gene copy number or colony size, this approach offers a reasonable estimate of strain diversity from randomly selected colonies in plaque-derived arrays.

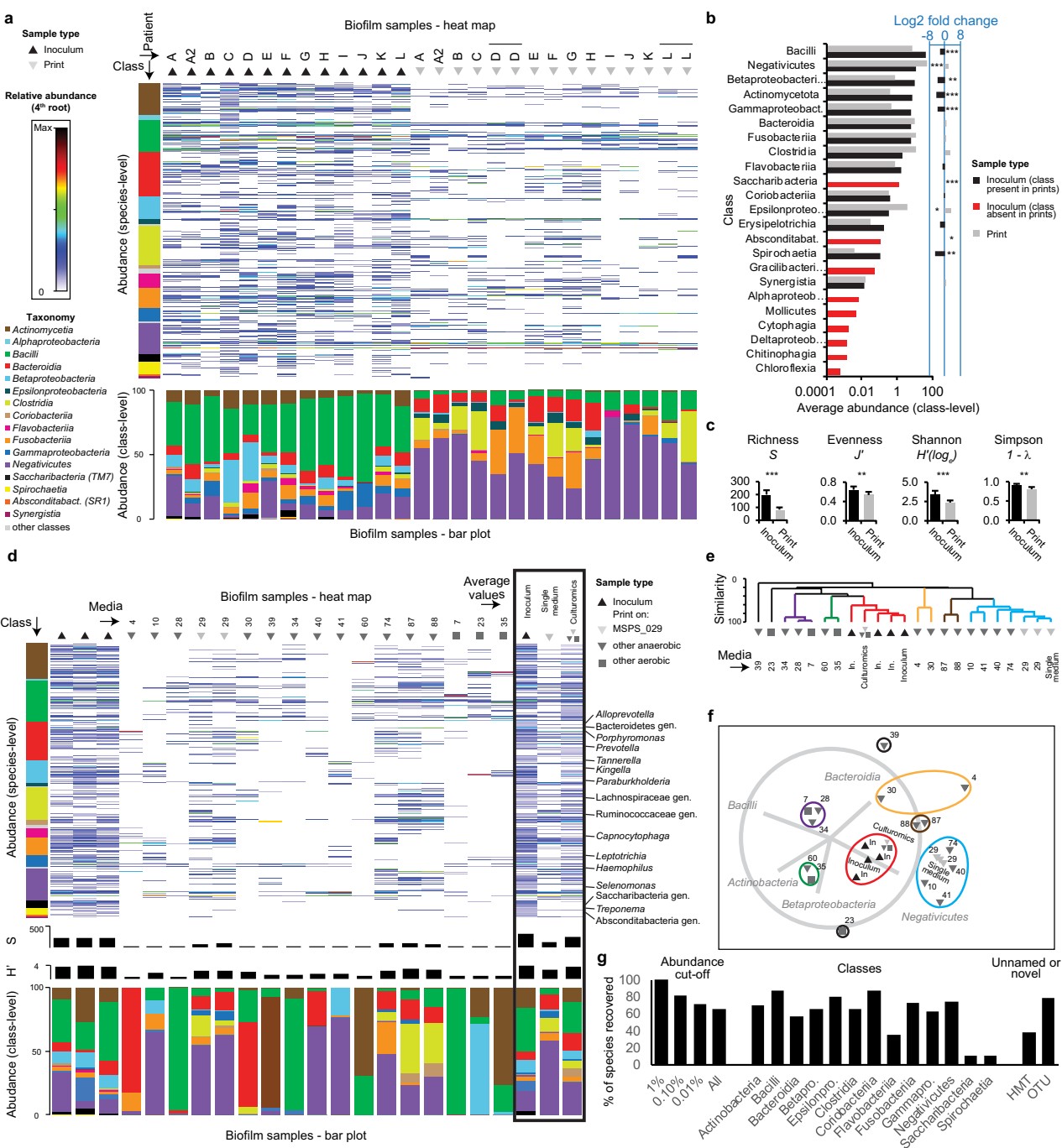

**Fig. 3 | Composition and diversity of bioprint arrays of colonies.** Composition and diversity of complex biofilm inocula and printed colony arrays as assessed by full 16S rRNA gene amplicon profiling. Data for prints on single robust medium and for culturomics are presented in (**a**–**c**), and (**d**–**g**), respectively. **a** Composition of colony arrays obtained on MSPS_029 medium. The species and class levels are shown in the heat map (top) and bar plot (bottom), respectively. Species were sorted by class. **b** Major microbial shifts observed between inocula ($n = 13$) and prints ($n = 15$). Two-sided Mann–Whitney U Test. Symbols * and *** indicate the Bonferroni-adjusted $p < 0.05$ and $p < 0.001$, respectively. **c** Diversity measures compared for inocula ($n = 13$) and prints ($n = 15$). Mean value and standard deviation for Richness (S), Evenness (J'), Shannon index (H'), and Simpson index (1-λ) are depicted. Two-sided Mann–Whitney U Test. Symbols ** and *** indicate the $p < 0.01$ and $p < 0.001$, respectively. **d** Composition of colony arrays obtained by culturomics in comparison to the original samples. Data is shown like in (**a**). Additionally,

the diversity of the colony arrays (S and H') is depicted in the middle area of the graph. Mean values for inoculum, the MSPS_029 medium, and culturomics are shown on the right in the black frame. Labels depict selected genera from which individual species present in the inocula were not recovered by culturomics. Different symbols indicate the sample type or culture condition. **e** Hierarchical cluster analysis of microbial profiles from (**d**). Bray–Curtis similarity values were calculated on standardized abundances of reads grouped to classes. Clusters at an arbitrary similarity of 60% were colored. **f** Non-metric Multi-Dimensional Scaling of microbial profiles from (**d**). Superimposed is a vector plot for classes and clusters from (**e**). 2D-stress was 0.13. **g** Species recovered by laser bioprinting. Percentage recovery is shown by abundance cut-off, by class (for those encompassing at least five species), and for unnamed oral taxa (only possessing HMT number) or for taxa that were not classified at species level (OTUs). For additional details for (**a**–**g**), refer to the Source Data.

## Laser-assisted culturomics tailored for human oral microbiome generates diversity reflecting the clinical situation

In order to improve recovery of microbiota, three samples from a single healthy volunteer were then bioprinted on 16 media (Fig. 3d–g, Supplementary Fig. 4c, d). Full 16S rRNA gene profiling revealed that the media MSPS_029, MSPS_074, MSPS_087, MSPS_088 supported the most diverse bioprint arrays of microbial colonies (Fig. 3d). Selective media complemented each other and favored the expansion of specific bacterial classes– with the exception of "*Candidatus* Saccharimonadia", *Spirochaetia* and few other low-abundant classes. Both hierarchical cluster analysis and non-metric multi-dimensional scaling analysis indicated that the microbial profiles on each single medium were distinct from the clinical fingerprints, yet the overall average composition of culturomics closely resembled the clinical situation (Fig. 3e, f). Diversity indices for inocula and colony arrays measured from all 16 different media combined were comparable (Fig. 3d). At the species level, 65% of the taxa from inocula were recovered by culturomics, compared to only 39% found on the single MSPS_029 medium (Fig. 3g). In case of the more abundant species, culturomics recovered all taxa that reached at least 1% of relative abundance in the inoculum and 82% of taxa that reached at least 0.1%. An important finding was that as much as 38% of unnamed oral taxa that only possessed a Human Microbiome Taxon (HMT) number as well as 79% of operational taxonomic units (OTUs) not classified at the species level were recovered by laser-assisted culturomics. Since these taxa are typically understudied, difficult to isolate, or both, culturomics using laser bioprinting offers a promising approach for recovering oral species that can enhance the representation of microbial diversity in complex biofilm models. Seventy-two very low abundant species were below the detection limit of 16S rRNA gene profiling in the original samples, but were successfully enriched by our method. High recovery of species may be attributed to potential interspecies metabolic interactions. *Porphyromonas pasteri*, an understudied abundant human commensal, is a representative case. It was strongly enriched on MSPS_087 and MSPS_088 media likely due to the specific combination of selective agents and the availability of exploitable metabolites (Supplementary Fig. 4d). Notably, both media lack additional vitamin K supplementation, leading us to hypothesize that naphthoquinone derivatives produced by neighboring species may act as growth factors for *Porphyromonas* spp., which are known to be vitamin K-dependent[37,38]. Using a simple agar plate assay, we were able to reproduce in vitro interactions between a *P. pasteri* strain and two co-present 'helper' strains, supporting this hypothesis and highlighting a promising avenue for future research (Supplementary Fig. 4e).

To specify the performance of individual media used in culturomics, we conducted a preliminary evaluation using diversity analyses across various taxonomic levels (Supplementary Fig. 6e, f, Supplementary Data 4), followed by a more detailed assessment at the ASV level (Supplementary Fig. 5b–e). Overall, laser-assisted culturomics recovered 73% of ASVs (cut-off = 0.1%) present in the original inoculum with only *Lautropia*, *Haemophilus* and "*Candidatus* Saccharimonadia" being major underrepresented components (Supplementary Fig. 5b, Supplementary Data 3). Cultivation under anaerobic conditions on media other than MSPS_029 enhanced the recovery of six major genera, while aerobic culturing was essential for the recovery of three major genera (Supplementary Fig. 5c). Next, we performed a stepwise evaluation using UpSet plots to determine which media best complement each other. Starting with MSPS_029 alone, we iteratively added the most complementary medium at each step. This allowed us to rank the media by their contribution and highlight their unique value (Supplementary Fig. 5d). We further validated these findings using ASV accumulation plots (Supplementary Fig. 5e).

## High-throughput microscopy-assisted taxonomic characterization of colony bioprints

Various optical or biochemical detection methods were implemented as proof-of-principle for bioprint arrays to enable isolation of axenic cultures with the desired characteristics (Fig. 4). Manual microscope-assisted colony-picking was performed. Fluorescence microscopy was used to identify organisms that emit light upon excitation with ultraviolet light or to detect bioprint components marked with fluorescent probe targeting a specific RNA sequence. Plate bioassays allowed macroscopic or microscope-aided detection of specific microbial activities. We selected these methods because they are high-throughput, allow specific detection of taxa or activities, and are usually low- or non-invasive.

Presumptive identification of microorganisms by their colony morphology is a well-established technique. Morphological data of microbial colonies were retrieved for reference strains and clinical isolates which were cultured either on classical plates or as bioprint arrays of microbial colonies. Colony taxonomy was validated using 16S rRNA gene sequencing. Colony morphology was very useful in distinguishing different taxa. The obtained taxonomic resolution ranged from order up to subspecies phylotypes. Colony characterization included shape, margin, size, color, surface appearance, texture, as well as hemolysis, and was strongly dependent on culture conditions. Strains of diverse species characterized by colony polymorphism were repeatedly isolated, e.g., certain members of the genera *Actinomyces*, *Lactobacillus sensu lato*, *Prevotella*, *Shuttleworthia*, *Streptococcus*, and *Veillonella*. Distinct species could produce similar colony morphologies, e.g., point colonies (of diameter <0.5 mm) characterized by their circular shape with an even margin, and no observable special characteristics. Sampling depth was increased at least 5-fold for these colony types, which were indicative of slow growing, more fastidious, usually hard-to-isolate species of a broad phylogeny. Mixed colony transfers were resolved by manually isolating and purifying individual strains through subsequent passages.

Microbial colonies can be differentiated by fluorescence, as certain opportunistic oral pathogens have cellular components that emit fluorescence upon excitation with ultraviolet light[39]. We used this feature for rapid, precise and low-invasive detection and isolation of such taxa, e.g., *Fusobacterium*, *Lancefieldella* and *Prevotella* species (Fig. 4b, Supplementary Data 5). The aforementioned methods allow rapid identification of taxa but are not appropriate for every species. Moreover, their accuracy and taxonomic resolution differ significantly for different phylotypes, *i.e.*, taxonomic groups at different levels of relatedness. As a much more specific yet tedious alternative, we used FISH-CLSM to detect desired taxa at spatial resolution in polymicrobial bioprints. As a proof of concept, we bioprinted 4 species that are critical for the early development of oral biofilms (Fig. 4c). In mixed bioprints, they formed spatially organized merged colonies, where *Streptococcus* sp. formed a core, while *Actinomyces* sp., *Fusobacterium* sp. and *Veillonella* sp. were located peripherally.

## High-throughput biochemical characterization of colony bioprints by enzyme assays

Oral microorganisms release diverse enzymes and metabolic end products that can be either beneficial or detrimental for the human body[40]. In order to perform functional profiling with bioprinting, we implemented biochemical testing on solid media for detection of chondroitin sulfatase, DNases, proteases (including variants for elastase and gelatinase), hyaluronidases, lecithinase, lipase, heparinase, bacteriocin and hydrogen sulfide production (Fig. 4d, Supplementary Fig. 7, Supplementary Data 5). We then processed dental plaque from a healthy individual. *Eikenella corrodens* was enriched due to its ability to produce a chondroitin sulfate sulfatase that digests major glycosaminoglycans found in dental ligament and on the bone surface. DNase, which can act as a defense against host-secreted extracellular DNA

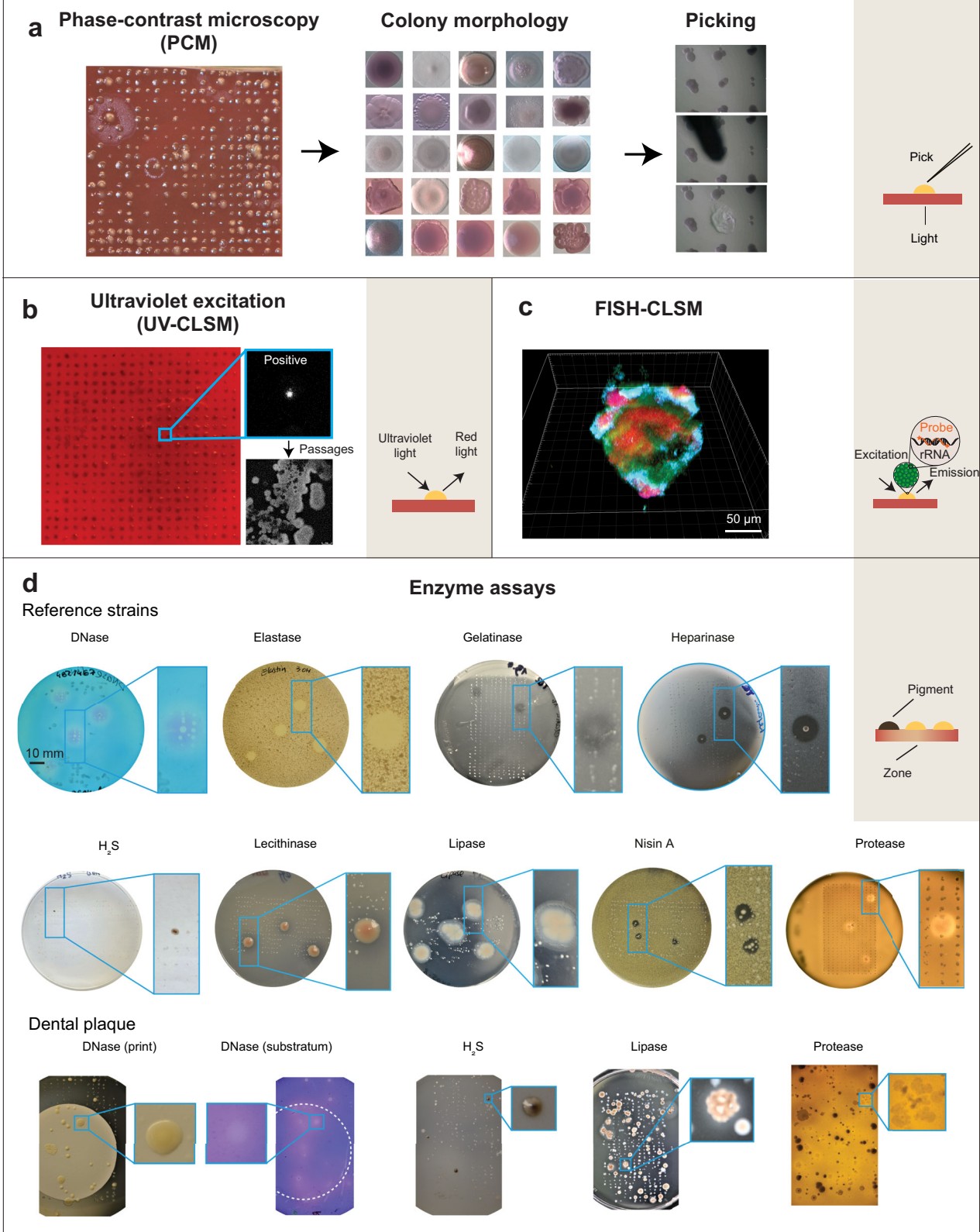

**Fig. 4 | Optical and biochemical characterization of bioprint arrays of colonies.**
**a** Microscopy of colony morphologies and manual passage of selected strains. Mixed colony transfer occasionally occurred, with individual strains separated and purified through subsequent manual passages. **b** Detection of UV-excited light-emitting (in range of 600–650 nm) bioprints. The signal was initially detected in a polymicrobial bioprint. Passaging yielded one UV-excitable isolate, which was responsible for the initial signal. **c** FISH-CLSM was used to characterize the spatial distribution of species in mixed bioprints. Three-dimensional reconstruction. **d** Development and application of assays for detection of specific enzymatic activities in bioprints arrays of colonies on indicative solid media (see SI). Bioprints showing positive signals are enlarged.

(that, like neutrophil extracellular traps, captures and kills pathogens), was detected around colonies formed by *Actinomyces*, *Rothia*, and *Streptococcus* spp. as well as by allochthonous pseudomonads. Proteolytic and lipolytic enzymes, that pre-process nutrients for a biofilm community and can cause damage to the host, were linked to *Actinomyces* and *Streptococcus* species. Hydrogen sulfate can act on human cells as signaling molecule or cytotoxic substance and was found to be produced by *Veillonella* species. Functional profiling with bioprinting indicated members of health-associated biofilms that can potentially contribute to an early shift from health to diseases[3]. Enzyme assays for detecting microbial hydrolases could serve as a quality control measure to identify microorganisms that may disrupt cell encapsulation during the printing process.

### Laser bioprinting generated the representative strain collection for human oral peri-implant diseases, including hard-to-isolate species

Chronic peri-implantitis is a destructive inflammatory process affecting tissues around dental implants and is the most prevalent implant-associated condition worldwide[41]. Its etiology is complex and no extensive culture collection specific to peri-implantitis is currently available. Processing of clinical samples, representing a single healthy patient and three patients with peri-implantitis, yielded tens of thousands of colonies, from which we isolated 249 representative strains (Fig. 5, Supplementary Data 6). Strains were classified using microscopy and either Sanger 16S rRNA sequencing or MALDI-TOF or phenotypic characteristics or a combination of those methods. Additionally, we have established Fourier Transform Infrared spectroscopy (FT-IR) for oral microorganisms to complement the aforementioned methods and to reduce the price of taxonomy assignment (Fig. 5a–d, Supplementary Fig. 8, Supplementary Fig. 9). Infrared signals of microorganisms are highly specific fingerprint-like patterns that can be used for probing the identity of microorganisms down to subspecies level[42]. To validate the method, we created training spectra for five different species of varying degrees of relatedness and used two spectral regions (Fig. 5a) to classify bioprints representing two independent test sets, as visualized with principal coordinate analysis (Supplementary Fig. 8a–c). The highest accuracy - of 100% - was obtained with linear discriminant analysis (LDA) that outperformed two machine learning (ML) approaches: artificial neural network (ANN) and random forest (RF), independently of the studied spectral region (Fig. 5b, Supplementary Fig. 8d, e, Supplementary Fig. 9a–h). Constraint ordinations showed clear groupings for different species (Fig. 5c, d, Supplementary Fig. 9c). The database of oral bacterial FT-IR reference patterns is currently under development.

A total of 124 species representing 14 classes of microorganisms, including bacteria, and fungi were isolated using 30 media (Fig. 5e, Supplementary Note, Supplementary Data 6). 84 strains originated from healthy individual A, 87, 20 and 58 strains came from three individuals having peri-implantitis, namely M, N and O, respectively. When only peri-implantitis isolates were considered, the included genera were responsible for 85.4% of the average transcriptional activity in the clinical biofilms from previous study[43] (Fig. 6, Supplementary Data 7). Due to limitations in taxonomic classification, genus level was the lowest consistently applicable level for our comparative analysis. The rarefaction curve of genera from clinical isolates suggests that recovering additional genera would require substantial effort, additional patient samples, or expanded culture conditions (Supplementary Fig. 6g). We isolated fastidious organisms relevant for oral ecology or pathologies (Supplementary Data 8). Multiple as yet unnamed species or species with poor genomic reference were recovered. For example, we isolated an oral strain of *Colibacter massiliensis* (previously *Megasphaera* sp. HMT-123). Comparative genomics of two *C. massiliensis* strains from distinct habitats revealed strain-specific features (Supplementary Fig. 10). High priority and most wanted taxa of the HMP Consortium, e.g., *Arachnia*

*rubra*, along with seven other taxa of moderate priority were also isolated. The isolated strains together with the reference strains are stored in the Biobank BIT.

## Discussion

Although complex oral and odontogenic biofilm communities are of high relevance for oral and systemic human health, extensive culture collections for oral microorganisms from specific conditions are scarce[8]. This is mostly due to the fastidious and diverse nature of oral microbiota. The past decades have brought significant advances in the cultivation of oral microorganisms[17,18,37,44–50]. However, in contrast to gut microbiota, the concept of assembling a comprehensive collection of all cultivable isolates from a single clinical specimen, known as culturomics, has not yet been adequately addressed[9,12,51]. Classical isolation methods are usually low-throughput, laborious, poorly documented, or they focus exclusively on a narrow group of microorganisms. In this article, we have used the major advantages of bioprinting, i.e., its speed, precision and programmability, to develop robust culturomics for human dental plaque. Our method allows the investigator to transfer several pico- or nanoliters of biofilm, with high spatial accuracy and at a fast rate of, e.g., thousands of bioprints per second, corresponding to the laser pulse repetition rate. Bioprinted patterns can reproduce complex designs at the micrometer scale and are therefore appropriate for diverse applications and analyses. Due to miniaturization, the volume of the consumed medium was reduced by up to two orders of magnitude compared to classical plating. Sample processing greatly benefitted from printing on membranes, which allowed the introduction of a semi-permeable barrier (for syntrophic co-culture), and made the colony array portable (medium replacement for longer incubation, medium analysis), replicable (membrane stamping for multiple independent analysis), storage-friendly (as membrane sandwich) and better analyzable (improved microscopy). Another main advantage of bioprinting is the implementation of interspecies interactions in culturing strategies. In the process of bioprinting, biofilm fragments are micromanipulated, resulting in the retrieval of synergistic consortia of co-localized microorganisms. Our bioprinting also generated five co-culture settings for viruses or fastidious bacteria, by introducing either host or 'helper' strains.

The performance of bioprinting was evaluated with in vitro monocultures of 100 diverse species and complex clinical biofilms of oral origin. High-throughput full 16S rRNA gene sequencing verified high microbial recovery, especially when multiple media were applied. Diverse optical and enzymatic methods were adapted to guide microbial isolations, usually in a high-throughput and low-invasive manner. We showed that FT-IR is an under-appreciated and cost-effective technique that effectively complements 16S rRNA gene profiling and MALDI-TOF MS for taxonomy assignment. Virulent components of the healthy flora were identified and isolated and this suggests that bioprinting may help to achieve early diagnosis of oral or odontogenic dysbiosis. With biofilms from peri-implantitis as an example, laser-assisted culturomics produced culture collections that included unnamed/hard-to-isolate taxa and represented taxa at the genus level that are responsible for a mean of 85.4% of transcriptional activity in clinical biofilms. Thus, this is a valuable resource for building predictive understanding of community function and dynamics[52]. We are currently characterizing the genetics and the physiology of the isolates, and applying bioprinting to achieve a unique perspective on polymicrobial resistance and interspecies interactions in implant-associated biofilms.

In our study, we have explored different culture conditions or enrichment strategies to recover high microbial diversity. However, further settings must be evaluated if we are to isolate additional oral microorganisms, especially bacteriophages, archaea, sulfate-reducing bacteria, epibionts from the "*Candidatus* Saccharimonadia" class, *Treponema* species, lipid-dependent fungi and amoeboid protists[23,53,54].

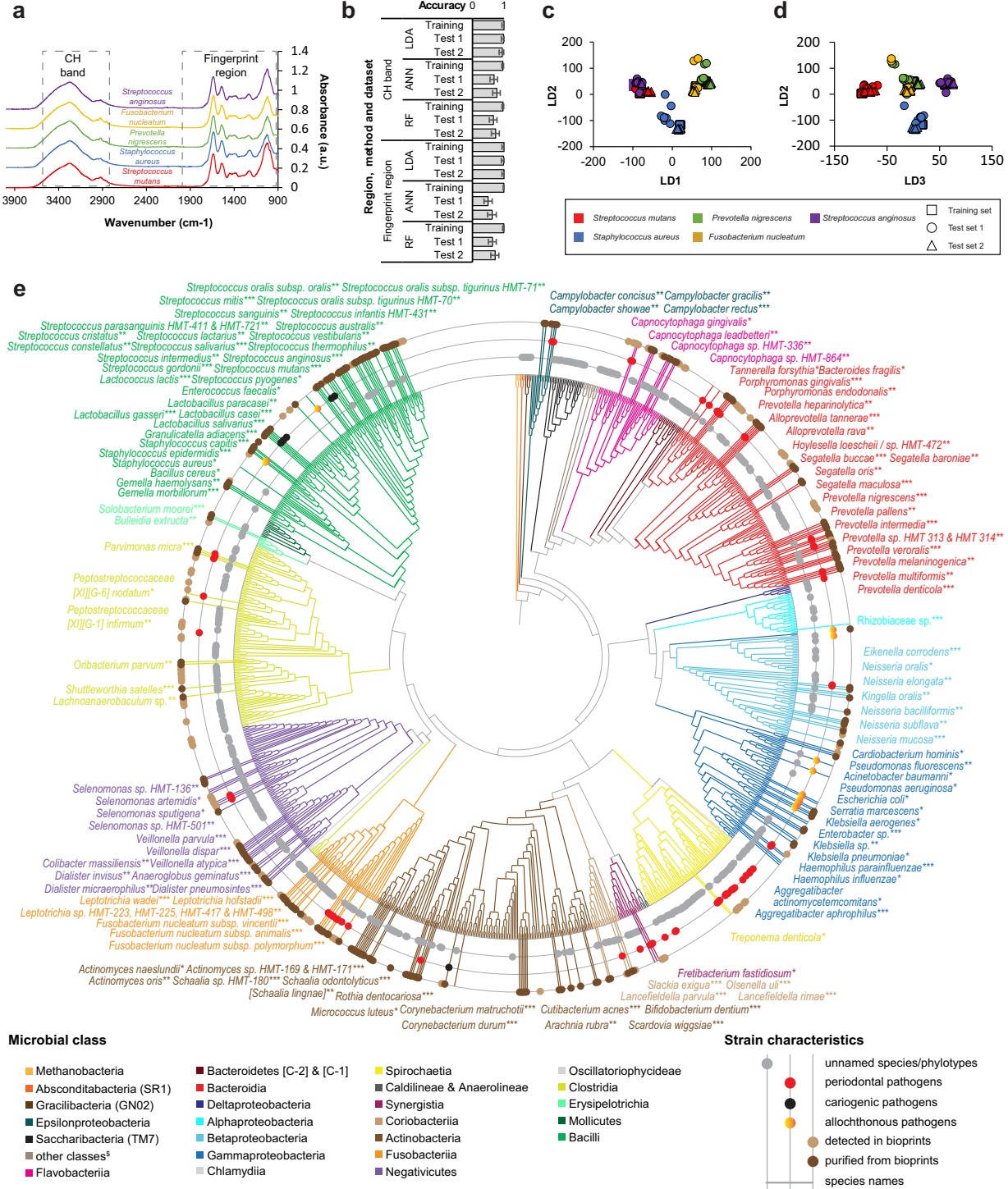

**Fig. 5 | Taxonomy of species detected in bioprints and isolates. a** Full original FT-IR spectra for five representative reference strains: *Fusobacterium nucleatum* SPS_023 representing the *Fusobacteriia* class, Staphylococcus aureus SPS_462 from the *Staphylococcaceae* family from the *Bacilli* class, *Streptococcus anginosus* SPS_004, *Streptococcus mutans* SPS_474, both from the *Streptococcaceae* family from the *Bacilli* class, and *Prevotella nigrescens* SPS_022 from the *Bacteroidia* class. CH band and fingerprint region are marked. **b** Accuracy in species assignment based on vector normalized FT-IR spectra fragments reached by three different classifiers; LDA - linear discriminant analysis, ANN - artificial neural networks, RF - random forest. Data for CH band fragment (2800–3000 cm⁻¹) and fingerprint region (900–1800 cm⁻¹) is presented across datasets. 95% CI are plotted. Data represent *n* = 331 spectra. **c** Constrained LDA ordination for vector normalized FT-IR spectra fragments for fingerprint region generated for five strains. LD1 and LD2

axes are shown. Different symbols indicate training set and two independent test sets. Symbols were colored by taxonomy. **d** Same as c but LD3 axis is plotted instead of LD1 axis. **e** A circular dendrogram based on full 16S rRNA gene phylogeny was generated for oral archaea and bacteria (including allochthonous species) based on 16S rRNA gene sequences. Branches were colored by class. The outer ring summarized the species either only detected in bioprints (light brown) or also isolated (dark brown). The middle ring indicate the clinically relevant species, namely key periodontopathogens, cariogenic species, and allochthonous species. The inner ring indicates the unnamed species that are either poorly characterized or still-to-be isolated. Selected bacterial species that were either bioprinted as reference monocultures (*) or were isolated with bioprinting (**) or both (***) are labeled.). For additional details for (**a**–**e**), refer to the Source Data.

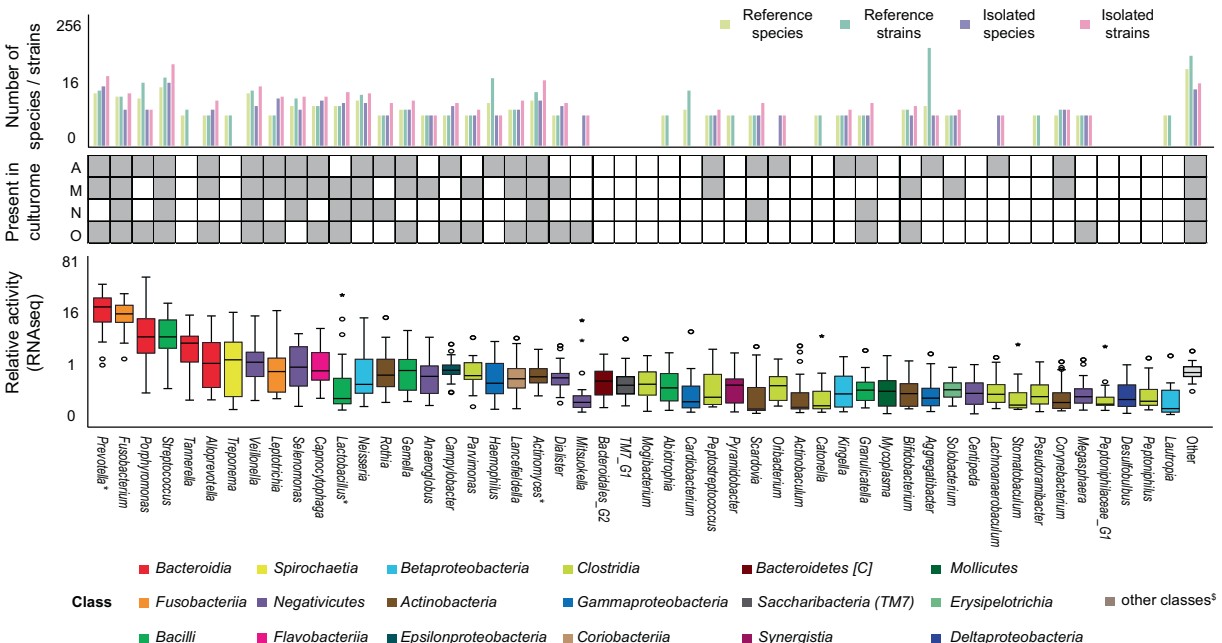

**Fig. 6 | Culture collection to study biofilms from peri-implantitis.** Number of reference species, reference strains, peri-implantitis isolate species and peri-implantitis isolate strains is plotted in the top panel for clinically relevant genera. Reference strains were obtained from other culture collections to cover microbial diversity in peri-implantitis and were used to develop and validate laser-assisted culturomics. Strains from peri-implantitis were isolated in this study. Relative transcriptional activity of genera in peri-implantitis biofilms ($n = 31$, displayed as boxplots with 95% CI, with 'o' and '*' denoting mild and extreme outliers, respectively) was retrieved from previous study[43] and plotted in a bottom panel. Top 50 taxa were sorted by decreasing activity and colored by class. Detailed information can be found in Supplementary Data 7 and in the Source Data.

Supplementary Data 9 summarizes potential reasons for cultivation failures and outlines possible solutions. As previously reported, we found that antibiotics are excellent selective agents, often with unexpected specificity. Therefore, a broader range of antibiotics and their combinations needs to be evaluated[55,56]. The assessment of bioink component toxicity, particularly for *Treponema* species, along with the potential effects of microbial metabolism of these components, needs to be conducted. Our platform could be further expanded by incorporation of biofilm disruption methods such as sonication, bead beating, or chemical/enzymatic treatment[57], integrating an anaerobic atmosphere during printing, generating semi-permeable compartments, implementing automated micromanipulation techniques, and utilizing colony classification based on machine learning[9]. We found that detection of taxon-specific fluorescence is promising, but a broader range of optical conditions and enzyme-specific fluorescence have to be evaluated[58,59]. Moreover, we are developing further chromogenic tests to expand the range of microbial activities that can be detected for colony bioprints. Non-invasive Raman spectroscopy is an interesting alternative to FT-IR, but has not yet been tested for bioprints so far[60]. The potential application of bioprinting for studying natural spatial arrangements[4], particularly its ability to capture interspecies physical interactions[35] (Fig. 1d, top left panel), warrants further exploration[34].

Our bioprinting technology can be used to culture microorganisms from other host-associated and environmental habitats, however, environments rich in motile microorganisms may require printing into spatially confined microcolonies to prevent cross-contamination. It also opens a broad range of microbial applications beyond generation of culture collections. Synthetic living polymicrobial structures have attracted increased attention as components of dynamic biomaterials, but are difficult to generate[61–63]. With our technology, any arbitrary two dimensional microbial pattern can be printed with high precision. The design-build-test-learn cycle has advantages for microbiome engineering and bioprinting seems to be a perfect tool that can be

integrated to address both top-down and bottom-up designs in this pipeline[64]. We envision printing biofilms on-demand for research and clinical use that closely reflect in vivo structures[4].

## Methods
### Statistics & reproducibility
This study focused on development of laser-assisted microbial culturomics using dental plaque as model complex biofilm. Data analysis was performed using R, a free software environment for statistical computing and graphics, and the PRIMER suite with PERMANOVA+, which offers univariate, graphical, and multivariate analysis tools[65,66]. The specific statistical routines used are detailed in the Methods section, where applicable. No statistical method was used to predetermine sample size. Sample sizes for both patients and media were set based on considerations of sequencing depth, biological variability, prior experience, and relevant literature on dental plaque and culturomics. In the case of 16S rRNA gene amplicon sequencing, taxa accumulation and rarefaction curves demonstrated that the number of observed taxa increased with both sample size and sequencing depth before reaching a plateau. The leveling of these curves suggests that the sampling and sequencing efforts were sufficient to comprehensively capture the microbial diversity. Furthermore, Good's coverage index was calculated for each sample at the taxonomic levels to assess sampling completeness[67]. A single 16S rRNA gene amplicon sequencing profile was excluded from analyses due to low sequencing depth. Sample sizes are listed in Methods sections and in figure legends, where applicable. The reproducibility of dental plaque bioprinting was evaluated using hierarchical clustering of 16S rRNA gene amplicon sequencing profiles obtained from printed colony arrays. Randomization and blinding were not applicable to this study.

### Strains and basic culture conditions
Reference strains were selected to represent key members of the oral biofilm community. Additionally, the collection was supplemented

with hard-to-isolate and allochthonous species known to play significant roles in specific clinical population. A total of 461 reference strains were obtained from culture collections and other research groups. 180 strains came from the Culture Collection University of Gothenburg (CCUG), 68 from Leibniz Institute DSMZ German Collection of Microorganisms and Cell Cultures GmbH (DSMZ), 8 from the American Type Culture Collection (ATCC) and 1 from The Czech Collection of Microorganisms (CCM). 177 strains came from M. Kilian (Aarhus, Denmark), 11 from R. Mutters (Marburg, Germany), 4 from N. Jakubovics (Newcastle, UK), 3 from T. Thurnheer (Zürich, Switzerland), 3 from N. Nørskov-Lauritsen (Odense, Denmark), 3 from B. Klein (Boston, USA), 1 from E. Rubalskii (Hannover, Germany), 1 from S. Jepsen (Bonn, Germany) and 1 from Ivo Steinmetz (Graz, Austria). Baker's yeast was a source of the species *Saccharomyces cerevisiae*. The reference species used in this study represent 16 classes of microorganisms, including viruses (marked with *), bacteria, and fungi (marked with **). The following classes were represented (in alphabetic order): *Actinomycetia* (formerly *Actinobacteria*), *Bacilli*, *Bacteroidia*, *Betaproteobacteria*, *Caudoviricetes*\*, *Clostridia*, *Coriobacteriia*, *Epsilonproteobacteria* (currently "*Campylobacteria*"), *Erysipelotrichia*, *Flavobacteriia*, *Fusobacteriia*, *Gammaproteobacteria*, *Negativicutes*, *Saccharomycetes*\*\*, *Spirochaetia* and *Synergistia*. Reference strains were used to establish laser-assisted culturomics. This included testing of microbial cells as a component of bioinks for bioprinting, selecting culture conditions, identifying strains supporting the growth of fastidious organisms, as well as adapting and validating methods for characterizing colony biofilms. For specific applications, see the other subsections of this Methods. In the following paragraphs, the reference species are grouped by class, alphabetized and listed with the identification numbers of the culture collections or strain names and culture conditions. The SPS number is the identification number used in our culture collection within the Biobank BIT[68]. All strains were cultured at 37 °C, if not stated otherwise. They were cultured aerobically, i.e., in an ambient atmosphere enriched with 10% of carbon dioxide, or anaerobically in an atmosphere of 80% nitrogen, 15% carbon dioxide and 5% hydrogen, if not stated otherwise. Standard conditions were defined as anaerobic cultivation on Fastdious Anaerobe Agar (also known as FAA, NCM0014B, Neogen) supplemented with 5% sheep blood (SR0051E, Thermo Scientific), designated MSPS_029. For easy referencing and cataloguing in our institute as well as for partners using our bacterial strains, we use MSPS number as an identification number of cultivation media. Solid media contained 1.4% agar, if not stated otherwise. Semi-solid media, that contained 0.7% agar, and liquid media, where agar was omitted, are indicated with 'SS' and 'L' in superscript, respectively.

The *Actinomycetia* class (formerly *Actinobacteria* class) was represented by 24 strains: *Actinomyces johnsonii* SPS_874 (CCUG 33932, PK 1259), *Actinomyces naeslundii* SPS_870 (CCUG 33928, PK 19), *Actinomyces naeslundii* SPS_876 (CCUG 33972, PK 29), *Actinomyces naeslundii* SPS_533 (ATCC 12104$^T$, CCUG 2238$^T$, DSM 43013$^T$, NCTC 10301$^T$, WVU 45$^T$), *Actinomyces oris* SPS_871 (CCUG 33929, PK 947), *A. oris* SPS_881 (ATCC 43146, CCUG 60842, MG-1), *Actinomyces* sp. SPS_869 (CCUG 33927, T14V), *Actinomyces viscosus* SPS_872 (CCUG 33930, PK 606), *Actinomyces viscosus* SPS_875 (CCUG 33934, LY7), *Alloscardovia omnicolens* SPS_939 (CCUG 47132), *Arcanobacterium haemolyticum* SPS_331, *Bifidobacterium dentium* SPS_892 (ATCC 27534$^T$, CCUG 17378$^T$, AK3$^T$), *Bifidobacterium longum* subsp. *longum* SPS_896 (ATCC 15707$^T$, CCUG 28903$^T$, DSM 20219$^T$, NCTC 11818$^T$), *Corynebacterium matruchotii* SPS_878 (ATCC 33806, CCUG 47160), *Corynebacterium matruchotii* SPS_877 (ATCC 14266$^T$, CCUG 46620$^T$, DSM 20635$^T$, NCTC 10254$^T$), *Cutibacterium acnes* SPS_943 (DSM 108415), *Cutibacterium acnes* SPS_530 (DSM 1897), *Cutibacterium acnes* SPS_546 (ATCC 11828, CCUG 6369), *Cutibacterium acnes* SPS_547 (CCUG 36661), *Cutibacterium acnes* SPS_548 (CCUG 50480), *Micrococcus luteus* SPS_551 (ATCC 10240, DSM 1790, DSM 20490, CCUG

21988, NCTC 7743), *Micrococcus luteus* SPS_550 (ATCC 15307$^T$, ATCC 4698$^T$, DSM 20030$^T$, CCUG 5858$^T$, NCTC 2665$^T$, CN 3475, CCM 169, NCIB 9278), *Rothia dentocariosa* SPS_899 (ATCC 17931$^T$, CCUG 35437$^T$, NCTC 10917$^T$) and *Schaalia odontolytica* SPS_873 (CCUG 33931, PK 984). All strains were cultured in standard conditions, except *Micrococcus* and *Rothia* species, which were cultured aerobically on Columbia agar with sheep blood (PB5039A, Oxoid/Thermo Fisher; designated MSPS_151) or anaerobically on MSPS_029 supplemented with 3 g/L potassium nitrate (Carl Roth, 8001.1), designated MSPS_029B.

The *Bacilli* class was represented by 50 strains: *Abiotrophia defectiva* SPS_894 (ATCC 49176$^T$, CCUG 27639$^T$), *Bacillus cereus* SPS_742 (ATCC 13061, ATCC 13640, DSM 6127, NCTC 9946), *Enterococcus faecalis* SPS_743 (ATCC 19433$^T$, CCUG 19916$^T$, DSM 20478$^T$, NCTC 775$^T$), *Gemella haemolysans* SPS_901 (ATCC 10379$^T$, CCUG 37985$^T$, NCTC 12968$^T$, NCTC 5414$^T$), *Gemella morbillorum* SPS_893 (ATCC 27824$^T$, CCUG 18164$^T$), *Granulicatella adiacens* SPS_895 (ATCC 40175$^T$, CCUG 27809$^T$), *Lactobacillus acidophilus* SPS_448 (ATCC 4356, DSM 20079), *Lactobacillus gasseri* SPS_897 (ATCC 33323$^T$, CCUG 31451$^T$, DSM 20243$^T$), *Lacticaseibacillus* (formerly *Lactobacillus*) *paracasei* subsp. *paracasei* SPS_449 (ATCC 27092, DSM 20312), *Lactococcus lactis* subsp. *lactis* SPS_549 (ATCC 11454, CCUG 21955), *Staphylococcus aureus* SPS_646 (ATCC 35556, DSM 4910), *Staphylococcus aureus* SPS_461 (ATCC 12600$^T$, DSM 20231$^T$, NCTC 8532$^T$), *Staphylococcus aureus* SPS_462 (ATCC 25923), *Staphylococcus aureus* SPS_463 (DSM 11822), *Staphylococcus aureus* SPS_464 (ATCC 6538, DSM 799, NCTC 10788), *Staphylococcus aureus* SPS_465 (ATCC 29213, DSM 2569), *Staphylococcus aureus* SPS_466 (CCUG 47326), *Staphylococcus aureus* SPS_545 (DSM 20232), *Staphylococcus aureus* SPS_802 (DSM 28763, EDCC 5055), *Staphylococcus aureus* SPS_960 (DSM 104437), *Staphylococcus epidermidis* SPS_467 (ATCC 14990$^T$, DSM 20044$^T$), *Staphylococcus epidermidis* SPS_468 (ATCC 35984, DSM 28319), *Staphylococcus epidermidis* SPS_469 (DSM 18857), *S. epidermidis* SPS_961 (DSM 18857), *Streptococcus anginosus* SPS_926 (CCUG 35783, SK 64), *Streptococcus anginosus* subsp. *anginosus* SPS_920 (ATCC 12395$^T$, ATCC 33397$^T$, CCUG 27298$^T$, DSM 20563$^T$, NCTC 10713$^T$), *Streptococcus equi* subsp. *zooepidemicus* SPS_741 (CCM 7316), *Streptococcus gordonii* SPS_923 (CCUG 35758, SK 120, PB 179), *Streptococcus gordonii* SPS_470 (ATCC 33399, DSM 20568, CCUG 18374, NCTC 3165, SK51), *Streptococcus gordonii* SPS_471 (ATCC 35105), *Streptococcus gordonii* SPS_882 (DL1, Challis), *Streptococcus gordonii* SPS_922 (ATCC 10558$^T$, CCUG 33482$^T$, NCTC 7865$^T$, SK 3$^T$), *Streptococcus intermedius* SPS_552 (ATCC 27335$^T$, DSM 20573$^T$, CCUG 32759$^T$, CCUG 17827$^T$, NCTC 11324$^T$), *Streptococcus mitis* SPS_472 (ATCC 49456$^T$, DSM 12643$^T$, NCTC 12261$^T$, SK142$^T$), *Streptococcus mutans* SPS_474 (ATCC 25175$^T$, DSM 20523$^T$, NCTC 10449$^T$), *Streptococcus mutans* SPS_473 (ATCC 700610, UA159), *Streptococcus oralis* SPS_883 (34), *Streptococcus oralis* SPS_476 (ATCC 9811, M7A), *Streptococcus oralis* subsp. *oralis* SPS_475 (ATCC 35037$^T$, DSM 20627$^T$, NCTC 11427$^T$, LVG/I$^T$, PB182$^T$), *Streptococcus parasanguinis* SPS_918 (ATCC 903, CCUG 21026, SK 132), *Streptococcus parasanguinis* SPS_921 (ATCC 15912$^T$, CCUG 30417$^T$, DSM 6778$^T$), *Streptococcus pyogenes* SPS_477 (ATCC 12344$^T$, DSM 20565$^T$, NCTC 8198$^T$), *Streptococcus salivarius* SPS_478 (DSM 20067), *Streptococcus sanguinis* SPS_924 (CCUG 35766, SK 160), *Streptococcus sanguinis* SPS_925 (CCUG 35769, NCTC 10904, SK 4, 804), *Streptococcus sanguinis* SPS_927 (CCUG 59319, SK 49), *Streptococcus sanguinis* SPS_479 (ATCC 10556$^T$, DSM 20567$^T$), *Streptococcus sobrinus* SPS_480 (ATCC 33478$^T$, DSM 20742$^T$, SL1$^T$), *Streptococcus* sp. SPS_928 (CCUG 62640$^T$, SK 95$^T$), *Streptococcus vestibularis* SPS_919 (ATCC 49124$^T$, CCUG 24893$^T$, DSM 5636$^T$, NCTC 12166$^T$). All strains were cultured in standard conditions, or anaerobically on MSPS_151, or on Todd Hewitt (CM189, Oxoid) agar (5210.4, Carl Roth) supplemented with 3% yeast extract (2363.3, Carl Roth), designated MSPS_153. Occasionally, aerobic cultivation was performed for aero-tolerant strains. A cross-streak of a 'helper' strain (*Staphylococcus aureus*, strain SPS_462 or *Cutibacterium acnes*, strain SPS_530 or

*Fusobacterium nucleatum*, strain SPS_447) was included on plates to improve the growth of very fastidious strains.

The *Bacteroidia* class was represented by 35 strains: *Alloprevotella tannerae* SPS_915 (ATCC 51259[T], CCUG 34292[T], NCTC 13073[T]), *Bacteroides fragilis* SPS_485 (ATCC 25285[T], DSM 2151[T], NCTC 9343[T]), *Porphyromonas asaccharolyticus* SPS_784 (ATCC 25260[T], CCUG 7834[T], CCUG 14451[T]), *Porphyromonas catoniae* SPS_785 (ATCC 51270[T], CCUG 41358[T], NCTC 13056[T]), *Porphyromonas endodontalis* SPS_786 (ATCC 35406[T], CCUG 16442[T], NCTC 13058[T], HG 370[T]), *Porphyromonas gingivalis* SPS_452 (Bonn), *Porphyromonas gingivalis* SPS_454 (ATCC BAA-308, W83), *Porphyromonas gingivalis* SPS_455 (ATCC 53978, W50), *Porphyromonas gingivalis* SPS_456 (A7A1-28), *Porphyromonas gingivalis* SPS_787 (CCUG 14449, 381), *Porphyromonas gingivalis* SPS_788 (CCUG 25211), *Porphyromonas gingivalis* SPS_789 (CCUG 25226), *Porphyromonas gingivalis* SPS_790 (CCUG 25837), *Porphyromonas gingivalis* SPS_792 (CCUG 26712), *Porphyromonas gingivalis* SPS_793 (CCUG 27724, FDC 397), *Porphyromonas gingivalis* SPS_803 (DSM 28984, HG 66), *Porphyromonas gingivalis* SPS_453 (ATCC 33277[T], DSM 20709[T]), *Porphyromonas gingivalis* SPS_451 (a derivative strain of DSM 20709[T] that lost pigmentation), *Porphyromonas gingivalis* SPS_791 (CCUG 25839), *Porphyromonas pasteri* SPS_964 (CCUG 37744), *Porphyromonas pasteri* SPS_965 (CCUG 66735), *Porphyromonas* sp. SPS_783 (CCUG 47443), *Prevotella denticola* SPS_912 (ATCC 35308[T], CCUG 29542[T]), *Prevotella intermedia* SPS_457 (ATCC 25611[T], DSM 20706[T]), *Prevotella melaninogenica* SPS_904 (ATCC 25845[T], CCUG 4944[T]), *Prevotella nigrescens* SPS_908 (ATCC 25261, CCUG 9992), *Prevotella nigrescens* SPS_458 (ATCC 33563[T], DSM 13386[T], NCTC 9336[T]), *Prevotella veroralis* SPS_911 (ATCC 33779[T], CCUG 15422[T]), *Hoylesella* (formerly *Prevotella*) *oralis* SPS_930 (ATCC 33269[T], CCUG 15408[T], DSM 20702[T], NCTC 11459[T]), *Segatella* (formerly *Prevotella*) *buccae* SPS_909 (ATCC 33574[T], CCUG 15401[T]), *Segatella* (formerly *Prevotella*) *oris* SPS_910 (ATCC 33573[T], CCUG 15405[T]), *Segatella* (formerly *Prevotella*) *salivae* SPS_916 (CCUG 51934[T], DSM 15606[T]), *Tannerella forsythia* SPS_529 (DSM 102835, W10960), and *T. forsythia* SPS_553 (ATCC 43037[T], CCUG 21028[T], CCUG 33064[T], FDC 338[T]). All strains were cultured in standard conditions, except fastidious *Porphyromonas* strains, which were cultured on MSPS_029 supplemented with 10µg/mL of 1,4-dihydroxy-2-naphthoic acid (also known as DHNA, 281255-25G, Sigma-Aldrich) and designated MSPS_029A, as well as, *Tannerella* strains which were cultured on MSPS_074 flooded with N-acetylmuramic acid (A3007, Sigma-Aldrich, also known as NAM) solution (designated MSPS_074A). 15 µl of NAM stock at a concentration of 2% was evenly distributed on a plate of 10 cm diameter. A cross-streak of a 'helper' strain (*Staphylococcus aureus*, strain SPS_462 or *Cutibacterium acnes*, strain SPS_530 or *Fusobacterium nucleatum*, strain SPS_447) was included on plates to improve the growth of very fastidious strains.

The *Betaproteobacteria* class was represented by 12 strains: *Eikenella corrodens* SPS_903 (ATCC 23834[T], CCUG 2138[T]), *Eikenella* sp. SPS_931 (CCUG 28283), *Kingella kingae* SPS_929 (ATCC 23330[T], CCUG 352[T], NCTC 10529[T]), *Lautropia mirabilis* SPS_898 (ATCC 51599[T], CCUG 34794[T], NCTC 12852[T]), *Neisseria mucosa* SPS_744 (ATCC 9913, CCUG 33779), *Neisseria mucosa* SPS_745 (ATCC 19696[T], CCUG 26877[T]), *Neisseria oralis* SPS_748 (ATCC 25999, CCUG 26878, NCTC 10777), *Neisseria oralis* SPS_528 (DSM 25276[T]), *Neisseria sicca* SPS_917 (ATCC 29256[T], ATCC 49276[T], CCUG 73594[T], DSM 17713[T]), *Neisseria* sp. SPS_747 (ATCC 19243, CCUG 26468), *Neisseria subflava* SPS_746 (ATCC 49275[T], CCUG 23930[T]), *Neisseria subflava* SPS_749 (ATCC 13120[T], CCUG 17913[T], CCUG 345[T], NCTC 8263[T]). All strains were cultured aerobically on MSPS_151. Strain taxonomy was updated based on Bennett, Jolley, Earle, Corton, Bentley, Parkhill and Maiden[69].

The *Caudoviricetes* class were represented by 13 strains. Bacteriophage strains are listed here in the following format [phage genus] [(morphology of the phage)] [genus of the host] [(SPS number of the host)] "phage" [name of the phage] [(SPS number of the phage)]. *Andhravirus* (Podovirus) *Staphylococcus* (SPS_961) phage

vB_SepP_UKE3 SPS_963 (DSM 108058), *Apdecimavirus* (Podovirus) *Enterobacter* (SPS_946) phage vB_EclP-aire SPS_948 (DSM 106789), *Karamvirus* (Myovirus) *Enterobacter* (SPS_945) phage vB_EclM-PT-JD26 SPS_947 (DSM 27525), *Kayvirus* (Myovirus) *Staphylococcus* (SPS_960) phage MRLN SPS_962 (DSM 26857), *Pahexavirus* (Siphovirus) *Cutibacterium* (SPS_943) phage vB_CacS-Bhz19 SPS_944 (DSM 108586), *Pakpunavirus* (Myovirus) *Pseudomonas* (SPS_459) phage JG004 SPS_957 (DSM 19871), *Pbunavirus* (Myovirus) *Pseudomonas* (SPS_956) phage JG024 SPS_958 (DSM 22045), *Przondovirus* (Podovirus) *Klebsiella* (SPS_955) phage vB_KpnP_Lessing SPS_954 (DSM 107143), *Rosenblumvirus* (Podovirus) *Staphyloccocus* (SPS_960) phage EBHT SPS_959 (DSM 26856), *Tequatrovirus* (Myovirus) *Escherichia* (SPS_950) phage vB_EcoM_G2540 SPS_951 (DSM 103895), *Tevenvirinae* (Myovirus) *Klebsiella* (SPS_568) phage vB_KpnM-PT-JD03 SPS_953 (DSM 27027), *Wifcevirus* (Myovirus) *Escherichia* (SPS_949) phage vB_EcoM_WFH SPS_952 (DSM 103160) and unclassified *Enterobacter* (SPS_532) phage SPS_556. Phages were aerobically cultured in semi-solid (0.7% Agar, Carl Roth, 5210.4) Lysogeny broth containing 10 g/L Peptone (Carl Roth, 8986.1), 5 g/L yeast extract (Carl Roth, 2363.3), 5 g/L NaCl (Sigma-Aldrich, 746398), MSPS_035[SS], or Brain heart infusion broth (Oxoid, CM1135), MSPS_152[SS], except *Cutibacterium* phage which was incubated in MSPS_152 supplemented with 0.5 g/L L-Cysteine hydrochloride (T203.2, Roth) 5 mg/L hemin (H5533-1G, Sigma), 5 mg/L histidine hydrochloride monohydrate (1697.1, Roth), and 10 mg/L vitamin K1 (3804.2, Carl Roth). Histidine decreased the rate at which the hemin precipitated from solution[70]. In all cases, semi-solid media contained host cells. Usually, liquid overnight host cultures were diluted 30 times to inoculate semi-solid medium. If the confluence of the bacterial lawn was not satisfactory, the size of the inoculum was changed.

The *Clostridia* class was represented by 5 strains: *Catonella morbi* SPS_933 (ATCC 51271[T], CCUG 33640[T]), *Parvimonas micra* SPS_450 (ATCC 33270[T], DSM 20468[T], CCUG 46357[T]), *Peptostreptococcaceae* [XI] [G-6] [*Eubacterium*] *nodatum* SPS_484 (ATCC 33099[T], DSM 3993[T]), *Pseudoramibacter alactolyticum* SPS_940 (ATCC 23263[T], CCUG 52346[T], DSM 3980[T]) and *Shuttleworthia satelles* SPS_936 (ATCC BAA-774[T], CCUG 45864[T], DSM 14600[T]). The *Coriobacteriia* class was represented by 4 strains: *Lancefieldella parvula* (formerly *Atopobium parvulum*) SPS_914 (ATCC 33793[T], CCUG 32760[T], DSM 20469[T]), *Lancefieldella* (formerly *Atopobium*) *rimae* SPS_913 (ATCC 49626[T], CCUG 31168[T], DSM 7090[T]), *Olsenella uli* SPS_932 (ATCC 49627[T], CCUG 31166[T], DSM 7084[T], VPI D76D-27C[T]) and *Slackia exigua* SPS_934 (ATCC 700122[T], CCUG 44588[T]). The "*Campylobacteria*" (formerly *Epsilonproteobacteria*) class and the *Erysipelotrichia* class were each represented by a single strain: *Campylobacter rectus* SPS_487 (ATCC 33238[T], DSM 3260[T]) and *Solobacterium moorei* SPS_460 (DSM 22971[T]), respectively. The *Flavobacteriia* class was represented by 3 strains: *Capnocytophaga gingivalis* SPS_486 (ATCC 33624[T], DSM 3290[T]), *Capnocytophaga ochracea* SPS_907 (ATCC 27872[T], CCUG 9716[T], DSM 7271[T], NCTC 12371[T]) and *Capnocytophaga sputigena* SPS_906 (ATCC 33612[T], CCUG 9714[T], DSM 3292[T], DSM 7273[T], NCTC 11653[T]). All those strains were cultured under standard conditions. A cross-streak of a 'helper' strain (*Staphylococcus aureus*, strain SPS_462 or *Cutibacterium acnes*, strain SPS_530 or *Fusobacterium nucleatum*, strain SPS_447) was included on plates to improve the growth of very fastidious strains.

The *Fusobacteriia* class was represented by 8 strains: *Fusobacterium nucleatum* subsp. *animalis* SPS_808 (ATCC 51191[T], CCUG 32879[T], NCTC 12276[T]), *Fusobacterium nucleatum* subsp. *fusiforme* SPS_807 (ATCC 51190[T], CCUG 32880[T], NCTC 11326[T]), *Fusobacterium nucleatum* subsp. nucleatum SPS_447 (ATCC 25586[T], DSM 15643[T], CCUG 32989[T]), *Fusobacterium nucleatum* subsp. *polymorphum* SPS_527 (ATCC 10953[T], CCUG 9126[T], DSM 20482[T], NCTC 10562[T]), *Fusobacterium nucleatum* subsp. *vincentii* SPS_805 (ATCC 49256[T], CCUG 37843[T]), *Fusobacterium periodonticum* SPS_804 (ATCC 33693[T], CCUG 14345[T]), *Fusobacterium simiae* SPS_806 (ATCC 33568[T], CCUG 16798[T]) and *Leptotrichia* sp.

HMT-225 SPS_942 (CCUG 60116). All those strains were cultured either under standard conditions, or anaerobically on Chocolate agar with Vitox (Oxoid/Thermo Fisher, PO5090A), designated MSPS_088.

The *Gammaproteobacteria* class was represented by 281 strains: *Acinetobacter baumannii* SPS_567 (ATCC 19606[T], CCUG 19096[T], NCTC 12156[T]), a group of 157 *Aggregatibacter actinomycetemcomitans* strains which were previously characterized[19] and 24 strains (SPS_397 – SPS_420) characterized as invasive in the CCUG database (CCUG 26903, CCUG 29180, CCUG 29772, CCUG 30070, CCUG 30681, CCUG 34586, CCUG 35851, CCUG 35886, CCUG 37000, CCUG 37002, CCUG 37004, CCUG 37006, CCUG 37399, CCUG 37418, CCUG 38577, CCUG 41853, CCUG 41889, CCUG 43274, CCUG 43298, CCUG 44891, CCUG 46863, CCUG 48512, CCUG 51667, CCUG 61155), 9 *Aggregatibacter aphrophilus* strains which were previously characterized[19] as well as 14 invasive strains SPS_420 – SPS_434 (CCUG 414, CCUG 3423, CCUG 11575, CCUG 14377, CCUG 26314, CCUG 34772, CCUG 34940, CCUG 36002, CCUG 36009, CCUG 37472, CCUG 37633, CCUG 37704, CCUG 51586, CCUG 60485, CCUG 3715), 5 *Aggregatibacter segnis* strains which were previously characterized[19], 10 invasive *Cardiobacterium hominis* strains SPS_435 – SPS_444 (CCUG 2711[T], CCUG 19387, CCUG 23288, CCUG 26559, CCUG 27427, CCUG 33794, CCUG 33980, CCUG 46845, CCUG 56945), *Cardiobacterium valvarum* SPS_445 (CCUG 48245[T]), *Citrobacter freundii* SPS_560 (ATCC 8090[T], ATCC 13316[T], CCUG 418[T], NCTC 9750[T]), *Enterobacter cloacae* SPS_945 (ATCC 23355, CCUG 33777, DSM 26481), *Enterobacter cloacae* SPS_946 (DSM 106614, NRZ36863), *Enterobacter cloacae* SPS_535 (CCUG 70660, NCTC 13405), *Enterobacter cloacae* SPS_536 (CCUG 70661, NCTC 13406), *Enterobacter hormaechei* SPS_566 (ATCC 49162[T], CCUG 27126[T]), *Enterobacter hormaechei* subsp. *hoffmannii* SPS_538 (CCUG 58962), *Enterobacter kobei* SPS_537 (CCUG 59627, NCTC 14475), *Escherichia coli* SPS_949 (DSM 101139), *Escherichia coli* SPS_950 (DSM 103260), *Escherichia coli* SPS_395 (ATCC 25922, DSM 1103), *Escherichia coli* SPS_539 (CCUG 59345), *Escherichia coli* SPS_540 (CCUG 59346), *Escherichia coli* SPS_541 (CCUG 59342), *Escherichia coli* SPS_542 (CCUG 52541), *Escherichia coli* SPS_543 (ATCC 35218, CCUG 30600,), *Escherichia coli* SPS_544 (CCUG 52544, NCTC 13353), 10 *Haemophilus influenzae*, 2 *Haemophilus parahaemolyticus*, 9 *Haemophilus parainfluenzae* strains and a single *Haemophilus pittmaniae* strain, which all were previously characterized[19], additionally *Haemophilus parainfluenzae* SPS_446 (CCUG 38942), *Haemophilus parainfluenzae* SPS_891 (ATCC 33392[T], CCUG 12836[T], NCTC 7857[T], HIM 673-1[T]), *Haemophilus parainfluenzae* SPS_900 (CCUG 36129, NCTC 10672) and *Haemophilus parainfluenzae* SPS_902 (CCUG 62655, HK2019), also *Hafnia alvei* SPS_564 (ATCC 13337[T], DSM 30163[T], CCUG 41547[T], NCTC 8105[T]), *Klebsiella aerogenes* SPS_563 (ATCC 13048[T], CCUG 1429[T], NCTC 10006[T]), *Klebsiella pneumoniae* SPS_570 (CCUG 68728, NCTC 13443), *Klebsiella pneumoniae* SPS_571 (CCUG 59348), *Klebsiella pneumoniae* SPS_574 (CCUG 59360), *Klebsiella pneumoniae* SPS_575 (CCUG 59350), *Klebsiella pneumoniae* subsp. *pneumoniae* SPS_955 (ATCC 33495, DSM 11678), *Klebsiella pneumoniae* subsp. *pneumoniae* SPS_569 (CCUG 58545), *Klebsiella pneumoniae* subsp. *pneumoniae* SPS_572 (CCUG 59358), *Klebsiella pneumoniae* subsp. *pneumoniae* SPS_573 (CCUG 59359), *Klebsiella pneumoniae* subsp. *pneumoniae* SPS_568 (ATCC 13883[T], CCUG 225[T], DSM 30104[T], NCTC 9633[T]), *Morganella morganii* subsp. *morganii* SPS_562 (ATCC 25830[T], DSM 30164[T], CCUG 6328[T], NCTC 235[T]), *Proteus hauseri* SPS_780 (DSM 30118), *Providencia stuartii* SPS_565 (ATCC 29914[T], DSM 4539[T], CCUG 14805[T]), *Pseudomonas aeruginosa* SPS_459 (ATCC BAA-47, DSM 19880, PAO1), *Pseudomonas aeruginosa* SPS_834 (ATCC 27853, CCUG 17619, DSM 1117), *Pseudomonas aeruginosa* SPS_956 (DSM 19882, PA14), *Serratia marcescens* SPS_561 (ATCC 13880[T], DSM 30121[T], CCUG 1647[T], NCTC 10211[T]). Strains were aerobically cultured on MSPS_074 (*Aggregatibacter* and *Haemophilus* strains) and on MSPS_035, MSPS_151 or MSPS_152 (all the other strains).

The *Negativicutes* class was represented by 20 strains: *Anaeroglobus geminatus* SPS_935 (CCUG 44773[T]), *Dialister invisus* SPS_938 (CCUG 47026[T], DSM 15470[T]), *Megasphaera micronuciformis* SPS_937 (CCUG 45952[T]), *Selenomonas artemidis* SPS_078 (OMZ 317), *Selenomonas artemidis* SPS_079 (OMZ 530), *Selenomonas artemidis* SPS_067 (ATCC 43528[T], DSM 19719[T]), *Selenomonas* sp. SPS_035 (ATCC 33150[T], DSM 2479[T]), *Selenomonas sputigena* SPS_080 (OMZ 397), *Selenomonas sputigena* SPS_068 (ATCC 35185[T], DSM 20758[T]), *Veillonella atypica* SPS_795 (ATCC 17744[T], CCUG 56974[T], DSM 20739[T], NCTC 11830[T]), *Veillonella criceti* SPS_798 (ATCC 17747[T], CCUG 56973[T], DSM 20734[T], NCTC 12020[T]), *Veillonella denticariosi* SPS_796 (CCUG 54362[T], DSM 19009[T]), *Veillonella dispar* SPS_482 (ATCC 17748[T], DSM 20735[T]), *Veillonella nakazawae* SPS_797 (CCUG 74597[T]), *Veillonella parvula* SPS_800 (ATCC 17745[T], CCUG 5122, DSM 2007, NCTC 11809), *Veillonella parvula* SPS_884 (PK1910), *Veillonella parvula* SPS_483 (ATCC 10790[T]), *Veillonella ratti* SPS_799 (ATCC 17746[T], CCUG 56038[T], DSM 20736[T]), *Veillonella rodentium* SPS_782 (ATCC 17743[T], DSM 20737[T], NCTC 12018[T]), *Veillonella rogosae* SPS_794 (CCUG 54233[T], DSM 18960[T]). All those strains were anaerobically cultured either on MSPS_029 or *Veillonella* medium containing 5g/L Tryptone (Carl Roth 8952.1), 3 g/L yeast extract, 7.5g/L Na-(DL)-lactate (L4263-500ML, Sigma-Aldrich), 0.75 g/L Na-thioglycolate (Sigma-Aldrich, T0632), 1mL/L (V/V) Tween 80 (Carl Roth, 1859.1), 1 g/L Glucose (Carl Roth, 6887.2) and 3 mg/L Putrescine (also known as 1,4-Diaminobutane, Sigma, D13208-25G), designated MSPS_150, or medium MSPS_150 supplemented with 3 g/L potassium nitrate (MSPS_150A). pH of MSPS_150A was adjusted to 7.5 with solid $K_2CO_3$ (P743.1, Carl Roth). A cross-streak of a 'helper' strain (*Staphylococcus aureus*, strain SPS_462 or *Cutibacterium acnes*, strain SPS_530 or *Fusobacterium nucleatum*, strain SPS_447) was included on plates to improve the growth of very fastidious strains.

The *Saccharomycetes* class was represented by *Candida albicans* SPS_888 (ATCC 10231, DSM 1386), *Candida albicans* SPS_889 (ATCC 90028, DSM 11225) and *Sachharomyces cerevisiae* SPS_764, which were aerobically cultured on Czapek DOX agar (CM0097, Oxoid), MSPS_036.

The *Spirochaetia* class was represented by *Treponema denticola* SPS_481 (ATCC 35405[T], DSM 14222[T]), which was anaerobically cultured in or on TYGVS medium[71], designated MSPS_124, or modified Wyss's OMIZ-W1 medium[37], with Val-Lys replaced with Glu-Glu, $ZnSO_4 \cdot 7H_2O$ replaced with $ZnCl_2$ and Lecithin omitted, designated MSPS_132. The following reagents were used to produce MSPS_124: tryptone, brain heart infusion, yeast extract, gelatin (4582.3, Carl Roth), $(NH_4)_2SO_4$ (3746.2, Roth), $MgSO_4 \cdot 7H_2O$ (P027.1, Roth), $K_2HPO_4$ (P749.1, Roth), $KH_2PO_4$ (3904.2, Roth), NaCl, KOH (K018.1, Carl Roth), thiamine pyrophosphate (also known as TPP, C8754-5G, Sigma), glucose, L-Cysteine hydrochloride, sodium pyruvate (8793.1, Roth), acetic acid (glacial, 3738.1, Roth), propionic acid (6026.2, Roth), n-butyric acid (3277.1, Roth), n-valeric acid (V9759-100mL, Sigma), isobutryic acid (I1754, Sigma), isovaleric acid (129542, Aldrich), D,L-methylbutyric acid (49659-1 ML, Merck) and heat-inactivated rabbit serum (R4505-100ML, Sigma) or heat-inactivated horse serum (H1138-500ML, Sigma). The following reagents were used to produce MSPS_132: L-Alanine (3076.1, Carl Roth), L-Arginine monohydrochloride (1689.1, Roth), L-Asparagine (KK37.1, Roth), L-Aspartic Acid (T201.1, Roth), L-Cysteine hydrochloride, L-Glutamic acid (1743.1, Roth), L-Glutamine (3772.1, Roth), Glycine (3790.2, Roth), L-Histidine hydrochloride monohydrate (1697.1, Roth), L-Isoleucine (1698.1, Roth), L-Leucine (1699.1, Roth), L-Lysine hydrochloride (L5626-100G, Sigma), L-Methionine, (1702.2, Carl Roth); L-Ornithine hydrochloride (T204.1, Roth), L-Phenylalanine (1709.2, Carl Roth), L-Proline (1713.1, Carl Roth), L-Serine (1714.1, Carl Roth), L-Threonine (1738.2, Carl Roth), L-Tryptophan (1739.1, Carl Roth), L-Tyrosine (1741.1, Carl Roth), L-Valine (1742.2, Roth), $CaCl_2$ $2H_2O$ (5239.2, Roth), $NaHCO_3$ (6885.2, Roth), KCl (6781.3, Roth), $MgSO_4 \cdot 7H_2O$, $NaH_2PO_4$ monohydrate (K300.1, Roth), $NH_4Cl$ (K298.1,

Carl Roth), CuSO$_4$ (451657-10G, Sigma), MnSO$_4$ H$_2$O (4487.1, Carl Roth), ZnCl$_2$ (3533.1, Carl Roth), FeSO$_4$ 7H$_2$O (P015.1, Carl Roth), Na$_2$SeO$_3$ (1E0Y.1, Carl Roth), NiSO$_4$ 6H$_2$O (1067270100, Sigma), SnCl$_2$ dihydrate (474762-5G, Sigma), NaVO$_3$ (590088-25G, Sigma), (NH$_4$)$_6$Mo$_7$O$_{24}$ 4H$_2$O (09878-25G, Sigma), Cholesterol (8866.1, Roth), Calcium D-(+)-pantothenate (3812.2, Roth), Choline chloride (C1879-500G, Sigma), Meso-inositol (6329.2, Roth), Thiamine hydrochloride (T911.1, Roth), Thiamine pyrophosphate (also known as TPP), Pyridoxal hydrochloride (271748-1G, Sigma-Aldrich), Pyridoxal 5′-phosphate monohydrate (82870-1G, Sigma-Aldrich), D-(+)-Biotin (3822.1, Roth), Folic acid (T912.1, Roth), Folinic acid, calcium salt (47612-250MG, Sigma-Aldrich), Nicotinamide riboside (72345-50G, Sigma), Nicotinic acid (3815.1, Carl Roth), Riboflavin (9607.1, Roth), Vitamin B12 (T915.1, Roth), Nicotinamide adenine dinucleotide (also known as NAD, AE11.2, Roth), Coenzyme A, sodium salt (C3144-100MG, Sigma), Flavine adenine dinucleotide disodium salt (also known as FAD, 5581.1, Carl Roth), Sodium 2-mercaptoethanesulfonate (M1511, Supelco), D,L-alpha-Lipoic acid (T1395-1G, Sigma), Hemin (H5533-1G, Sigma), DHNA, D-glucose, D-fructose (4981.1, Roth), D(+)-Maltose monohydrate (8951.1, Roth), Ascorbic acid (3525.3, Roth), N-Acetylmuramic acid (also known as NAM), Citric acid monohydrate (5110.3, Carl Roth), N-Acetylglucosamine (8993.3, Roth), D-Mannitol (8883.1, Carl Roth), D-Glucuronic acid, sodium salt (2622.2, Roth), D-(+)-Galacturonic acid monohydrate (48280-5g, Sigma), Isobutyric acid, 2-Methylbutyric acid, Valeric acid, Isovaleric acid, Pyruvic acid, sodium salt, Fumaric acid, disodium salt (F1506-25G, Merck), Formic acid, sodium salt (4404.1, Roth), D,L-Lactic acid, sodium salt, Hypoxanthine (6416.1, Roth), Uracil (7288.1, Roth), Thymidine (3005.1, Carl Roth), N-(2-acetamido)−2-aminoethanesulfonic acid (also known as ACES, A9758-100G, Sigma), D,L-Carnitine (7212.1, Roth), Putrescine, Phenol red (P3532-5G, Sigma), and Glu-Glu (G3640-100MG, Sigma).

The *Synergistia* class was represented by *Fretibacterium fastidiosum* SPS_531 (DSM 25557) which was anaerobically cultured on MSPS_029 in the presence of cross-streak of *Fusobacterium nucleatum* SPS_447, and *Pyramidobacter piscolens* SPS_941 (CCUG 55836$^T$, DSM 21147$^T$) which was cultured in standard conditions.

## Laser bioprinting of microbial inks

Laser bioprinting based on the laser-induced forward transfer technique was adapted for microorganisms as based on a protocol for mammalian cells[25]. The setup consists of a pulsed laser and a horizontal plate transparent to the laser radiation; we used a 1 mm thick glass slide here and refer to the plate as the "donor" in the following. This donor is first coated with a thin layer of material that absorbs laser radiation, a 60 nm thick gold layer. A layer of the bioink, to be bioprinted, is then blade-coated on top of the absorption layer. The donor is mounted upside down and the laser pulses are focused through the glass into the absorption layer. This layer is vaporized at the focus, creating an expanding vapor bubble that collapses after a few microseconds. However, due to inertia, the bioink continues to move forward and a bioink jet is formed that lasts for a few hundred microseconds[72]. The bioink jet deposits on a substrate underneath the donor as a droplet. The volume of the droplet depends on the gold and bioink layer thicknesses, the laser pulse energy and on bioink properties such as viscosity and surface tension, and ranges from picoliters to nanoliters. By moving the donor and laser focus relative to the substrate, defined two-dimensional or three-dimensional bioprinting patterns can be generated layer-by-layer. Due to the laser pulses being short and focused, cell-impairing heating is negligible[73].

A Nd:YAG laser (DIVA II; Thales Laser, Orsay, France) was used, with 1064 nm wavelength, approximately 10 ns pulse duration (FWHM) and 20 Hz repetition rate. The laser pulses were focused with a 50 mm achromatic lens into a 40 µm diameter ablation spot. After adjustment to the bioink layer thickness (approx. 60 µm) and viscosity, the laser pulse energy was set at 13.5 µJ, corresponding to a laser fluence between 1 and 2 J/cm². A mixture of 4 parts fibrinogen (from human plasma, mixed at 20 mg/mL in 0.1 M Tris-buffered saline, pH 7.4; Sigma-Aldrich, Deisenhofen, Germany), 2 parts glycerol (Sigma-Aldrich) and 1 part hyaluronic acid (1 wt% hyaluronic acid from *Streptococcus equi* in 0.1 M Tris-buffered saline, pH 7.4; Sigma-Aldrich) was usually used.

## Bioink optimization and printing in multi-well plate format

Four bioink components, commonly used for mammalian cell printing, were evaluated in order to determine the optimum composition of the sol in the microbial ink, which should meet the criteria of sharp print patterns on the substrate, should not inhibit microbial growth and should slow dehydration. In addition, for future planned experiments on 3D printing of biofilms, cross-linking should be possible that meets the aforementioned criteria. Therefore, we investigated the impact of crosslinking on bacterial viability using *Staphylococcus aureus* SPS_462 as a model organism for the following gels: Fibrinogen was crosslinked post-printing using thrombin, rat tail collagen type I was neutralized (inducing a delayed crosslinking) with sodium hydroxide, tenfold concentrated phosphate-buffered saline, and Dulbecco's Modified Eagle Medium prior to printing, with the pH adjusted to 7.2–7.5, while alginate gelation was induced with calcium chloride. However, cross-linking was not applied in the culturomics presented in this study where the droplets were 2D printed. Cells of *Staphylococcus aureus* SPS_462 were printed with either collagen, alginate, fibrinogen or hyaluronic acid on sterilized glass slides as the collector substrate. The incubation took place in a liquid medium, which was a challenge for this technology, as the formation of stable patterns in the liquid phase is much more difficult than in the gas phase due to shear forces. In contrast to bioprints arrays incubated in liquid medium, biofilm colonies printed on a solid medium (e.g., on an agar cube or on a filter membrane placed on an agar plates) are prone to dehydration. Glycerol was introduced at a range of concentrations into the bioink mixture to address this challenge. For incubation periods longer than 4 days, the drying was further reduced by increasing the humidity around the bioprints. This was achieved either by setting the atmosphere controller of the anaerobic workstation or by the use of small incubation chambers into which additional moisture was introduced using wet paper towels. The effect of cell density on the print patterns was characterized using 10$^4$–10$^6$ cells of *Staphylococcus aureus* SPS_462 on a single slide and standard culture conditions. LIVE/DEAD staining combined with fluorescent microscopy was used to characterize cell counts and viability of *Staphylococcus aureus* before and after printing. To ensure the independent and undisturbed growth of the microorganisms in the bioprints, the collector substrate was adapted to a multi-well plate format.

The following strains were printed as monocultures: *Actinomyces naeslundii* SPS_533, *Actinomyces* sp. SPS_016, *Cutibacterium acnes* SPS_530, *Schaalia. odontolytica* SPS_873, *Micrococcus luteus* SPS_551, *Rothia dentocariosa* SPS_899, *Scardovia wiggsiae* SPS_517, *Bifidobacterium dentium* SPS_506, *Corynebacterium matruchotii* SPS_878, *Corynebacterium durum* SPS_861, *Olsenella uli* SPS_048, *Lancefieldella parvula* SPS_510, *Lancefieldella rimae* SPS_507, *Slackia exigua* SPS_706, *Bacteroides fragilis* SPS_485, *Porphyromonas gingivalis* SPS_451, *Tannerella forsythia* SPS_529, *T. forsythia* SPS_553, *Alloprevotella tannerae* SPS_041, *Segatella buccae* SPS_021, *Prevotella denticola* SPS_049, *Segatella maculosa* SPS_050, *Prevotella intermedia* SPS_457, *Prevotella nigrescens* SPS_022, *Prevotella veroralis* SPS_024, *Capnocytophaga leadbetteri* SPS_015, *Capnocytophaga gingivalis* SPS_486, *Bacillus cereus* SPS_742, *Gemella morbillorum* SPS_20, *Staphylococcus aureus* SPS_462, *Staphylococcus epidermidis* SPS_467, *Staphylococcus capitis* SPS_643, *Granulicatella adiacens* SPS_684, *Enterococcus faecalis* SPS_743, *Lactobacillus casei* SPS_490, *Lactobacillus gasseri* SPS_488, *Lactobacillus salivarius* SPS_491, *Lactococcus lactis* subsp. *lactis* SPS_549, *Streptococcus anginosus* SPS_004, *Streptococcus gordonii*

SPS_007, *S. gordonii* SPS_017, *Streptococcus mutans* SPS_474, *Streptococcus mutans* SPS_473, *Streptococcus pyogenes* SPS_477, *Streptococcus salivarius* SPS_478, *Streptococcus* sp. (the Mitis group) SPS_005, *Shuttleworthia satelles* SPS_504, *Parvimonas micra* SPS_450, *Peptostreptococcaceae* [XI][G-6] [*Eubacterium*] *nodatum* SPS_484, *Solobacterium moorei* SPS_522, *Selenomonas artemidis* SPS_067, *Selenomonas sputigena* SPS_080, *Dialister pneumosintes* SPS_012, *Anaeroglobus geminatus* SPS_501, *Veillonella dispar* SPS_013, *Fusobacterium nucleatum* subsp. *vincentii* SPS_023, *Fusobacterium nucleatum* subsp. *animalis* SPS_063, *Leptotrichia hofstadii* SPS_003, *Leptotrichia* sp. (wadei-related) SPS_002, *Fusobacterium nucleatum* subsp. *polymorphum* SPS_527, Rhizobiaceae sp. SPS_811, *Neisseria oralis* SPS_528, *Eikenella corrodens* SPS_010, *Neisseria mucosa* SPS_001, *Campylobacter rectus* SPS_487, *Cardiobacterium hominis* strains SPS_435, *Escherichia coli* SPS_395, *Serratia marcescens* SPS_561, *Klebsiella aerogenes* SPS_563, *Klebsiella pneumoniae* SPS_570, *Enterobacter* sp. SPS_532, *Aggregatibacter aphrophilus* SPS_424, *Aggregatibacter actinomycetemcomitans* SPS_033, *Aggregatibacter actinomycetemcomitans* SPS_404, *Haemophilus influenzae* SPS_262, *Haemophilus parainfluenzae* SPS_446, *Acinetobacter baumannii* SPS_567, *Pseudomonas aeruginosa* SPS_459, *Treponema denticola* SPS_481, *Fretibacterium fastidiosum* SPS_531, *Saccharomyces cerevisiae* SPS_764, [*Candida*] *glabrata* SPS_524 and *Enterobacter* phage SPS_556. In total 103 strains represented 9 phyla, 17 classes, 24 orders, 39 families, 49 genera and 100 species.

### Generation of bioprint arrays' replicas, storage of bioprints arrays and their improved microscopy

*Staphylococcus aureus* SPS_462 was printed on various membranes, which were then placed on a solid medium, to study the microbial growth. The four membranes tested were Whatman qualitative filter paper Grade 1 (CF1), Whatman qualitative filter paper Grade 4 (CF2), hydrophilic polycarbonate membrane (HPM) and hydrophilic polyvinylidene fluoride membrane (PVDF). Membrane-based concepts for bioprint arrays were adapted from few publications[46,74–76]. The effect of pre-wetting the filter membranes with culture medium and of simultaneous printing on multiple layers created by stacking up to three membranes on printing outputs was evaluated. As renewing or changing the medium can be beneficial when isolating or culturing biofilms, *Staphylococcus aureus* bioprints that had been cultured for one day were transferred onto a fresh solid medium to confirm that the filter membranes could be used to improve bioprint transfer. Copying biofilm arrays can be very useful if multiple independent analysis need to be performed to characterize the colony biofilms. In order to duplicate the colony plates, usually a sterile tissue on the stamp is pressed on the top side of the colony plate and transferred on a new plate to produce the duplication. The concept of the stamp was adapted for prints because replication of bioprints enables broader biofilm characterization. *Staphylococcus aureus* strain SPS_462 was primarily bioprinted on medium directly or on HPM membranes (see Supplementary Fig. 2a, b) layered on top of medium. To copy of biofilm arrays either agar matrix or membranes were transferred with sterilized tweezers to a fresh medium and cultured for another day. Three copies were obtained with either the agar matrix or the HMP membrane matrix. The performance of other filter membranes (Supplementary Fig. 2a) was also evaluated during replications of print arrays.

Storage of biofilm arrays for future use is crucial for research, medicine and biotechnology. The long shelf life of printed biofilms would allow them to be analyzed or used later. The effect of preservation on biofilm bioprints was evaluated using *Staphylococcus aureus*, Whatmann membranes (Z240079, Whatman) and preservation fluid (37 g/L BHI, 25% glycerol). A first membrane with printed biofilm arrays was covered with a second membrane. Biofilm arrays sandwiched between membranes were wetted with preservation fluid. After storage at −80 °C for 7 days, the filter membranes with

microorganisms were placed on fresh medium and incubated for 24 h. This suggesting the possibility of storing the prints as long as bacterial stock cultures, for months or even years, depending on a taxon.

Spatial visualization of biofilms (e.g., by using fluorescent dyes and microscopy) is crucial for understanding these complex microbial systems. Furthermore, microscopical imaging, e.g., in combination with specific fluorescent probes, can be especially helpful when biofilms are screened for bacteria with specific characteristics. Applying these techniques to biofilm arrays can considerably reduce both the processing time and the microscopy time needed to perform such imaging-based analysis. Staining procedures for biofilm arrays normally include steps of fixation, lysis and multiple washings, which are usually hard to apply directly on a solid medium. Therefore, membranes and glass slides were evaluated as a substratum (i.e., a carrier material) for optical analysis.

### Participants

The study tested oral biofilms from 15 participants (Supplementary Data 10), including 12 healthy volunteers (A – L) and 3 patients with peri-implantitis[77] (M · O). Healthy volunteers were recruited to provide dental plaque samples for method development. The peri-implantitis cohort forms part of an ongoing clinical cross-sectional study conducted at the Department of Prosthetic Dentistry and Biomedical Materials Science, Hannover Medical School, Germany, within the framework of the 'SIIRI Biofilm Implant Cohort (BIC)'. SIIRI (Safety Integrated and Infection Reactive Implants) is an interdisciplinary research initiative aimed at recruiting hundreds of participants and longitudinally monitoring their peri-implant microbiomes over a period of up to 12 years. The goal is to better understand the onset and progression of peri-implant diseases and to develop strategies for their early detection and prevention. In this study, which prioritized optimizing the isolation and characterization of microorganisms rather than linking microbial data with clinical parameters, sex and/or gender were not considered.

### Sample collection and processing

Sterilized swabs (EH12.1, Carl Roth) were used to collect the supragingival plaque. Biofilms were placed in 1 mL Reduced Transport Fluid[78], also known as RTF, containing 0.45 g/L $K_2HPO_4$, 0.9 g/L NaCl, 0.45 g/L $(NH_4)_2SO_4$, 0.1875 g/L $KH_2PO_4$, 0.1875 g/L $MgSO_4$, 1 mM Ethylenediamine-tetra acetic acid disodium salt dihydrate (also known as EDTA, 8043.2, Carl Roth), 0.4 g/L $Na_2CO_3$ and 0.2 g/L Dithiothreitol (also known as DTT, #R0861, Thermo Scientific). After gentle but thorough mixing by pipetting, 100 μL of each sample was directly stored at −80 °C for DNA isolation and PacBio sequencing of 16S rRNA gene amplicons. 10 μL of sample was utilized for bioprinting, 10 μL of sample was used for microscopy, and the rest of the sample was mixed 1:1 (V/V) with stock medium [37 g/L BHI, 25% glycerol (3783.1, Carl Roth)] to generate glycerol stocks that were stored at −80 °C for further use, e.g., repeated printing or plating. Submucosal biofilms were collected from patients with peri-implantitis with a curette and were processed in the same way as the supragingival samples.

### Solid media for culturomics

Compositions and culture conditions were retrieved from the Manual of Clinical Microbiology[79], Bergey's Manual of Systematic Bacteriology[80], Difco™ & BBL™ Manual: Manual of Microbiological Culture Media[81], the BacDive Database[82] and the references therein. In total, 82 media were tested. Non-selective media were primarily evaluated on the basis of total number of retrieved colonies and number of colony morphotypes observed. Selective media were evaluated for their specificity. 16 media were further assessed by 16S rRNA gene profiling of colony biofilms formed on these substrates.

*Actinomycetia* were targeted using media modified from multiple studies[83–92]. Medium MSPS_006 contained 25 g/L Heart Infusion (REF 238400, BD), 10 g/L D(+)-Galactose (4987.1, Carl Roth), 17 mg/L Bromocresol purple (0317.1, Carl Roth), 14 g/L agar, 100 mg/L Fosfomycin (34089, Sigma-Aldrich), 2 mg/L amphotericin B (Y0000005, Sigma-Aldrich) and was incubated in an aerobic atmosphere with 10% $CO_2$ at 37 °C. Medium MSPS_30 contained 46 g/L FAA, 5% sheep blood, 10 mg/L colistin, 15 mg/L nalidixic acid and was incubated in an anaerobic atmosphere at 37 °C. Medium MSPS_034 contained 46 g/L FAA, 5% sheep blood, 2.5 mg/L metronidazole (M1547-25g, Sigma-Aldrich), 25 mg/L nalidixic acid and was incubated in an anaerobic atmosphere at 37 °C. Medium MSPS_037 contained 1 g/L tryptone, 5 g/L Lab-Lemco powder (LP0029, Oxoid), 10 g/L sodium gluconate (2622.1, Carl Roth), 0.125 g/L sodium fluoride (2618.1, Carl Roth), 10 mg/L colistin, 14 g/L agar and was incubated in an aerobic atmosphere with 10% $CO_2$ at 37 °C. Medium MSPS_39 contained 46 g/L FAA, 0.25 g/L nalidixic acid, 5 mg/L colistin and 5% sheep blood and was incubated in an anaerobic atmosphere at 37 °C.

*Bacilli* were targeted using media modified from two studies[93,94]. Medium MSPS_007 contained 90 g/L Mitis Salivarius Agar (01337-500G-F, Sigma-Aldrich) and 10 g/L Potassium Tellurite (60539-10G, Sigma-Aldrich). De Man, Rogosa and Sharpe broth (also known as MRS, X925.1, Roth) with 1.4% agar was designated as MSPS_033. Both media were used for aerobic cultures. Supplementation with 10 mg/L Pyridoxal hydrochloride (in MSPS_029C, see below) was used to retrieve the most fastidious nutritionally variant streptococci, also known as NVS,[95] in which Cysteine alone does not allow good growth.

*Bacteroidia* were targeted using media modified from two studies[96,97]. Medium MSPS_004 contained 46 g/L FAA, 5% Sheep blood (SR0051E, Thermo Scientific), 100 mg/L Kanamycin (T832.3, Carl Roth), and 7.5 mg/L Vancomycin. Medium MSPS_028 contained 46 g/L FAA, 5% Sheep blood, 15 mg/L Nalidixic acid (CN32.1, Carl Roth,), 10 mg/L colistin and 75 mg/L Bacitracin. Incubation was performed anaerobically.

*Betaproteobacteria* were targeted using media modified from two studies[98,99]. Medium MSPS_002 contained 10 g/L Tryptone (8952.1, Carl Roth), 5 g/L Yeast extract (2363.3, Carl Roth), 10 g/L Sucrose (4621.2, Carl Roth), 3 mg/L Vancomycin (0242.1, Carl Roth), 14 g/L Agar (5210.4, Carl Roth). Medium MSPS_021 contained 36.4 g/L THB, 5 g/L yeast extract, 5% sheep blood + 10 mg/L Hemin, 10 mg/L NAD, 3 mg/L Vancomycin, 10 mg/L Colistin (CN31.1, Carl Roth), 12.5 U/L Nystatin (N6261, Sigma-Aldrich) and 14 g/L Agar. Incubation was performed aerobically.

*Epsilonproteobacteria* (currently "*Campylobacteria*") were targeted using media modified from Macuch and Tanner[100]. Medium MSPS_041 contained 46 g/L FAA, 3 g/L Potassium nitrate (8001.1, Carl Roth), 2 g/L Sodium formate (4404.1, Carl Roth), 4 g/L Sodium fumarate (17013-01-3, Merck), 9 mg/L Vancomycin, 5% sheep blood and was incubated anaerobically.

*Flavobacteriia* were targeted using media modified from two studies[101,102]. MSPS_026 contained 36.4 g/L THB, 3 g/L Yeast extract, 5% Sheep blood, 75 mg/L Bacitracin, 100 mg/L Polymyxin B, 14 g/L Agar and was incubated aerobically. Medium MSPS_031 contained 46 g/L FAA, 10% Cooked sheep blood, Vitox supplement (SR0090A, Oxoid), 3.75 mg/L Colistin, 1 mg/L Vancomycin, 0.5 mg/L Amphotericin B, 1.5 mg/L Trimethoprim and was incubated anaerobically.

*Fusobacteriia* were targeted using media modified from Walker, Ratliff, Muller, Mandell and Socransky[103]. MSPS_010 contained 46 g/L FAA, 5% Sheep blood, 0.2 g/L L-tryptophan (73-22-3, Wako), 5 mg/L Crystal Violet (T123.1, Carl Roth), 4 mg/L Erythromycin (4166.1, Carl Roth) and was incubated anaerobically.

*Gammaproteobacteria* were targeted using media modified from four studies[104–108]. MSPS_020 contained 36.4 g/L Todd Hewitt broth (THB, CM189, Oxoid), 3 g/L yeast extract, 5% sheep blood, 10 mg/L hemin (H9039, Sigma-Aldrich), NAD (AE11.2, Carl Roth), 300 mg/L

Bacitracin (5655.1, Carl Roth) and 14 g/L agar. MSPS_025 contained 36.4 g/L THB, 3 g/L Yeast extract, 10% Horse serum (H1138-500ML, Sigma-Aldrich), 75 mg/L bacitracin, 5 mg/L Vancomycin and 14 g/L agar. UTI chromogenic Agar was designated MSPS_109. Aerobic incubation was performed.

*Negativicutes* were targeted using media modified from Rogosa[109]. Medium MSPS_040 contained 46 g/L FAA, 5% sheep blood, 1 g/L Tween 80 (1859.1, Carl Roth), 0.75 g/L sodium thioglycolate (T0632, Sigma-Aldrich), 2 mg/L basic fuchsin (3256.2, Carl Roth), 1.25% sodium lactate (L4263-500ML, Sigma-Aldrich), 5 mg/L Streptomycin (236.1, Carl Roth) and was incubated in an anaerobic atmosphere.

*Saccharomycetes* were targeted using MSPS_036, MSPS_054 (Malt extract agar, 70145-500G, Sigma-Aldrich), MSPS_055 (Potato dextrose agar, 70139-500g, Sigma-Aldrich), and MSPS_057 (Bismuth Glycine Glucose Yeast Agar, also known as BIGGY agar, 73608-500G, Sigma-Aldrich). Isolations were performed aerobically.

A broad group of anaerobic microorganisms was isolated using media MSPS_029, MSPS_042, MSPS_074, MSPS_120, MSPS_151, MSPS_152, MSPS_153 and MSPS_154. MSPS_042 contained 43.1 g/L Brucella Agar (5752.1, Carl Roth), 5% sheep blood, 5 mg/L hemin, 5 mg/L L-histidine, and 10 mg/L vitamin K1. Additional variants of MSPS_029 medium included supplementation with DHNA (MSPS_029A) or nitrate (MSPS_029B) or nitrate and 100 mg/L pyridoxal hydrochloride (MSPS_029C). MSPS_120 contained 37 g/L BHI, 5 g/L yeast extract, 2 µg/L vitamin K1, 5 mg/L hemin, 0.5 g/L cysteine and 5% each of newborn calf serum (P30-0402, PAN-Biotech), horse serum, and sheep serum (P30-4102, PAN-Biotech). MSPS_154 is a M1 Minimal Agar (ATCC Medium 2511).

A broad group of aerotolerant microorganisms were isolated using media MSPS_023, MSPS_035, MSPS_048 (Mueller Hinton Agar with Horse Blood, Oxoid, PB5303A), MSPS_074, MSPS_151, MSPS_152, MSPS_153 and MSPS_154. MSPS_023 contained 36.4 g/L THB, 3 g/L yeast extract, 5% sheep blood and 14 g/L agar.

Microorganisms resistant/tolerant to specific antibiotics were isolated with MSPS_042A – F. These media were modified MSPS_042 which contained 8 mg/L Amoxicillin (A8523-1G, Sigma-Aldrich), 16 mg/L Metronidazole, 8 mg/L Tetracycline (HP63.1 Roth), 4 mg/L Clindamycin (C5269-10MG, Sigma-Aldrich), 0.5 mg/L Ciprofloxacin (17850-5G-F, Sigma-Aldrich), 2 mg/L Azithromycin (PZ0007-5MG, Sigma-Aldrich) and were incubated anaerobically. A lower concentration of Amoxicillin was also tested (1.6 mg/L, medium designated MSPS_042A2). Similar media were prepared using Mueller Hinton Agar (MSPS_48, CM0405, Oxoid, supplemented with 17 g/L agar, 20 mg/L NAD and 5% sheep blood) as a basis (media designated MSPS_48A – F).

Additional commercial ready-to-use media, which are typically not used for oral microbiology, were obtained from Oxoid: MSPS_059: Aeromonas (PO0325A) targeting *Aeromonas* spp., MSPS_060: Polymyxin Egg Yolk Mannitol Bromothymol blue Agar (also known as PEMBA, PO5048A) targeting *Bacillus* spp., *Staphylococcus* spp., *Serratia* spp., and *Proteus* spp., MSPS_061: Bordetella Selective Medium (PB5065A) targeting *Bordetella* spp., MSPS_062: BURKHOLDERIA CEPACIA AGAR (PO0938A) targeting *Burkholderia* spp., MSPS_063: CAMPYLOBACTER C.A.T. AGAR (PO0839A) targeting *Campylobacter* spp., MSPS_064: Karmali Selective Medium (PO5041A) targeting *Campylobacter* spp., MSPS_065: Skirrow Selective Medium (PO5040A) targeting *Campylobacter* spp., MSPS_066: TSC SELECTIVE AGAR (Tryptose Sulfite Cycloserine Agar) (PO0520A) targeting *Clostridium* spp., MSPS_067: China Blue Lactose Agar (PO5060A) targeting taxa utilizing lactose, MSPS_068: Hoyles Medium (PO0143A) targeting *Corynebacterium* spp., MSPS_069: Dermasel Selective Medium (PO5037A) targeting dermatophyte fungi, MSPS_070: MacConkey Agar (PO5146A) targeting Gram-negative bacteria, MSPS_071: Violet Red Bile Agar with MUG (PO5031A) targeting *Escherichia coli*, ß-D-glucuronidase, MSPS_072: Bile Aesculin Agar (PO0169A) targeting *Enterococcus* spp., MSPS_073: GARDNERELLA SELECTIVE AGAR WITH

SHEEP BLOOD (PB0134A) targeting *Gardnerella* spp., MSPS_075: CHOCOLATE G.C. SELECTIVE AGAR (PB0963A) targeting *Neisseria* spp., MSPS_076: *Haemophilus* Selective Agar (PO5097A) targeting *Haemophilus* spp., MSPS_077: *Helicobacter pylori* selective agar (PB0398A) targeting *Helicobacter pylori*, MSPS_078: *Legionella* BCYEα with L-Cysteine (PO5072A) targeting *Legionella* spp., MSPS_079: *Mycoplasma/Ureaplasma* Agar (PO5081A) targeting *Mycoplasma* spp., MSPS_080: *Pasteurella* Selective Medium (PB5175A) targeting *Pasteurella* spp., MSPS_081: *Pseudomonas* CFC Selective Medium (PO5132A) targeting *Pseudomonas* spp., MSPS_082: Hektoen Enteric Agar (PO5257E) targeting *Salmonella* and *Shigella*, MSPS_083: Desoxycholate Citrate Agar (PO5257E) targeting *Salmonella* and *Shigella*, MSPS_084: S.S. Agar (PO5210E) targeting *Salmonella* and *Shigella*, MSPS_085: X.L.D. Medium (PO5210E) targeting *Salmonella* and *Shigella*, MSPS_086: Baird Parker Agar (PO5014A) targeting *Staphylococcus* spp., MSPS_087: CNA Staph/Strep Selective Agar (PB0308A) targeting *Streptococcus* spp., *Staphylococcus* spp., MSPS_088: Columbia CAP Selective Agar with sheep blood (PB5082A) targeting Gram-positive bacteria as well as MSPS_089: Mannitol Salt Agar (Chapman) (PO5027A) targeting *Staphylococcus* spp., halophilic species. Medium MSPS_088 was flooded with DHNA to obtain medium MSPS_088A.

## Detection of species exhibiting fluorescence under ultraviolet light exposure

To detect specific oral bacteria, such as *Porphyromonas spp.* and *Prevotella spp.* of the *Bacteroidia* class[110], *Veillonella* spp. from the *Negativicutes* class[111,112], and *Fusobacterium* spp. from the *Fusobacteriia* class[79], ultraviolet (UV) light can be used. UV light excites the fluorescence of bacterial cell components, which can be detected by confocal laser scanning microscopy (CLSM). *Prevotella nigrescens*, *Prevotella denticola*, *Porphyromonas gingivalis*, *Fusobacterium nucleatum* subsp. *vincentii*, *Staphylococcus aureus* and *Streptococcus anginosus* strains were used as controls, the latter two species as negative controls for UV-induced fluorescence. Clinical samples were streaked or printed on MSPS_029 and incubated anaerobically. Plates or prints were evaluated with CLSM microscopy (see the CLSM section). To pick colonies from fluorescence-positive isolates, we used either sterilized loops (#86.1562.010, SARSTEDT) or toothpicks, and passaged the bacteria on MSPS_029 medium. In case of mixed colonies, passages were repeated until pure cultures were obtained, but not less than three times.

## Solid and semi-solid media for the detection of biochemical activities

To detect diverse microbial activities at high-throughput, we adapted the existing biochemical tests for oral microbiota and integrated them with printing of complex clinical samples on solid media. Evaluation of tests started with classical streaking of positive and negative reference strains on test media, followed by spotting them into arrays using toothpicks. Media components, incubation times and presence of membranes were adjusted to maximize the positive signal and minimize noise. Optimized parameters were used for printing the inks of the reference positive and negative strains, as well as their mixtures (with positive/negative cell ratio of $10^{-2}$). After further adjustments, final parameters were applied for printing of complex clinical samples. Prints giving a positive signal were purified and taxonomy was assigned to isolates using 16S rRNA gene amplicons sequencing.

To detect hydrogen sulfide, reference strains (*Parvimonas micra* strain SPS_450 as a positive control, *Streptococcus mutans* strain SPS_474 and *Staphylococcus aureus* strain SPS_462 as negative controls, and *Fusobacterium nucleatum* subsp. *vincentii* strain SPS_023 and *Porphyromonas gingivalis* strain SPS_451 as potentially positive strains) were streaked, spotted or printed on MSPS_099. MSPS_099 was adapted from Turng, Guthmiller, Minah and Falkler[113] and contains FAA

supplemented with 4 mM glutathione (6832.2, Carl Roth) 0.2 g/L lead acetate trihydrate (215902-25G, Sigma) and 5% horse serum. FAA contains a mix of peptones (23 g/L) and L-Cysteine HCl (0.5 g/L) that, like glutathione, can be metabolized to hydrogen sulfide, which subsequently reacts with lead acetate to form a visible precipitate of lead sulfide. Incubation was performed in anaerobic atmosphere at 37 °C for three to four days. Streaks or prints which produced hydrogen sulfide were marked by a presence of black precipitate in medium under and/or around prints. Printing on cellulose acetate membranes with a pore size of 0.45 μm (1110650ACN, Sartorius) placed on MSPS_099 was also tested.

To detect DNase activity, reference strains *Staphylococcus aureus* strain SPS_462 (positive), *Staphylococcus epidermidis* strain SPS_467, *Staphylococcus epidermidis* strain SPS_469, and *Enterobacter* sp. strain SPS_532 (all three negative) were streaked, spotted or printed on different membranes (Z240079, Whatman; Z240567, Whatman; 1110650ACN, Sartorius; SVLP04700, Millipore; TCTP04700, Millipore; 11209299001, Roche) placed on MSPS_094[114,115]. MSPS_094 is a DNase Test Agar that was obtained commercially (8295.1, Carl Roth). Incubation was performed in aerobic atmosphere at 37 °C for two days. Subsequently, membranes were transferred into new empty dishes. The MSPS_094 plates from which the membranes had been removed were flooded with Toluidine blue O solution (0.1%, C.I. 52040, 0300.1, Carl Roth), incubated for 10 min at room temperature, and rinsed twice with distilled water. A pink hue denoted the site of DNase-positive cells, whereas blue color denoted presence of polymerized DNA.

To detect heparinase activity, the reference strains (*Bacillus* sp. strain SPS_554 – positive, *Staphylococcus aureus* strain SPS_462 - negative) were streaked, spotted or printed on cellulose acetate membranes with a pore size of 0.45 μm (1110650ACN, Sartorius), and placed on MSPS_097 plates. This medium was adapted from Zimmermann, Langer and Cooney[116] and consists of FAA with heparin sodium salt (1 g/L, 7692.1, Carl Roth). Incubation was performed anaerobically for two days. The MSPS_097 plates from which the membranes had been removed were flooded with protamine sulfate solution (2%, 1.10123, Sigma-Aldrich) and incubated for 1 h at room temperature. A pink hue denoted the site of DNase-positive cells, whereas blue color denoted presence of polymerized DNA. The best visibility of the clear zones that indicated heparinase activity was in transmitted light against a black background.

To detect hyaluronidases activity, reference strains (*Cutibacterium acnes* strain SPS_546 – very strong positive, *Staphylococcus aureus* strain SPS_462 – positive, *Staphylococcus epidermidis* strain SPS_467, *Streptococcus mitis* strain SPS_472, all three negative) were streaked, spotted or printed on cellulose acetate membranes with a pore size of 0.45 μm (1110650ACN, Sartorius) placed on MSPS_098. This medium was adapted from Smith and Willett[117], *i.e.*, it is FAA with a sodium salt of hyaluronic acid (0.4 g/L, 53747-1G, Sigma-Aldrich) and bovine albumin fraction V (0.1 g/L, 2834.2, Carl Roth). Incubation was performed in anaerobic atmosphere at 37 °C for two days. The MSPS_098 plates from which the membranes had been removed were flooded with acetic acid (2N, 3738.1, Carl Roth) and incubated 10 min at room temperature[117]. Clear zones indicated the activity.

To detect chondroitin sulfatase activity, reference strains (the same as those used for detection of hyaluronidases activity) were streaked, spotted or printed on cellulose acetate membranes with a pore size of 0.45 μm (1110650ACN, Sartorius) placed on MSPS_093, which was adapted from Smith and Willett[117]. This medium contains FAA with a sodium salt of chondroitin sulfate (0.4 g/L, C4384 −250MG, Sigma-Aldrich) and bovine albumin fraction V (0.1 g/L, 2834.2, Carl Roth). Incubation was performed in anaerobic atmosphere at 37 °C for two days. The MSPS_093 plates from which the membranes had been removed were flooded with acetic acid and incubated 10 min at room temperature[117]. Clear zones indicated the activity.

To detect bacteriocin activity, combinations of reference producer strains (also in combination of negative strain in cell/cell ratio of 1:100) were streaked, spotted or printed on a lawn of reference indicator strains. Density of the cultures, type of lawn (solid medium flooded with cultures or cultures embedded in semi-solid agar), type of medium, culture conditions, and incubation time was adjusted. *Micrococcus luteus* strain SPS_551 was aerobically grown overnight in liquid MSPS_153 (MSPS_153$^L$, agar was omitted). Next, the culture was diluted 1:25 in the a semi-solid version of the same medium (MSPS_153$^{SS}$, 0.7% agar) and used to overlay fresh MSPS_153 plates (with 1.4% agar). After at 15 min to 1 h of incubation, *Lactococcus lactis* subsp. *lactis* strain SPS_549 (positive control) and *Streptococcus mutans* strain SPS_474 (negative control) was streaked, spotted or printed onto it. Overnight cultures adjusted to $OD_{600} = 0.135$ in 0.85% NaCl, 746398-1KG, Sigma) were used as inoculum. Clear zones of inhibition around colonies/prints indicated Nisin activity.

To detect proteolytic activity, three different tests were applied[118–120]. Elastase activity was detected using reference strains (*Pseudomonas aeruginosa* strain SPS_459 – positive, *Streptococcus mutans* strain SPS_474 - negative). Strains or samples were streaked, spotted or printed on MSPS_095, which is Brain Heart Infusion (BHI, CM1135, Oxoid) supplemented with elastin (10 g/L, E1625, Sigma-Aldrich) and agar (1.4%). Incubation was performed in aerobic or anaerobic atmosphere for up to five days. A zone of clearing around or under cultures indicated activity. Protease activity was detected in the same way but reference strains were plated on medium MSPS_102, in which elastin was replaced with milk powder (15 g/L, T145.1, Carl Roth). Gelatinase activity was detected[120] using the reference strains *Staphylococcus aureus* strain SPS_462 and *Staphylococcus epidermidis* strain SPS 467 (both positive), and *Streptococcus mutans* strain SPS_474 (negative). Strains or samples were streaked, spotted or printed on cellulose acetate membranes[46] with a pore size of 0.45 μm (1110650ACN, Sartorius) placed on MSPS_096, that is FAA with gelatin (10 g/L, 4274.3, Carl Roth). Anaerobic incubation was performed for up to three days. Next, bacteria were transferred on filter to a new plate and the old plate was flooded with 5 mL saturated ammonium sulfate solution (3746.2, Carl Roth). Flooded plates were incubated at 37 °C for 10 min. Saturated ammonium sulfate forms a white, cloudy precipitate with gelatin. Areas where the gelatin had been digested remained clear. Colony biofilms producing gelatinase had a clear halo.

To detect lipase activity, two different tests were applied[121,122]. Lipase and lecithinase activities were detected aerobically using reference strains (*Pseudomonas aeruginosa* strain SPS_459; *Cutibacterium acnes* strains SPS_546, SPS_547, SPS_548, all positive, and *Streptococcus mutans* strain SPS_474 – negative) cultured for up to four days. Strains or samples were streaked, spotted or printed on MSPS_101, which is FAA supplemented with sterile egg yolk emulsion (10%, 0402.1, Carl Roth). The positive cultures had a turbid halo. Additionally, lipase activity was detected on MSPS_100, which contains FAA supplemented with TWEEN 80 (1%, 4859.1, Carl Roth). The positive culture was surrounded by an opaque halo.

**Special isolation technique: inclusion of 'helper' or 'host' strains**
Some species can only grow in the proximity of other species' colonies or streaks. In such cases, so called "helper" strains have to be present during the isolation process. For example, in the past, before the role of nicotinamide adenine dinucleotide was discovered, certain *Haemophilus* species had been cultured around the streak of *Staphylococcus* species[79]. Oral parasitic *Saccharibacteria* can only grow when attached to the cells of the associated host from the class *Actinomycetia*[16,48,49,53]. Other obligate parasites are viruses. Oral bacteriophages can only replicate inside the cell of its host bacterial species[19,54].

The potential for metabolic interactions between oral species, where one species supplies essential factors for another's growth, was

inferred using a custom database. This database is part of the Database for Oral Microbial Interaction Networks (DOMINO), which integrates information on species, enzymes, metabolites, interactions, and references[123]. Built on the Neo4j platform, DOMINO connects to external databases such as the Human Oral Microbiome Database (eHOMD), the Human Metabolome Database (HMDB), the Kyoto Encyclopedia of Genes and Genomes (KEGG), and the U.S. National Library of Medicine's PubMed database. Interaction data were manually curated from literature. A comprehensive search was conducted in PubMed using the terms "fastidious" in combination with "isolation" and "oral". The exact query box entry was *("fastidious"[All Fields] OR "fastidiousness"[All Fields]) AND ("isolate"[All Fields] OR "isolate s"[All Fields] OR "isolated"[All Fields] OR "isolates"[All Fields] OR "isolating"[All Fields] OR "isolation and purification"[MeSH Subheading] OR ("isolation"[All Fields] AND "purification"[All Fields]) OR "isolation and purification"[All Fields] OR "isolation"[All Fields] OR "isolations"[All Fields]) AND ("mouth"[MeSH Terms] OR "mouth"[All Fields] OR "oral"[All Fields]).* A total of 106 articles with full-text availability, published in 1976 or later, were retrieved. Titles and abstracts were manually screened for eligibility. The inclusion criterion was research or review articles addressing the isolation of oral or related species with a focus on co-cultures. References cited in articles were used to cross-validate the primary search results. Additionally, a secondary validation was performed by manually searching for articles that cited the most highly referenced articles identified in the primary search. 25 relevant publications were identifed[16–18,38,44–49,79,80,95,124–135]. Aditionally we screened the databases or websites provided by culture collections: the DSMZ[82] and the CCUG for information on co-cultures used for cultivation of fastidious microorganisms and 4 relevant cases were found. In total, 79 DOMINO entries related to interspecies interactions are now detailed in Supplementary Data 1. Each interaction include information taxonomy of 'helper' and dependent strain, respective links to eHOMD, relevance for oral habitat, strain details, co-culture conditions, comments and references. A custom graph was used to visualize genus-level networks (Fig. 1c). Based on the highest number of interactions, the most robust "helpers" were identifed.

Bacteria that depended on factors that had to be provided by other microorganisms were cultivated together with so-called helper or host strains. Four pairs of either "helper" strains and "satellite" strains (*i.e.*, *Staphylococcus aureus*, strain SPS_462 and *Haemophilus influenzae*, strain SPS_262; *Cutibacterium acnes*, strain SPS_530 and *Tannerella forsythia*, strain SPS_529; *Fusobacterium nucleatum*, strain SPS_447 and *Fusobacterium fastidiosum*, strain SPS_531) or host strain and parasite strain were tested (*Enterobacter* sp., strain SPS_532 and *Enterobacter* phage, strain SPS_556). Three co-culture settings were applied and culture conditions were optimized to address the growth requirements of co-cultured strains. In the first setting, the dependent strain was streaked or spread on a solid medium and the "helper" strains were streaked or spotted on the same plate. Depending on the location on the plate, the cells of both strains were either in contact or spatially separated. Growth factors could diffuse through the agar. Gradients of growth factors were generated around the helper colonies and resulted in satellite growth of dependent strains. In the second setting, a lawn of the "helper" strain was cultured on a solid medium and the dependent strain was streaked on a cellulose acetate membrane with a pore size of 0.45 μm (CAM, 1110650ACN, Sartorius) that was placed on the helper lawn. The strains were physically separated by the membrane but growth factors could transfer through the membrane. Although cells of certain species may be able to cross the filter with pore of this size, this type of filter is broadly used for co-culturing[46]. In the third setting, the host bacterial strain was grown in a layer of semi-solid medium and the dependent bacteriophage was either spotted or streaked on it.

These classical setups for co-cultures were adapted to laser bio-printing. Co-print[34], array-around-streak printing patterns, checkered-

printing patterns and settings with membrane separation were designed to optimize the distance between interacting strains and to increase the interaction space between them. The reference co-culture model used here consists of *Staphylococcus aureus* strain SPS_462 and *Haemophilus influenzae* strain SPS_262. They were cultured alone or together on MSPS_023 or on MSPS_023 supplemented with NAD (designated MSPS_023A). A spot-in-lawn printing pattern (for the obligate parasite, a virus, *Enterobacter* phage strain SPS_556) was applied on a lawn of its bacterial host, *Enterobacter* sp. strain SPS_532. A similar approach was used to culture *Fretibacterium fastidiosum* strain SPS_531 on a lawn of *Fusobacterium nucleatum*.

## Phase Contrast Microscopy (PCM)

Phase contrast microscopy is a valuable tool for the isolation of bacteria based on colony morphology. In this technique, a phase plate shifts the phase of light passing through the sample, producing variations in refractive index to generate contrast in the image. This enables the observation of structures that are normally transparent, such as bacterial cells and their appendages. Phase contrast microscopy is not only non-invasive but also allows biomass samples to be taken under magnification from point colonies that are barely visible macroscopically. We utilize CellSens (Version 1.18, Olympus) to create images of microorganisms, such as microbial colonies (40-fold – 100-fold magnification) and bacterial cells and aggregates (40-fold – 1000-fold magnification). The software connected a USB camera to a phase contrast microscope (Olympus CX41) with phase contrast ring slits (CX-PH1/PH2/PH3) and objective lenses (PlanCN-Ph 10X, 40X, 100XO; SC50, Olympus).

## Confocal Laser Scanning Microscopy (CLSM) and Fluorescence In Situ Hybridization (FISH)

Confocal laser scanning microscopes acquire fluorescent images of thin layers of dense samples (e.g., biofilms) with minimal out-of-focus interference and background noise. Briefly, samples were transferred onto the microscope stage of a confocal laser scanning microscope (TCS SP8, Leica Microsystems, Mannheim, Germany). At each focal plane, images of thin (0.5–1.5 micrometer) optical sections were acquired. A series of images were captured by automatedly moving the objective in the vertical direction. These images were superimposed to provide three-dimensional spatial information about the sample. In the case of microbial cells exhibiting fluorescence under ultraviolet (UV) light exposure, the excitation/emission maxima were 405 nm/600–650 nm. In the case of cells stained with LIVE/DEAD staining (L7012, Thermo Fisher Scientific), the excitation/emission maxima were 480/500 nm for SYTO 9 stain and 490/635 nm for propidium iodide. 5 photos from 5 random positions were taken with a z-step size of 3 µm. In the case of cells stained with FISH probes, the following detectors were used: A Hyd detector was used to measure ALEXA Fluor®405 signals with a 405 nm laser and an emission range of 413–477 nm, as well as ALEXA Fluor®568 signals with a 561 nm laser and an emission range of 576–648 nm. A PMT detector was used to detect ALEXA Fluor®488 signals with a 488 nm laser and an emission range of 509–576 nm as well as ALEXA Fluor®647 signals with a 633 nm laser and an emission range of 648-777 nm. Image stacks with multiple layers of scanning were prepared with a z-step size of 1 µm,. Seven FISH probes[136–140] were used according to the protocol published in Kommerein, Stumpp, Musken, Ehlert, Winkel, Haussler, Behrens, Buettner and Stiesch[141]. Briefly, bioprints on membrane HPM, Isopore Membrane Filter (TCTP04700, Millipore) were fixed using a microwave with the highest power for 1 minute. After washing with DPBS (D8537, Sigma-Aldrich), membranes were fixed with 50% ethanol (8025, J.T.Baker). Fixed bacteria on membranes were incubated in 350 µL lysozyme solution for 10 minutes at 37 °C in a wet chamber for permeabilization. The reaction was stopped by applying 100 µl of 100% ethanol for three minutes, after which the biofilm was air-dried. A mixture of 50 µL of urea-NaCl buffer containing 1 M urea, 0.9 M NaCl, and 20 µM Tris-HCl (pH 7.0), along with 4 µl of 100 µM probe was applied to the samples. Probe MIT588 targeting *S. oralis* with sequence ACA GCC TTT AAC TTC AGA CTT ATC TAA was labeled with ALEXA Fluor®405. Probe ANA103 targeting *A. naeslundii* with sequence CGG TTA TCC AGA AGA AGG GG was labeled with ALEXA Fluor®488. Probe VEI217 targeting *V. dispar* with sequence AAT CCC CTC CTT CAG TGA was labeled with ALEXA Fluor®568. Probe Str405 targeting *Streptococcus* spp. with TAG CCG TCC CTT TCT GGT was labeled with ALEXA Fluor®488. Probe Act476 targeting *Actinomyces* spp. with ATC CAG CTA CCG TCA ACC was labeled with ALEXA Fluor®488. Probe Fus714 targeting *Fusobacterium* spp. with GGC TTC CCC ATC GGC ATT was labeled with ALEXA Fluor®647. Probe Eub338 targeting bacteria with GCT GCC TCC CGT AGG was labeled with ALEXA Fluor®405. The hybridization was conducted for 25 min at 46 °C in a Mini-Incubator 4010 (GFL, Burgwedel, Germany). After the hybridization process, the membranes were washed twice with 100 µL of prewarmed urea-NaCl washing buffer containing 4 M urea, 0.9 M NaCl, and 20 µM Tris-HCl (pH 7.0), and then 100 µL of urea-NaCl washing buffer were applied to the biofilms and incubated for 5 minutes at 48 °C. The washing process was repeated two more times, followed by a final wash with aqua bidest. Finally, the biofilms were covered with 150 µL DPBS and visualized using the sequential imaging mode of the confocal microscope.

## Purification, preservation and classification of isolates

Different non-selective and selective media were used to obtain single colonies of oral isolates (see above). Colonies with unique morphologies were identified for isolation of bacterial strains. Monocultures were created by 2–5 passages of a single colony. When possible (depending on transparency of the culture medium and the expected oxygen tolerance of the strains to be isolated), phase contrast microscopy was applied to evaluate microcolonies and fine morphological differences, e.g., unique coils characteristic for different *Leptotrichia* species. Anaerobic and aerotolerant strains were usually purified on MSPS_029 and MSPS_023, respectively, except strains that failed to grow on these media and required more fastidious conditions (Vitox supplement, sera, 'helper' strains). Mixed colonies were selected based on morphology, imaged via macro-mode photography, phase contrast, or confocal microscopy, manually classified into morphotypes, passaged for purification, and taxonomically identified by 16S rRNA sequencing. Repeatedly observed mixed colonies across multiple passages indicated polymorphism. Strains were preserved as glycerol stocks. Isolates were classified using 16S rRNA gene (or fungal internal transcribed spacer) amplicons sequencing, MALDI-TOF, phenotyping or a combination of these methods.

## DNA isolation, 16S rRNA gene-targeting PCR, Sanger sequencing, and taxonomy assignment

Biomass for DNA isolation was stored in 100 µL of molecular grade water (95284-100 ML, Sigma) at −20 °C until used, but not shorter than 20 min, to introduce the freeze-thaw cycle enhancing cell wall disruption. For Sanger sequencing of bacterial 16S rRNA genes, in most cases, biomass was heated for 30 minutes at 95 °C with mixing at 350 rpms to extract DNA. Lysates were centrifuged at "short" setting to separate the cell debris from the liquid phase. 10 µL of supernatant was used as a template for PCRs targeting the 16S rRNA genes. PCR was performed using KAPA HiFi HotStart ReadyMix (#07958935001, Roche), primers 27F (0.3 µM; AGRGTTYGATYMTGGCTCAG) and 1492R (0.3 µM; RGYTACCTTGTTACGACTT), a final volume of 50 µL, an annealing temperature of 55 °C, a synthesis time of 90 s, and 35 cycles. For the occasional lysates with unsuccesful PCR reactions, the PCR template was diluted 100 fold, or the DNA was further purified using the FastDNA Spin Kit for soil (116560200-CF, MP Biomedicals) or a phenol-chloroform extraction.

For Sanger sequencing of fungal internal transcribed spacer amplicons [ITS, situated between the small-subunit ribosomal RNA (rRNA) and large-subunit rRNA genes], 600 bp DNA fragments of ITS were amplified using primers ITS 1 (F: 5′–GGAAGTAAAAGTCGTAA CAAGG–3′) and ITS 4 (R: 5′–TCCTCCGCTTATTG ATATGC–3′). 35 cycles of PCR were done using DNA samples (5 μL), ITS primers (100 μM, 0.75 μL), molecular grade water, KAPA HiFi HotStart ReadyMix (12 μL), the annealing temperature of 50 °C and the synthesis time of 60 s.

Purity and concentration of PCR products were evaluated using DNA gel electrophoresis [peqGreen (#37-5010, AxonLab), 90 V, 45 min, agarose (1%, A1091, ITW Reagents)] and using the Qubit dsDNA HS Assay Kit (#Q32851, Invitrogen), respectively. PCR products were purified using the MinElute PCR Purification Kit (Qiagen) and eluted in molecular grade water.

For bacterial Sanger sequencing, 300 ng of purified DNA was mixed with 3 μL of either 27F or 1492R primers. For fungal Sanger sequencing, 300 ng of purified DNA was mixed with 3 μL of either ITS 1 or ITS 4 primers. Subsequently, the DNA mixture was sent for Sanger Sequencing (Microsynth, Göttingen, Germany). In case of poor quality or complete failure of sequencing, the process was repeated with additional passages for strain purification or with an alternative primer sets to address mixed cultures and/or poor primer hybridization, respectively. Sequencing chromatograms were manually examined for potential miscalls and to verify that the sequence was truncated properly. Curated sequences underwent taxonomy assignment via an online service provided by the expanded Human Oral Microbiome Database[142]. BLASTn was run with standard parameters (e = 0.0001, no low complexity filter, 20 descriptions, 20 alignments, other: -q −3 −r 2 −G 5 −E 2) on eHOMD 16S rRNA RefSeq database (version 15.22, starts at position 9) as reference. Species names were assigned based on the highest score (bits) for an identity higher or equal to 98.5% and were manually checked. In case of ambiguous assignments, the taxonomy was assigned to the lowest taxonomic level that was common for all references identified as the best match (that must differ by at least two mismatches from other matches). Occasionally, for organisms that were poorly represented in eHOMD, alternative services were used, e.g., the naive Bayesian classifier provided by the Ribosomal Database Project or 16S RefSeq records provided at National Center for Biotechnology Information (NCBI).

## Characterization of microbial community composition with PacBio 16S rRNA gene amplicons sequencing

PacBio 16S rRNA gene amplicons profiling was performed for clinical inocula and printed product biofilm arrays as described in Desch, Freifrau von Maltzahn, Stumpp, Dalton, Yang and Stiesch[143] with some modifications. Briefly, DNA was isolated with the FastDNA Spin Kit for soil (116560200-CF, MP Biomedicals). Almost full length bacterial 16S rRNA genes (i.e., encompassing all variable regions V1 - V9) were amplified with degenerated primers: 27F (0.3 μM; AGRGTTYGA-TYMTGGCTCAG) and 1492R (0.3 μM; RGYTACCTTGTTACGACTT) using KAPA HiFi HotStart ReadyMix (#07958935001, Roche). The following parameters were applied during PCR reaction: after initial denaturation at 95 °C for 3 min, 25 PCR cycles were performed (95 °C for 30 s, 55 °C for 30 s, and 72 °C for 90 s). For two samples that were characterized with lower DNA concentration, the number of cycles was adjusted to 27 to create enough product for the subsequent experimental steps. Finally, synthesis was completed at 72 °C for 10 min and samples were stored at 4 °C until further use. Quality and rough quantity of PCR products were estimated with gel electrophoresis (peqGreen DNA and RNA Dye, peqlab; 1% agarose; 90 V; 45 min; GeneRuler 1 kb ladder). The MinElute PCR Purification Kit (28004, Qiagen) was applied to purify and concentrate the PCR products. The concentration of purified PCR products was measured with the Qubit dsDNA HS Assay Kit (#Q32851, Invitrogen). For multiplexing, symmetrically barcoded SMRTbell Libraries from 32 ng of each PCR product

were synthesized using PacBio Barcoded Overhang Adapters as described in the manufacturer manual entitled "Procedure & Checklist - Preparing SMRTbell® Libraries using PacBio® Barcoded Overhang Adapters for Multiplexing Amplicons" (Part Number 101-791-700 version 06 from Jan 2021). The Amplicons were sequenced using the PacBio Sequel system. The resulting HIFI sequences were filtered at a minimum accuracy of 0.999 and further analyzed using an in-house pipeline. Within this pipeline, TagCleaner 0.16[144] was used to remove primer and barcode sequences, and any sequences with more than one mismatch to a barcode sequence or more than five mismatches to a primer sequence were discarded. Sequences shorter than 1000 bp or longer than 2100 bp were discarded. The remaining sequences were then compared to two different databases, a modified bacteria-only version of the SILVA SSU 132 Ref NR 99 database[145] enriched with Human Oral Microbiome Database eHOMD 16S rRNA RefSeq Version 15.1 sequences[146] and the All-Species Living Tree Project (LTP) database version LTPs132_SSU[147], supplemented with unnamed and phylotype sequences from eHOMD 16S rRNA RefSeq, to determine the species represented by each sequence. Sequences were considered to be assigned to species only if identification based on both databases was unambiguous and consistent. All other sequences were clustered into operational taxonomic units (OTUs) using UPARSE as implemented in USEARCH 10.0.240, with a minimum intra-OTU similarity of 97%[148]. The representative sequences of the OTUs as well as all sequences identified to species level were also classified at the genus lecel and above using RDP classifier 2.13[149] with a minimum bootstrap support value of 0.8. OTUs with representative sequences that were not classified to class level or below were retained only if the SILVA SSU NR 99 database contained at least 50 individual sequences that were at least 95% identical over a length of at least 399 bp. All identified species and OTUs were combined into a final set of species-level taxa for use in statistical analyses. In total 691,199 quality controlled sequences were generated for 48 samples. One sample was excluded due to low sequencing depth. Sequencing depth for the 48 remaining samples ranged from 6640 to 26728 reads with a median of 14467 reads and average of 14693 reads. Sequencing data is available at European Nucleotide Archive (PRJEB75613).

According to our estimation sequences originating from non-growing cells were unlikely to influence the results, because input to output cell counts ratio was likely lower than $10^{-4}$. Briefly, the input cell number per droplet was estimated using phase contrast and fluorescence microscopy to range between $2 \cdot 10^1$ and $2 \cdot 10^2$. The estimated cell output per microbial colony was based on data from a model organism[150], typically around $2 \cdot 10^9$ cells per colony. However, to account for the miniaturized size of our bioprints, approximately 0.5 mm in radius compared to 5 mm in standard *E. coli* colonies, we scaled the output down by a factor of $10^3$, based on the volume difference between spheres of those radii. This results in an estimated output of $2 \cdot 10^6/2 \cdot 10^2$ cells per bioprinted colony.

Statistical analyses were performed with either PRIMER, version 7, or PERMANOVA+ (an add-on package which extends the methods of PRIMER)[65,66]. Non-metric multi-dimensional scaling (nm-MDS) was performed with the PCO PERMANOVA+ routine on the basis of the Bray−Curtis similarity matrix. Input data were first standardized by maximum for variables and next fourth root transformed. Analysis was performed at the species- and class-level. A vector overlay was used to visualize the relationship between the microbial classes and the ordination axes. Each vector begins at the center of a circle (0, 0) and ends at the coordinates (x, y), with x an y indicating a value proportional to the Pearson correlation coefficient (optimized for the physical plot size) between that variable and each of the ordination axes 1 and 2, respectively. The length and direction of the vector indicate the strength and direction, respectively, of the relationship between the variable and the ordination axes. Group-average agglomerative hierarchical clustering was performed on the basis of the Bray−Curtis

similarity matrix for variables describing the biofilm composition. Diversity measures were calculated using the DIVERSE routine. Measures included: Shannon diversity index, species richness ($S$), Pielou's evenness and Simpson index.

Amplicon sequence variant (ASV) analysis was additionally carried out following the approach outlined previously[123]. We utilized high-quality fastq reads (minimum Q30) derived from the same raw sequencing dataset. Demultiplexing and primer removal were performed using the TagCleaner tool. Sequence processing was conducted with the R package dada2 (v3.13)[151], applying the filterAndTrim function with the parameters: minQ = 3, minLen = 1000, maxLen = 2100, maxN = 0, rm.phix = FALSE, and maxEE = 2. Following dereplication, error modeling was conducted using the settings: error-EstimationFunction = PacBioErrfun, randomize = TRUE, BAND_SIZE = 32, and multithread = TRUE. ASVs were inferred using the pseudo-pooling strategy. Species-level taxonomy was assigned where possible using a BLAST-based alignment approach, while higher taxonomic classifications were generated using the RDP classifier with a minimum bootstrap threshold of 80%. These results were further refined by manual curation, referencing unnamed taxa in the expanded Human Oral Microbiome Database (eHOMD)[142]. ASVs lacking species-level identification were additionally aligned to operational taxonomic units (OTUs) using usearch with the usearch_global –id 97 command. Final curation involved removing ASVs classified as chloroplast, eukaryotic, or archaeal, as well as those unclassified at the class level and not assigned to any OTU. ASVs linked to previously excluded OTUs or known contaminants were also removed. To further minimize false positives, any ASVs that did not align to either a species or an OTU were excluded from the final dataset.

Various community structure metrics were computed using the DIVERSE routine in PRIMER 7[66] to assess diversity at the ASV, species, and genus levels. Prior to analysis, read counts were rarefied to 6,070 per sample using the vegan package in R. The calculated diversity indices included total feature count ($S$, also Hill number N0), Margalef's richness index ($d$), Pielou's evenness ($J'$), Brillouin's diversity ($H$), Fisher's alpha, and Shannon entropy ($H'$, log base e). Additional measures included Simpson's diversity ($1-\lambda'$), and Hill numbers: N1 (Exp($H'$)), N2 (1/SI), N∞ (1/Pmax), N10 = (N1/S), N10' = ((N1 − 1)/(S − 1)), N21 = (N2/N1), and N21' = ((N2 − 1)/(N1 − 1)). For comparison, we computed the mean ± standard deviation for inoculum samples (Ino and SD1) and bioprints (29 and SD2), based on a representative MSPS_029 sample (see Supplementary Data 4, highlighted in green). In the case of culturomics, diversity calculations were performed on averaged profiles, Ino and Cul, which were derived by combining each of inoculum and media-specific profiles. This reflects data from three separate inocula per donor and bioprints cultivated across 16 different media types, under the assumption that each medium contributes equally to downstream isolate recovery (Supplementary Data 4, highlighted in blue). Rarefaction at $n$ = 100 [ES(100)] was also used to estimate sampling efficiency. The rarefaction value at $n$ = 100 serves as a proxy for isolation success when 100 colonies are picked, without accounting for variables such as 16S rRNA gene copy number or colony size.

### Full genome sequencing of Colibacter massiliensis SPS_974

Genome sequencing was performed as in ref. 19, with modifications. *C. masiliensis* cells were lyzed with mutanolysin (50 U/mL) and lysozyme (2.5 mg/ml) in TE buffer. DNA was purified with MagAttract (QIAGEN) and sheared with g-TUBEs (Covaris) by processing 1500 µg DNA in 150 µl solution at 5500 rpm for 2 min. The sequencing library was prepared using SMRTbell® Express Template Prep Kit 2.0 (for 10 kb library) and sequenced on a PacBio Sequel machine. The genome was assembled from HighFi reads within SMRT Link 10.1. SPS_974 genome were annotated along reference genome of strain Marseille-P2911[152] using RAST server[153,154] and uploaded to the Type (Strain) Genome Server[155]. Genome sequence is available under Accession: PRJNA1297070 [https://www.ncbi.nlm.nih.gov/bioproject/1297070].

### RNA Sequencing-based evaluation of culture collection

To illustrate how well our strain collection represents microbial genera active in peri-implantitis biofilms we used RNAseq data generated in our previous study[43]. Briefly, submucosal plaque and peri-implant crevicular fluid samples were collected using sterile paper points and a titanium curette, preserved in RNAprotect, and stored at −80 °C. Total RNA was extracted through enzymatic and mechanical lysis, followed by RNA purification, rRNA depletion, cDNA synthesis, and short-read single-end sequencing on the Illumina platform (50 cycles). Quality-controlled reads were mapped to reference genomes of oral microorganisms[142]. Reads were aggregated at the genus level, ranked by mean relative activity in biofilms, and the 50 most active genera were visualized using box plots. We match those genera with information on our culture collection, including numbers of reference strains, reference species, isolated species and strains. Information was summarized in Fig. 6 and Supplementary Data 7.

### Fourier Transform Infrared (FT-IR) spectroscopy

For the training set, bacterial biomass (*Streptococcus anginosus* SPS_004, *Prevotella nigrescens* SPS_022, *Fusobacterium nucleatum* subsp. *vincentii* SPS_023, *Staphylococcus aureus* SPS_462, *Streptococcus mutans* SPS_474) was harvested from cultures grown on MSPS_029, at 37 °C, under 10% $H_2$, 10% $CO_2$, 80% $N_2$ atmospheric conditions, for 70 h. The material from the individual plates was divided into 8 replicate samples. For the bioprint set, bioprinted cultures on MSPS_029 were first cultured for 70 h, then single colonies were expanded on MSPS_029 under the same condition as above for 70 h, and divided into 8 replicate samples. Samples were transferred to 100 µL 4% paraformaldehyde (w/v in phosphate buffered saline, PFA). Bacterial specimens were fixed during 24 h incubation in PFA-solution at 4 °C, before use for spectroscopic analysis (i.e., FTIR measurements). Bioprinted bacterial material from the above mentioned strains, for use as test data sets, was prepared twice (i.e., two independent experiments). 'Bioprint set-2' and 'set-3' included, respectively, 10 and 9 monocultures, with two replicates for each of the bacterial strains indicated above, except for *S. anginosus* with only one replicate in 'Bioprint set-3'.

Infrared (IR) spectra were recorded using a Nicolet iS5 FTIR spectrometer (Thermo-Fisher) equipped with a triglycine sulfate detector and an attenuated total reflection (ATR) accessory with a pressure arm, as well as a diamond/ZnSe crystal. IR spectra were collected from bacterial cell pellets obtained via centrifugation (2 min at 1150 × $g$) after PFA fixation. After transfer onto the ATR crystal, bacterial samples were left to dry for 3–5 min, until complete evaporation of the liquid. Spectra were collected in the 4000−525 cm$^{-1}$ spectral range, at 4 cm$^{-1}$ optical resolution, while recording 6 co-added interferograms. The pressure arm was used for pressing the bacterial sample onto the ATR crystal. Prior to each measurement, a background spectrum was recorded after cleaning the ZnSe crystal with 70% (v/v) ethanol.

### Spectral data analyses of FT-IR spectra

After discarding spectra of poor quality, a total number of 331 spectra was obtained for further analysis. For spectral analysis, two different spectral ranges were selected, namely the fingerprint region (1800−900 cm$^{-1}$) and the CH-stretching region (3000−2800 cm$^{-1}$). The CH-stretching region includes $CH_3$- and $CH_2$-stretching vibration bands, originating from lipids present in the bacterial cell membrane. In the 1640−1560 cm$^{-1}$ region, the protein amide-I and amide-II absorbance bands can be found originating from C=O and N-H molecular group vibrations[156]. Moreover, the shape of the protein amide-I and II bands give information on the overall protein structure.

For use with multivariate data analysis, original spectra were subjected to baseline correction and vector normalization. This was done to reduce non-relevant variance and scale spectra in a uniform way. Spectral pre-processing includes baseline correction (i) and vector normalization (ii) as described in the following equations:

i) Baseline correction: assuming $a_k$ is the vector of the analytical spectrum and $w_k$ is the corresponding wavenumber, both have a length n. The slope and offset of a specific spectrum then is used for baseline correction, while the baseline corrected spectrum is expressed as $a'_k$:

$$slope = \frac{a_n - a_1}{w_n - w_1} \quad (1)$$

$$offset = a_1 - \frac{a_n - a_1}{w_n - w_1} \times w_1 \quad (2)$$

$$a'_k = a_k - (w_k \times slope + offset), k = 1, 2, 3 \ldots, n \quad (3)$$

ii) Vector normalization: The vector normalized spectrum can be presented as $a'''_k$, with the sum of the first n feature vectors equalling 1:

$$a''_k = a'_k - \frac{\sum_{k=1}^{n} a'_k}{n}, k = 1, 2, 3 \ldots, n \quad (4)$$

$$a'''_k = \frac{a''_k}{\sqrt{\sum_{k=1}^{n} a''^2_k}}, k = 1, 2, 3 \ldots, n \quad (5)$$

$$\sum_{k=1}^{n} a'''^2_k = 1, k = 1, 2, 3 \ldots, n \quad (6)$$

Bacterial spectral properties were compared both in the 3000–2800 cm$^{-1}$ and 1800–900 cm$^{-1}$ spectral regions (i.e., after baseline correction and vector normalization). All analyses were performed using the statistical programming environment R (version 4.0.4) (R Core Team 2020).

Principal Component Analysis (PCA), Linear Discriminant Analysis (LDA), Artificial Neural Network (ANN) and Random Forest (RF) models were applied for model training and evaluation.

Covariance-based PCA is an unsupervised learning method, which can reduce the multidimensional data into several linearly uncorrelated variables named principal components. The aim of PCA is to obtain an orthonormal matrix in which most of the data information can be maintained. Therefore, the projected data points are expected to be as scattered as possible and this dispersion can be mathematically expressed in terms of variance. The first principal component (PC1) contains most of the original information. In addition, in case of PCA of different spectra, the corresponding loadings plots reveal the wavenumber ranges underlying differentiation between clusters within the dataset (i.e., groups with similar spectral features). PCA was carried out using the R-package 'factoextra'.

Linear Discriminant Analysis (LDA), also known as Fisher LDA, is a supervised learning dimension reduction technique/approach. LDA is typically used to project data on a low dimension, in such a manner that projection points of the same category are as close to each other as possible while at the same time the centers of the data/projection points belonging to specific categories are as far from each other as possible. In case of use with spectral analysis, the variables created through LDA (i.e., LDA factors) are linear combinations of the

absorbance intensity values for different wavelengths. To increase the robustness, here a 5-fold cross-validation was implemented. Therefore, vector-normalized spectra originating from different bacterial species were divided randomLy into 5 groups, for validation per group without repeating, while using the remaining four dataset-groups as training data. This procedure was repeated until data from all groups were validated. Vector-normalized spectra in both the 1800–900 and 3000–2800 cm$^{-1}$ spectral regions were used for LDA. LDA model construction was done using the R-package 'MASS'. To assess the accuracy of classification according to the LDA model, performance metrics were derived for 'Bioprint-set2' and '-set3' as well as a merged dataset consisting of set-2 and set-3 combined. Confusion matrices and chord diagrams were prepared to visualize (mis)classification of the spectral data classification from different bacterial strains.

For data analysis by means of a supervised learning approach, an artificial neural network (ANN) model was constructed. The Adam algorithm was used for optimization of stochastic objective functions based on a first-order gradient. The theoretical convergence of the Adam algorithm was analyzed as described in two publications[157,158]. Namely, the interval of convergence rate was provided, and the convergence rate was proved optimal in the online convex optimization. Loss and accuracy functions were used to evaluate the ANN model and monitor the error during the training process. For ANN analysis, analysis was done for both the 1800–900 and 3000–2800 cm$^{-1}$ spectral regions. ANN analysis was done using the R-package 'keras'[159] on an octa-core CPU laptop equipped with a single NVIDIA GeForce GTX 1650 graphics card. The open-source machine learning framework 'TensorFlow' was employed for ANN, as well as for algorithm training and evaluation. Leave-one-out cross-validation (LOOCV) was applied to evaluate the performance of the train dataset. The final output are probability values between 0 and 1. To improve the robustness of the model, the ANN analysis was repeated 1000 times, and probability values are the average values derived from all iterations.

A random forest (RF) model was constructed in a leave-one-out cross validation setting. This involved choosing each observation as internal validation, while using the remaining data as 'training' set. This process was repeated until all data were used for validation. RF is a supervised approach that makes use of a defined number of decision trees. When the classification is performed, new input samples are entered, and each decision tree in the forest is evaluated and classified. Therewith, each decision tree will get its own classification result, the RF model summarizes and evaluates all decision tree results, and the result with the most votes is the final prediction result. This is the basic flow of the 'Bagging' algorithm. During the construction of the RF model, two tuning parameters are used to optimize the performance, namely 'number of trees' and 'mtry', which represents the number of variables contained in each decision tree. The error was found to be at its minimum when 'mtry' was set to 7, and the error was stable when 'ntree' was set to 650. Choosing a larger number of trees did not affect the outcome of the analysis. RF classification analysis was done on vector-normalized spectra, for both the 1800–900 and 3000–2800 cm$^{-1}$ spectral regions (i.e., fingerprint and CH-region), using the R-package 'RandomForest'. The decision trees were built based on the 'Gini index', and a 'CART' classification decision tree was built.

**Phenotyping of microbial isolates**

Cellular morphology, motility and capacity for co-aggregation were assessed using phase contrast microscopy, Gram-stain examinations and co-aggregation assays. Aerotolerance testing was performed by comparing growth in anaerobic and aerobic (enriched with 10% $CO_2$) atmosphere. Catalase, indole and urease production was assessed using 3% hydrogen peroxide (8070.4, Carl Roth), Kovac's reagent (2950.1, Carl Roth) and the rapid urease test, also known as RUT[160],

respectively. Requirement for nitrate, hemin and NAD was studied using disks D43, D45, D44 and D51 (MAST Group). To detect growth in the presence of drugs or growth inhibitors, the following susceptibility test disks from Thermo Scientific were used: Oxoid An-Ident discsincluding Erythromycin, 60 µg, Rifampin, 15 µg, Colistin, 10 µg, Penicillin, 2 units, Kanamycin, 1000 µg and Vancomycin, 5 µg (R65006), Oxoid Amoxycillin Antimicrobial Susceptibility discs (11953032), Oxoid Amoxycillin/Clavulanic Acid Antimicrobial Susceptibility discs (11913812), Oxoid Azithromycin Antimicrobial Susceptibility discs (10660125), Oxoid™ Cefoxitin Antimicrobial Susceptibility discs (11944022), Oxoid Ciprofloxacin Antimicrobial Susceptibility discs (10351253), Oxoid Clindamycin Antimicrobial Susceptibility discs (11994012), Oxoid Doxycycline Antimicrobial Susceptibility discs (11943022), Oxoid Metronidazole Antimicrobial Susceptibility discs (11913972), Oxoid Minocycline Antimicrobial Susceptibility discs (11924002), Oxoid Moxifloxacin Antimicrobial Susceptibility discs (10431893), Oxoid Ofloxacin Antimicrobial Susceptibility discs (11912952), Oxoid Penicillin G Antimicrobial Susceptibility discs (12761760), Oxoid Spiramycin Antimicrobial Susceptibility Disks, 100 µg (10451323) and Oxoid Tetracycline Antimicrobial Susceptibility discs (11963872).

## Matrix-Assisted Laser Desorption Ionization Time-Of-Flight Mass Spectrometry (MALDI-TOF MS)

Species identity was determined using a MALDI Biotyper (Bruker Daltonics GmbH & Co. KG, Bremen, Germany) according to the manufacturer's instructions. In brief, overnight cultures of the isolates on MSPS_029 plates were prepared for analysis using the standard procedures and chemicals for the direct transfer preparation method described in the user manual. Hard-to-lyse microorganisms, including spore formers that could not be identified using the direct transfer method, were tested again using the direct transfer-formic acid preparation method as described in the user manual. This commercial method generates PDF reports that include classification scores ranging from 0 to 3. Scores of 2.00–3.00 indicate highly probable identification, scores of 1.70–1.99 suggest probable identification, and scores below 1.70 are considered unreliable for classification. Taxonomic assignments were based on the best match, with consensus between top-ranking results used to determine genus or species level, and this was confirmed across two technical replicates. The analysis was conducted using Server Version 4.1.100, applying the MALDI Biotyper MSP Identification Standard Method 1.1 and the MALDI Biotyper Preprocessing Standard Method 1.1.

## Ethics statement

The investigation took place at the Department of Prosthetic Dentistry and Biomedical Materials Science at Hannover Medical School in Germany and received ethical approval from the institutional review board (approval numbers 5544 and 9477). All participants provided written informed consent.

## Reporting summary

Further information on research design is available in the Nature Portfolio Reporting Summary linked to this article.

## Data availability

The raw 16S rRNA gene amplicon sequencing data is available in GenBank and the European Nucleotide Archive under Project ID PRJEB75613. The assembled *Colibacter massiliensis* genome is available under Accession: PRJNA1297070 [https://www.ncbi.nlm.nih.gov/bioproject/1297070]. Source data are available with this paper, as referenced in the figure legends. Source data are provided with this paper.

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

## Acknowledgements

M.S. and S.P.S. would like to acknowledge Deutsche For-schungsgemeinschaft (DFG, German Research Foundation) for financial support (SFB/TRR 298 SIIRI – Project-ID 426335750). This study was also funded by the DFG under Germany's Excellence Strategy - EXC 2155 - project number 390874280 (M.S.). L.K. and B.C. would like to acknowledge financial support from German Cluster of Excellence Ex62/2 Rebirth and funding by the European Union (ERC, Laser-Tissue-Perfude, 101054009). I.Y. would like to acknowledge financial support from the "Federal and State Program Promoting Female Professors" (Grant No. 01FP19068J) and the program "Female Professors for Lower Saxony" (Reference No. 22-76251-99 P4/20). We would also like to thank Diana Strauch, Rainer Schreeb, Marly Dalton and Janine Steincke for technical assistance, as well as Amruta Joshi and Wiebke Behrens for bioinformatics support.

## Author contributions

S.P.S and B.S. conceived the study and supervised the project. T.Q., L.K., R.M. and S.P.S. designed the experiments. J.G. collected clinical samples. W.W., S.K., B.C., M.S. and S.P.S. provided essential resources. T.Q., L.K., R.M., Y.T., A.L.S., L.D.P., D.L., S.K. and S.P.S performed the experiments and collected the data. T.Q., R.M., A.W., I.Y., D.L. and SPS conducted data analysis. T.Q., R.M. and S.P.S. wrote the manuscript with comments from all authors.

## Funding

## Competing interests

The authors declare no competing interests.
