## [Transparent Peer Review file · Nature Communications]

Laser-assisted microbial culturomics

Corresponding Author: Dr Szymon Szafranski

Version 0:

Reviewer comments:

Reviewer #1

(Remarks to the Author)

In this manuscript, the authors introduce an innovative approach that leverages laser-assisted bioprinting to culture and characterize diverse microbes from both synthetic and clinical samples. This approach allows for high-throughput, precise deposition of microbes under various culture conditions to create comprehensive culture collections. The integration of multiple analytical techniques such as microscopy, spectroscopy, sequencing, and functional profiling (enzyme assays) enhances the versatility and depth of the microbial analysis. The strengths of this work include the pioneering use of high-precision bioprinting for culturomics, abilities to recover fastidious taxa via flexible media and mixed-species co-printing, as well as the multi-modal analyses. A notable limitation is the need for manual passaging of selected strains, but this could be addressed in future studies by incorporating automation strategies (e.g., robotics). In addition, the assays used to assess the physiology of the isolates or co-cultures are simplistic which provide limited information about their spatial organization or growth as biofilms. Nevertheless, the major goal is to demonstrate the capability and feasibility of using laser-assisted bioprinting to culture a highly challenging, diverse and heterogenous human oral microbiota, which is successfully accomplished in this study. The manuscript is well-written and introduces a conceptually novel approach that will likely advance our understanding of oral microbiome and polymicrobial interactions in health and disease. I have minor comments/questions to clarify some aspects of the methodology and its broader application:

1. While the bioprinting technique is high-throughput and precise, the subsequent steps of colony isolation and verification appear labor intensive, especially for complex communities such as dental plaque. Could the authors provide more context on how mixed colonies were selected for passing and how passing was performed?
2. Similarly, it's unclear whether the microscope-assisted colony picking (Fig. 4a) was performed manually. Clarifying this could help assess the feasibility of the method for large-scale culturomics studies.
3. The statement "high recovery of species was attributed to microbial syntrophy" lacks experimental evidence and is primarily based on assumptions regarding media composition and a simplistic in vitro assay. Either tone it down or provide further experimental data to support it.
4. The term "spatial taxonomic characterization" can be misleading as it suggests a natural spatial arrangement, whereas in the experiment the different species were "artificially" bioprinted to form colonies with different positioning or mixing. Moreover, colonies growing on solid media are not considered biofilms per se in the context of oral microbial communities. The use of microscopy to help assess the colony morphology and to detect specific taxa is certainly helpful for taxonomic characterization. Thus, it would be more accurate to describe this as "microscopy-assisted taxonomic characterization" rather than using "spatial".
5. The method described ("gentle but thorough mixing by pipetting") may not be effective in disrupting the highly structured dental biofilms, like those embedded in an adhesive EPS-rich matrix. Would a more robust dispersing method, such as sonication, be more effective in reducing the occurrence of mixed colonies and subsequent workload for culture purification?
6. There is insufficient description of clinical sampling of dental plaque, which is critical given the site-specificity and heterogeneity of the oral microbiome. Was the plaque pooled from different dental surfaces or collected at specific sites (tooth surfaces)? Supragingival or subgingival? How many samples were collected per subject?
7. A few typos noted: spelling of "phase-contrast microscopy" in Fig 4a; In Supplementary Information, line 4: change "form" to "from".

Reviewer #2

(Remarks to the Author)

The authors present a potential tool to approach one of the most difficult research challenges in biology at-large. Understanding the mechanisms at play, and thereby the functions, of specific compositions and communities of microbiota requires precise knowledge of not only the species present, but their specific metabolic processes and their subsequent influence on their microenvironment. Achieving this understanding from conventional microbial cultivation and analysis techniques is a standing challenge due, in a general sense, to the limitations of quantifying and observing complex, multi-species interactions with approaches that have been honed for decades to assess single species in isolation.

It is my opinion that culturomics will become a critical tool in assessing the impossibly numerous combinations of not only microbes, but host phenotypes and their combinatorial effects on their environment, in this case, the healthy vs diseased status of the oral microbiome. The authors understand the sheer scale of this challenge and have leveraged the unique properties of laser-induced forward transfer (LIFT) bioprinting, namely its speed and precision, to establish extensive microarrays of microbial cultures in a high-throughput manner. While utilizing LIFT bioprinting to pattern microbial cells is not in itself novel, this approach of utilizing patient sourced biomaterial as a printing stock to produce a wide range of microcolonies of varied composition and is potentially unique and powerful.

I do have several important questions that should be addressed within the body of the manuscript text:

1) With the extreme proximity of the printed microcolonies of microbes it seems that their subsequent culture in either liquid or hydrogel media would readily result in cross-talk between these microcolonies due to diffusion of metabolites and other biomolecules. The effect of diffusion from one colony will likely affect the growth and phenotypes of its nearest neighbors, which could lead to a further cascading affect across the entire microarray. Are there any steps taken to prevent microcolonies from influencing one another, or any quality control steps to ensure that if diffusive cross-talk is occurring that these colonies are excluded or evaluated differently than others? This is especially relevant in liquid cultures which result in "escapee" cells which depart from the hydrogel via various methods such as cell division or motility are able to physically transport themselves to other regions of the microarrays.

2) Multiple bioink materials and subsequent hydrogel matrices are used here to transfer bacteria to various acceptor surfaces for subsequent growth and assessment. However, have the authors evaluated or considered the potential of microbial metabolism of the encapsulation material itself, and how this may affect subsequent culturing and assumption of metabolic behaviors due to their consumption of the material? E.g. liberation of additional carbon or nitrogen compounds from the polymers. I can see at least one method that would likely work well already outlined in the manuscript, however I see no mention of its use to act as a quality control for microbes which metabolize the encapsulation materials.

3) I see many mentions of the bioinks being gelled, however I see no mentions of reagents or approaches to accomplish this. Are these microdroplets truly crosslinked? If so, I would like to see these listed in the materials and methods as it is a critical step in the fabrication process flow of these arrays and introduce additional materials which may influence microbial behavior (e.g. CaCl₂ for alginate gelation).

Minor Edits:

- Figure S1(a) says functionalized glass slides were used, what kind of functionalization? (d) scale bar is unlabeled in
- I don't see an explanation for how these are confirmed as "biofilm arrays", are biofilm polymers being identified or quantified? Or are these "biofilms" because they are biofilm-associated microbes encapsulated in an exogenous hydrogel? Is it because some bioink stocks contained disrupted biofilm fragments? Please include a sentence explaining this rationale.

If the above questions and concerns can be addressed, I would suggest this manuscript is suitable for publication in Nature Communications as it represents a meaningful path forward to begin parsing the incredibly complex nature of many microbiomes.

Reviewer #3

(Remarks to the Author)

Reviewer #4

(Remarks to the Author)

The authors present an interesting, elegant and well developed laser-based approach that allows to generate a large number of small microbial colonies on solid media or filter materials. They have gathered a large dataset that is presented in a considerable number of complex figures. The description of the laser-assisted printing process is well comprehensible for the reader, well documented, robust, and hence convincing. While laser-based techniques for the separation of single microbial cells (optical tweezers) have already been introduced more than two decades ago (which should be mentioned in the introduction), the production of entire arrays is the main asset of the high throughput method described here. Another asset is that the printing process can also be readily applied to biofilms from which bacteria often are difficult to cultivate using other techniques. On the downside, it has to be kept in mind that this technology will not be applicable for motile (by flagella or gliding) microorganisms. If motile bacteria constitute a significant fraction of a microbial community (like in marine or lacustrine environments), their growth will result in overgrowth and cross-contamination of entire print arrays. The technique therefore will be of particular interest for the generation of cultures from communities consisting of largely immotile

microorganisms but not applicable to many other environmental habitats (L. 304).

However, beyond the high throughput of the laser-assisted cell separation technique, the description is not structured very well and hence the thrust of this paper does not become clear.

- The concept to separate single cells and grow them separately on different media is not novel (see e.g. microfluidic approaches, and many others). What is novel is the very technique employed to separate the cells. If this is intended to be a methodological paper, it would be important for a better assessment of the advantages of this technique to (1) prove more clearly whether and at which number of printed colonies the rarefaction curves of the diversity of isolates reach saturation (there is only some indirect evidence for this as “reproducibility” of the hierarchical clustering results for similarity analysis of the taxonomic composition of the cultured strains is mentioned in L.150, Fig.3e), (2) quantitatively analyse through a Venn diagram or better an UpSet plot the extent to which different media overlap by yielding the same types of microorganisms, and (3) quantify the actual fraction of sequence types (rather than species) missed by the current approach and their taxonomic affiliation.

- The notion that a high-throughput of cultivation in different media yields many novel types of bacteria that can be grown is also not new. Typically even a combination of media remains selective, as also reported in this paper (LL 144-146). If the main thrust of this paper would be the recovery of key, not-yet-cultivated isolates from dental plaque, it would therefore be important (1) to determine the optimal combination of media for the growth of not-yet-cultivated microorganisms (what would be the minimum number of media types) e.g., through an UpSet plot, (2) to establish an approach that would allow to rapidly, at a high-throughput matching the laser-assisted printing, and reliably identify strains of taxa for which isolates are currently missing, or of strains of established taxa with distinctive, unknown properties (L. 250, how do the authors know that their isolates are fastidious, what is their definition), (3) to discuss why certain microbial groups are missed (possibly fastidious anaerobes, fragile cells – filamentous bacteria etc). The authors did not apply an anoxic atmosphere during bioprinting (L. 297) and it is to be expected that they missed fastidious anaerobes as a consequence. I would therefore refrain from sweeping generalizations such as in LL. 260ff - the last three decades have seen major improvements of cultivation methods (also high throughput), have yielded numerous really exciting and relevant isolates (published also in very high ranking journals).

- Just incorporating all isolates, instead of a selection, into a culture collection would not be meaningful, practicable and efficient. Most of the strains are largely uncharacterized at the time of isolation and hence require substantial additional experimental work, yet they often will be similar to the well-characterized strains in public collections and hence not of further interest. Regarding the concept of a complete collection of all isolated cultures as promoted in the current paper, it also has to be noted that institutional collections are often discontinued due to ever increasing costs and lack of sustainable funding. Contrary to the generalizing statement (L. 23), there is not dearth of robust culture collections (there are 859 registered culture collection in the World Data Center for Microorganisms). The authors may have intended to say that few isolates from the human microbiome are currently available, but such a statement would not be appropriate in general. Notably, the list of the isolates from oral microbiome they were able to obtain (see LL. 59ff) is rather long.

What would be needed, are additional analyses that provide deeper, quantitative insights into the above questions so that the implications of the data become clear. Aside from the laser-assisted bioprinting itself, the methodological descriptions in some parts are incomplete or sometimes even remain unclear. Many of the rather generic statements and exemplary descriptions (see below) would not be needed.

One important indicator for the superiority of a novel technique for cultivation is the recovery of numerous phylogenetically novel microorganisms, particularly of higher taxa like families, orders or particularly classes. The authors retrieved 249 isolates from 14 classes and 124 species. In par. LL. 144-150 they provide only general, and partly only indirect, evidence for the success their approach. For instance, the title of the chapter L. 135f claims that nearly half of the original species richness was recovered. For this, good estimators of the number of species present (which is not the species richness per se) are required. The authors neither provide rarefaction curves (which at a sufficiently high number of replicas should be saturated and hence provide a reliable estimate), Chao non-parametric diversity estimators, or similar measures. As a note on the side, the authors may want to consider using state-of-the-art Hill coefficients rather than the Shannon or Simpson diversity and Evenness indices. Which of the existing microbial lineages are systematically missed by which medium and in combination? Hierarchical clustering based on Bray-Curtis similarity or NMDS do not explicitly provide this information. Fig. 4b demonstrates that typically not-yet-cultivated bacteria like the Saccharibacteria (should read ‘Candidatus Saccharibacteria’, Abscondibacteria (use quotation mark, has no standing in nomenclature) or Gracillibacteria (use quotation mark, has no standing in nomenclature) were also missed by the present approach. Fig. 5 is particularly relevant for the understanding of these results. At first sight it would seem that most isolates depicted in Fig. 5e are unnamed phylotypes, yet most of them were neither in bioprints nor purified (hence not cultivated at all). For some lineages (e.g. Oscillatoriaphycidae) the entire information is missing – how did they get into this graph then? These results have to be reported much more clearly.

A key issue is also the taxonomic level at which comparative work is done. Based on the reasoning of the authors that full 16S rRNA gene sequences provide a higher resolution of phylogenetic differences, I would have expected that all comparisons between microorganisms cultured and originally present would have been done on the sequence variant level. Yet, the authors remained exclusively on the species level in their analyses. They state that overall 65% of the species were recovered. It can be debated, whether this is to be interpreted as “closely resembling the clinical situation” (L. 160), particularly since on the sequence variant level (applied by other studies), this value is expected to be even lower. Furthermore, the description of unnamed oral taxa and OTUs (LL. 165 – 169) remains enigmatic. It is therefore unclear, to which extent the “culturomics based on laser bioprinting seems...appropriate...for the exploration of oral microbial dark matter” (LL. 168-169). For similar reasons, the implications of the isolates obtained cannot be fully evaluated as the authors report on their in situ transcriptional activity only on the genus level (L. 34). As another complication, the authors seem to have identified part of their isolates just by MALDI-TOF (L. 233) but the fraction of these isolates and how they are represented in the figures remains unclear. I also noted that at least *S. mutans* and *S. anginosus* cannot be readily

distinguished by FT-IR (Fig. 5c, d).

Fig. 2 could be central for the understanding of the Ms. The three legends on the top and bottom left could be stacked on top of each other for easier reading. Sectoral lines should be more distinct. The panel on the bottom right is designed like a legend but they correspond to the circle depicted for bacteria and archaea, and in analogy, should be shown as a sector of a circle. In addition, the abbreviations for genera are non-standard. In Fig. 3a and d, relative abundances can almost not be determined – a different way of presentation should be chosen or the panels left out.

While the first oral strain of *Colibacter massiliensis* was isolated by the authors (L. 253), a first strain of this species (Marseille-P291 1) existed before. Knowing possible differences in the characteristics of the two hence would be of particular interest. At the same time, the authors isolated a representative of *Arachnia rubra* which – contrary to the statement in their manuscript - is not the first oral isolate (Sk-1 was isolated from gingival sulcus samples in 2018).

I commend the authors for implementing an approach that enables the exploitation of interactions using laser imprinting. Yet, mentioning just one example (*P. pasteri*, LL. 172-176) does not allow the reader to assess, whether “High recovery of species was attributed to microbial syntrophy” (L. 171f). A quantitative assessment is needed here. In addition, the relevance of the FISH-approach (LL. 205ff) does not become clear in this respect. Were more colonies tested in this respect (than the one shown) to detect co-cultures? That might yield very important information. The statement in L. 477 of the SI is not valid when formulated in such a generic way. Dependence on helpers critically depends on cultivation conditions. Often the dependence is not given, if growth factors, specific growth substrates or dynamics of substrate (e.g., removal of hydrogen of syntrophic anaerobes) can be provided during incubations.

The current design of the figures and tables make it often hard or even impossible to find the evidence for the general conclusions and statements that the authors make in the text. This renders it even more difficult for the unspecialized reader to follow the reasoning of the paper. Large parts of the manuscript proper and some parts of the figures are far too generic to understand the approach and the conclusions. The authors present and visualize numerous exemplary data (e.g. Fig. S4c) or provide non-specific, basic information that should be eliminated (Fig. 1a, Fig. 4d, Fig. S5, Fig. S6a,d, Fig. S7a,f). In particular, LL. 312-321 are not at all instructive for understanding the methods. LL. 177-185 are all very basic and standard and not needed here. Instead, the experimental strategy should be given here. As another example, LL. 507-510 in the SI are basic textbook descriptions and hence superfluous. The implications of Fig. 1e remain unclear; what does it demonstrate, except that printing of a portrait is possible? What are the colors? The statement that microorganisms can be identified by colony morphology is a sweeping generalization (L.186) that needs to be qualified: only under very selective growth conditions, this might be true in some cases. And yet, there are many examples where novel types of bacteria did not follow the “presumptive identification” criteria, yielding false positives or negatives. Also, Fig. 4a does not provide any proof for this statement. The par. LL. 210-225 reads as if novel properties of well established pathogenic species had been detected, yet at least several were well known before. The authors need to more clearly distinguish novel findings from existing knowledge.

Throughout the text, SI is only referred to as unspecifically and the reader is expected to search through the 32 pages of the SI text in trying to find the description or evidence the authors refer to. The authors claim that “bacteriophage-bacterium relationships were reproduced” (L. 118) but I could not find evidence for this statement, also after consulting the SI. Several sections of the Discussion reiterate the results. It would be preferable to discuss the implications of the findings in more detail. For better comparison, I would have also expected that the following ref. was considered, at least in the discussion section:

Khelaifia S, Virginie P, Belkacemi S, Tassery H, Terrer E, Aboudharam G. Culturing the Human Oral Microbiota, Updating Methodologies and Cultivation Techniques. *Microorganisms*. 2023 Mar 24;11(4):836. doi: 10.3390/microorganisms11040836.

Specific comments

L. 29 16S rRNA gene profiling

L. 38 be more precise: do you mean diverse biofilm-associated species or something else? Otherwise the meaning of the sentence is not understood.

L. 40 Microbial strains cannot be personalized, a treatment is

L. 42 There is no need for culture collections (see above) but there is a need for particular isolates that are so far underrepresented or missing in existing culture collections.

L. 88 16S rRNA gene profiling

L. 329 it would help if the authors had information to which extent the droplets containing cells are heated during the laser pulse. This heating has at least been an issue with laser tweezers.

L. 125 highly diverse cell envelope structure

L. 143 “input to output cell counts ratio” unclear and not described in SI

L. 147 45% of the original richness

LL. 196ff Statement unclear

L. 252f first oral strains....which are..

L. 421 Reference is incomplete

L.466 In the outermost ring, the media...

L.482 Why does medium 29 appear twice? This is not clear here.

L. 487 ..by class (for those encompassing...

Supplementary Material

L. 39 How were the research groups / strains selected?

L. 57 To which source does the identification number of cultivation media refer to?

L. 59 and throughout The names of the higher taxa (e.g., Actinomycetia) should be given in italics. Also, type strains should be designated with a superscript “T” following taxonomy conventions.

L. 118, L. 153 “Note taxonomy update (Hitch et al., 2022).” “Taxonomy updated following (Nouioui et al., 2018).” – what is

this supposed to mean?

L. 154, L. 203 which helper strains were used?

L. 285 "The concept of the stamp was adapted for prints" – and how was this achieved?

L. 288 Whatman

L. 292 depending

L. 316 retrieved

L. 436 Heparinase is a lyase (and not a DNase) with eliminative cleavage of polysaccharides containing (1→4)-linked D-glucuronate or L-iduronate residues and (1→4)- α -linked 2-sulfoamino-2-deoxy-6-sulfo-D-glucose residues to give oligosaccharides with terminal 4-deoxy- α -D-gluc-4-enuronosyl groups at their non-reducing ends. How then was this enzyme detected?

L. 444, L. 449 either explain principle of test reaction or provide reference

L. 467 either explain principle of test reaction or provide reference

L. 490, 491 use strains (plural, four times)

L. 497 A 0.45 μ m pore size may allow cells with smaller diameters to cross the membrane. This cannot be excluded by this approach.

L. 514 either all in capital letters or not

L. 533 I did not find any Table 10

L. 552 16S rRNA gene-targeting

L. 569 primers

L. 573 (reference) missing

L. 580 microbial community composition

L. 581 PacBio

L. 585 The following parameters

L. 585ff put all information past "were performed" in parenthesis

L. 603 identified to the species level

L. 605 delete dash in front of "representative"

L. 609 Were tests with negative controls (no DNA template) performed to check for sequences produced from reagents alone? To our experience, this can make substantial contributions to the sequence dataset.

L. 621 nucleatum

L. 698 Cellular morphology, motility and

L. 701 hemin

L. 718 in the culture collection

L. 1069f give species names in italics

Reviewer #5

(Remarks to the Author)

The manuscript by Qu et al describes the development of a laser-assisted microbial culturomics system and the deployment of it for isolation of oral microbes. The authors describe the versatility of the system to print in different configurations including co-cultures, in plates, on glass slides, and across membranes. Then the system was used to isolate clinical biofilms from dental plaques, which showed the ability to recover most of the abundant species including some previously uncultured isolates. The authors further deployed the system in combination with microscopy, spectroscopy and enzyme assays to improve isolation performance including the development of algorithms that improved colony discrimination.

Overall, the technology described is interesting and will have general interest for the microbial cultivation field. The isolates generated are of interest to those studying oral microbiome. While there is a lot of possible data in this paper, the description of how they are generated and their organization is very poor. The paper is quite difficult to parse and much of the technical details are either not available or hard to find. This paper should be re-written and reorganized to improve overall clarity of what is done and how it was done.

Major concerns:

- 1) Details for different parts of the study are not well described. For example, Figure 1C shows syntrophic species interactions that can support fastidious microbes, but it's not clear how that figure was generated and the underlying data for it. The supplemental information only lists references, which is not very informative otherwise. Another example is Figure 6, how is this figure generated? What RNAseq was performed?
- 2) It is not clear how the isolation is performed following the printing. Are colonies picked by hand? If so, how many total colonies were picked and what level of redundancy was found based on the most optimal colony differentiation strategy (enzyme, spectra, nutrients)?
- 3) Genomic and taxonomic data associated with the isolation is missing. The specific strains isolated across the study is not listed anywhere. A supplemental table of all isolates generated, and associated genomic info (e.g. 16S) is needed at the minimum. The SI contains just a text dump of isolates which is not useful for most people. Are these strains available to the larger research community?
- 4) Figure 2 is really difficult to interpret. There are so many different things being shown. The isolation media are listed as numbers, but I am not able to find their correspondence anywhere. A table would have better summarized these results than the circular plot. Also, much of the text is bunched up and it is just overall impossible to understand all the different elements.
- 5) It is not clear from the laser printing methodology how one can obtain clonal isolates directly from the biofilm. A single droplet may contain dozens of different microbes. The manuscript is not very clear on how individual isolates are generated from the spots. This is true for Figure 3 for example.

6) The paper outlines different analysis steps that were done to study the biofilms including optical and biochemical. It is not clear how different modalities truly contributed to isolating new strains. The manuscript simply shows some examples of plates, but a quantitative analysis is needed.

7) The manuscript stated that they used LDA to distinguish different microbes versus NN or RF, but it seems that this is tested only on 5 type strains. Was this visualization modality really used for the isolation of the collection or just a proof of concept on a few different strains? This is also true for Fig 6(a-d)

8) Overall, the supplemental figures are fuzzy and low resolution so it is hard to read all the data on there.

Version 1:

Reviewer comments:

Reviewer #1

(Remarks to the Author)

The authors have made substantial revisions to the manuscript and have adequately addressed previous concerns.

Reviewer #2

(Remarks to the Author)

The authors have satisfactorily addressed my comments and concerns; I would now suggest that this manuscript is suitable for publication in Nature Communications.

Reviewer #3

(Remarks to the Author)

Reviewer #4

(Remarks to the Author)

Most missing information has been added to the manuscript and open questions have been clarified. However, I still find figures 1 to 5 very complex.

Minor comments:

L. 121: These species were consequently have been employed... ; delete "have been"

LL. 354-364: extremely long, unstructured sentence that is hard to read and needs to be split into several, separate statements.

L. 170 and also L. 197, L. 328: there is no class 'Saccharibacteria' described so far. Only the phylum 'Candidatus Saccharimonadota' (with Candidatus in italics and Saccharimonadota not) is currently recognized and hence should be used here instead of the wrong 'Saccharibacteria'.

Table S9: Leptotrichia needs to be in italics

Reviewer #5

(Remarks to the Author)

The revised manuscript is much improved. The reviewer has no further issues.

Version 2:

Reviewer comments:

Reviewer #4

(Remarks to the Author)

The previous concerns have now been satisfactorily addressed.

Dear Reviewers,

Thank you very much for evaluating our manuscript *Laser-assisted microbial culturomics* (NCOMMS-24-31352-T) and your valuable feedback. We have carefully addressed your comments and questions, including those related to amplicon sequence variant (ASV) analyses, rarefaction analyses, genome sequencing, and diversity assessments. The revised manuscript also includes three new supplementary figures and nine supplementary tables. Please find our detailed point-by-point responses below.

We are looking forward to hearing from you,

Sincerely,

Szymon Szafranski
on behalf of all co-authors

Reviewer #1 (Remarks to the Author):

In this manuscript, the authors introduce an innovative approach that leverages laser-assisted bioprinting to culture and characterize diverse microbes from both synthetic and clinical samples. This approach allows for high-throughput, precise deposition of microbes under various culture conditions to create comprehensive culture collections. The integration of multiple analytical techniques such as microscopy, spectroscopy, sequencing, and functional profiling (enzyme assays) enhances the versatility and depth of the microbial analysis. The strengths of this work include the pioneering use of high-precision bioprinting for culturomics, abilities to recover fastidious taxa via flexible media and mixed-species co-printing, as well as the multi-modal analyses. A notable limitation is the need for manual passaging of selected strains, but this could be addressed in future studies by incorporating automation strategies (e.g., robotics). In addition, the assays used to assess the physiology of the isolates or co-cultures are simplistic which provide limited information about their spatial organization or growth as biofilms. Nevertheless, the major goal is to demonstrate the capability and feasibility of using laser-assisted bioprinting to culture a highly challenging, diverse and heterogenous human oral microbiota, which is successfully accomplished in this study. The manuscript is well-written and introduces a conceptually novel approach that will likely advance our understanding of oral microbiome and polymicrobial interactions in health and disease. I have minor comments/questions to clarify some aspects of the methodology and its broader application:

Author's Response: We have included an outlook on spatial activity and organization analyses (page 14, line 338, page 15, lines 341-343) and added few new references (Dar *et al.*, 2021; Sarfatis *et al.*, 2025). Our discussion addressed the other limitations and potential solutions highlighted by the reviewer (pages 14-15, lines 337-341).

1. While the bioprinting technique is high-throughput and precise, the subsequent steps of colony isolation and verification appear labor intensive, especially for complex communities such as dental plaque. Could the authors provide more context on how mixed colonies were selected for passing and how passing was performed?

Author's Response: Mixed colonies were primarily selected based on morphology, captured via macro-mode photography (full bioprint array or agar plate), phase contrast microscopy (individual bioprints or small plate areas), or both. Occasionally, confocal laser scanning microscopy was used to detect the induced fluorescence of specific anaerobes in mixed colonies. Transparent media suited phase contrast microscopy, while opaque media required camera imaging and confocal laser scanning microscopy. Screening began with primary macro-mode-based selection, followed by targeted or systematic microscopy. Colonies were manually classified into morphotypes using images processed with Adobe Illustrator (v. CS6). Specific or representative colonies were manually passaged under phase contrast microscope onto agar plates MSPS_029, or occasionally other media. Due to limited precision/resolution of our manual approach, multiple strains were transferred together in some cases. Resulting colonies were photographed and evaluated. Colonies representing individual morphotypes were purified by multiple passaging, and taxonomy was generally assigned by 16S rRNA gene amplicons sequencing. Recurrent mixed colonies, resulting from multiple passages, indicated polymorphism. The respective Supplementary Information section was updated accordingly (page 10, line 224 and SI, page 20, lines 638–640). We have also revised the legend of Fig. 4 for improved clarity (page 26, lines 581-582).

Similarly, it's unclear whether the microscope-assisted colony picking (Fig. 4a) was performed manually. Clarifying this could help assess the feasibility of the method for large-scale culturomics studies.

Author's Response: Yes, we performed manual transfer. We have revised the manuscript (page 10, line 224-225) and legend of Fig. 4 for improved clarity (page 26, line 581).

2. The statement “high recovery of species was attributed to microbial syntrophy” lacks experimental evidence and is
primarily based on assumptions regarding media composition and a simplistic *in vitro* assay. Either tone it down or
provide further experimental data to support it.

**Author's Response:** We used more tentative language (pages 8-9, lines 185-192). A closer investigation of this phenomenon
led to our follow-up study with the draft title "*Metatranscriptomics-Guided Discovery of Naphthoquinone Interactions in*
*Implant-Associated Biofilms Between Taxonomically Diverse Suppliers and Bacteroidia Recipients*" which is currently in
preparation.

3. The term “spatial taxonomic characterization” can be misleading as it suggests a natural spatial arrangement, whereas
in the experiment the different species were “artificially” bioprinted to form colonies with different positioning or
mixing. Moreover, colonies growing on solid media are not considered biofilms *per se* in the context of oral microbial
communities. The use of microscopy to help assess the colony morphology and to detect specific taxa is certainly
helpful for taxonomic characterization. Thus, it would be more accurate to describe this as “microscopy-assisted
taxonomic characterization” rather than using “spatial”.

**Author's Response:** We have revised the title of the results section accordingly (page 9 line 204). The term “spatial” has been
retained now only in the context of the Fluorescence *in situ* Hybridization combined with Confocal Laser Scanning Microscopy
(FISH-CLSM) analysis, which served as a proof-of-principle method for spatial localization (page 10, lines 231-236). The
potential application of bioprinting for studying natural spatial arrangements, particularly its ability to capture interspecies
physical interactions (see Fig. 1d, top left panel), was introduced in the discussion (page 15, lines 341-343). Bioprint-based
retrieval and expansion of physically bound cells from different species can be validated using co-aggregation assays as well
as single-cell force spectroscopy combined with fluidic force microscopy. We are actively exploring this research direction.

The method described (“gentle but thorough mixing by pipetting”) may not be effective in disrupting the highly structured
dental biofilms, like those embedded in an adhesive EPS-rich matrix. Would a more robust dispersing method, such as
sonication, be more effective in reducing the occurrence of mixed colonies and subsequent workload for culture purification?

**Author's Response:** More robust dispersion methods, such as sonication or moderate bead beating, could improve biofilm
partitioning but may also damage mechanically sensitive members like *Treponema* spp.. Alternatively or additionally, chemical
or enzymatic treatment could be considered. The discussion has been updated accordingly (page 14, lines 333-335) and a
reference 57 was added (Wang *et al.*, 2023).

There is insufficient description of clinical sampling of dental plaque, which is critical given the site-specificity and
heterogeneity of the oral microbiome. Was the plaque pooled from different dental surfaces or collected at specific sites (tooth
surfaces)? Supragingival or subgingival? How many samples were collected per subject?

**Author's Response:** The description of clinical sampling has been updated in Supplementary Information (page 12, lines 360-
367) and sample characteristics were provided in the form of a table (Tab S10).

4. A few typos noted: spelling of "phase-contrast microscopy" in Fig 4a; In Supplementary Information, line 4: change
"form" to "from".

**Author's Response:** Typos were corrected. Documents were proofread again.

**Reviewer #2 (Remarks to the Author):**

The authors present a potential tool to approach one of the most difficult research challenges in biology at-large. Understanding
the mechanisms at play, and thereby the functions, of specific compositions and communities of microbiota requires precise
knowledge of not only the species present, but their specific metabolic processes and their subsequent influence on their
microenvironment. Achieving this understanding from conventional microbial cultivation and analysis techniques is a standing
challenge due, in a general sense, to the limitations of quantifying and observing complex, multi-species interactions with
approaches that have been honed for decades to assess single species in isolation.

It is my opinion that culturomics will become a critical tool in assessing the impossibly numerous combinations of not only
microbes, but host phenotypes and their combinatorial effects on their environment, in this case, the healthy vs diseased status
of the oral microbiome. The authors understand the sheer scale of this challenge and have leveraged the unique properties of
laser-induced forward transfer (LIFT) bioprinting, namely its speed and precision, to establish extensive microarrays of
microbial cultures in a high-throughput manner. While utilizing LIFT bioprinting to pattern microbial cells is not in itself novel,
this approach of utilizing patient sourced biomaterial as a printing stock to produce a wide range of microcolonies of varied
composition and is potentially unique and powerful.

I do have several important questions that should be addressed within the body of the manuscript text:

1) With the extreme proximity of the printed microcolonies of microbes it seems that their subsequent culture in either liquid
or hydrogel media would readily result in cross-talk between these microcolonies due to diffusion of metabolites and other
biomolecules. The effect of diffusion from one colony will likely affect the growth and phenotypes of its nearest neighbors,
which could lead to a further cascading affect across the entire microarray. Are there any steps taken to prevent microcolonies
from influencing one another, or any quality control steps to ensure that if diffusive cross-talk is occurring that these colonies
are excluded or evaluated differently than others? This is especially relevant in liquid cultures which result in “escapee” cells
which depart from the hydrogel via various methods such as cell division or motility are able to physically transport themselves
to other regions of the microarrays.

**Author's Response:** To mitigate the effects of cross-talk due to diffusion and motility, we developed a printing approach onto
agar in 96-well plates (Fig. S1c). This setup controls diffusion and restricts motility, ensuring that microbial colonies remain
spatially confined (page 6, lines 113-114). However, in settings where diffusion occurs, we found this phenomenon to be highly
beneficial. The controlled diffusion environment facilitates the growth of fastidious microorganisms that depend on the
metabolites from neighboring colonies. Additionally, it enables to study enzyme sharing (Fig. 4d). This led us to recognize the
value of a third, mixed configuration, where microorganisms are confined within semi-permeable compartments. These
compartments would restrict physical movement while allowing metabolic communication with surrounding colonies, such as
dedicated “helper” organisms. The discussion section has been updated accordingly (page 15, line 341-343).

2) Multiple bioink materials and subsequent hydrogel matrices are used here to transfer bacteria to various acceptor surfaces
for subsequent growth and assessment. However, have the authors evaluated or considered the potential of microbial
metabolism of the encapsulation material itself, and how this may affect subsequent culturing and assumption of metabolic
behaviors due to their consumption of the material? E.g. liberation of additional carbon or nitrogen compounds from the
polymers. I can see at least one method that would likely work well already outlined in the manuscript, however I see no
mention of its use to act as a quality control for microbes which metabolize the encapsulation materials.

**Author's Response:** In this phase of our work, our primary focus has been on developing a universal robust bioink that is
largely compatible with both biofilms and individual microbes, including bacteria and viruses. While the bioink performed
well overall, we did observe a toxic effect on *Treponema* species, highlighting the need for further development and refinement
(page 7, lines 137-140). In our microbiological experiments, we utilized complex, rich media, meaning that any potential
metabolic interactions with the bioink material would primarily be relevant only at the very initial stage of growth. Nonetheless,
we recognize the importance of evaluating microbial metabolism of bioink components, especially in chemically defined
environments, and have updated the discussion accordingly (page 14, lines 331–333). Moreover, the presented enzyme assays
for detecting microbial hydrolases can be adapted as a quality control measure to identify microorganisms that may interfere
with encapsulation (page 11, lines 251-253).

3) I see many mentions of the bioinks being gelled, however I see no mentions of reagents or approaches to accomplish this.
Are these microdroplets truly crosslinked? If so, I would like to see these listed in the materials and methods as it is a critical
step in the fabrication process flow of these arrays and introduce additional materials which may influence microbial behavior
(e.g. CaCl₂ for alginate gelation).

**Author's Response:** We tested and utilized crosslinkable fibrinogen-, collagen- and alginate-based bioinks to enable future
applications in 3D bioprinting of biofilms (Fig. S1). Therefore, we investigated the impact of crosslinking on bacterial viability
using *Staphylococcus aureus* as a model organism for the following gels: Fibrinogen was crosslinked post-printing using
thrombin, rat tail collagen type I was neutralized (inducing a delayed crosslinking) with sodium hydroxide, tenfold concentrated
phosphate-buffered saline, and Dulbecco's Modified Eagle Medium prior to printing, with the pH adjusted to 7.2–7.5, while
alginate gelation was induced with calcium chloride. However, crosslinking was not applied in the culturomics presented in
this study where the droplets were 2D printed. To avoid any misunderstanding, we have revised the text in several sections for
clarity, e.g., we changed “hydrogel” to “bioink”, (page 5, lines 105, 110; page 7, line 139, etc., and SI, page 9, lines 256-263
page 37, lines 1235, etc.).

Minor Edits:

• Figure S1(a) says functionalized glass slides were used, what kind of functionalization? (d) scale bar is unlabeled in

**Author's Response:** Glass slides were functionalized with hydrogels to prevent rapid dehydration of the micro-droplets, which
would otherwise occur quickly due to their small volume. The text in the Supplementary Information has been updated
accordingly (SI, page 37, lines 1234 and 1240).

• I don't see an explanation for how these are confirmed as “biofilm arrays”, are biofilm polymers being identified or quantified?
Or are these “biofilms” because they are biofilm-associated microbes encapsulated in an exogenous hydrogel? Is it because
some bioink stocks contained disrupted biofilm fragments? Please include a sentence explaining this rationale.

**Author's Response:** We used the term colony biofilms for structures grown on agar or filter membranes due to their surface
adherence, high cell density, spatial heterogeneity, and potential polymer production, consistent with usage by leading

researchers (Gloag *et al.*, 2018). However, given the absence of liquid medium and shear stress, we now use more tentative
language. We've replaced 'biofilm' with 'colony' when referring to agar-grown structures (page 4, lines 79, 82, 85; page 6, line
118, etc.) while retaining 'biofilm arrays' for printed arrays cultured under liquid conditions, which align with typical biofilm
characteristics.

If the above questions and concerns can be addressed, I would suggest this manuscript is suitable for publication in Nature
Communications as it represents a meaningful path forward to begin parsing the incredibly complex nature of many
microbiomes.

**Reviewer #3 (Remarks to the Author):**

I co-reviewed this manuscript with one of the reviewers who provided the listed reports. This is part of the Nature
Communications initiative to facilitate training in peer review and to provide appropriate recognition for Early Career
Researchers who co-review manuscripts.

**Reviewer #4 (Remarks to the Author):**

The authors present an interesting, elegant and well developed laser-based approach that allows to generate a large number of
small microbial colonies on solid media or filter materials. They have gathered a large dataset that is presented in a
considerable number of complex figures. The description of the laser-assisted printing process is well comprehensible for the
reader, well documented, robust, and hence convincing. While laser-based techniques for the separation of single microbial
cells (optical tweezers) have already been introduced more than two decades ago (which should be mentioned in the
introduction), the production of entire arrays is the main asset of the high throughput method described here.

**Author's Response:** We have updated the introduction to include information about optical tweezers (page 4, lines 62-63).

Another asset is that the printing process can also be readily applied to biofilms from which bacteria often are difficult to
cultivate using other techniques. On the downside, it has to be kept in mind that this technology will not be applicable for
motile (by flagella or gliding) microorganisms. If motile bacteria constitute a significant fraction of a microbial community
(like in marine or lacustrine environments), their growth will result in overgrowth and cross-contamination of entire print
arrays. The technique therefore will be of particular interest for the generation of cultures from communities consisting of
largely immotile microorganisms but not applicable to many other environmental habitats (L. 304).

**Author's Response:** We recognized this limitation (Fig. S3) and addressed it by developing a printing method on agar within
96-well plates (Fig. S1c). This setup spatially confines microbial microcolonies, effectively preventing the spread of motile
oral microorganisms (*e.g.*, *Campylobacter*, *Capnocytophaga*, *Selenomonas* spp.) across print arrays. The results and discussion
sections have been updated accordingly (page 6, lines 113–114; page 15, lines 341-343).

However, beyond the high throughput of the laser-assisted cell separation technique, the description is not structured very well
and hence the thrust of this paper does not become clear.

**Author's Response:** We have revised the manuscript to improve its structure and overall clarity. We also included multiple
supplementary tables that provide detailed results.

The concept to separate single cells and grow them separately on different media is not novel (see *e.g.* microfluidic approaches,
and many others). What is novel is the very technique employed to separate the cells. If this is intended to be a methodological
paper, it would be important for a better assessment of the advantages of this technique to (1) prove more clearly whether and
at which number of printed colonies the rarefaction curves of the diversity of isolates reach saturation (there is only some
indirect evidence for this as "reproducibility" of the hierarchical clustering results for similarity analysis of the taxonomic
composition of the cultured strains is mentioned in L.150, Fig.3e), (2) quantitatively analyse through a Venn diagram or better
an UpSet plot the extent to which different media overlap by yielding the same types of microorganisms, and (3) quantify the
actual fraction of sequence types (rather than species) missed by the current approach and their taxonomic affiliation.

**Author's Response:** Regarding point (1): we have now included a rarefaction analyses based on 16S rRNA gene amplicon
data at the genus, species, and Amplicon Sequence Variant (ASV) levels (see Fig. S6a –S6f). This analysis allowed us to
estimate the sampling effort required to recover a given number of taxa. While our analysis was not adjusted for confounding
factors such as variation in 16S rRNA gene copy number or colony size, data that were unavailable, we still consider this
approach to provide a reasonable approximation of expected diversity within strains passage by random picking from colony
arrays derived from dental plaque. The measure was most reliable for Medium MSPS_029, where the highest number of
observations ($n = 15$) was available. In addition, we present genus-level rarefaction curves for isolates (Fig S6g). This level of
resolution was chosen because species-level annotation was not consistently achievable for all isolates. A direct comparison

between predicted and experimental recovery was challenging, since our predictions were based on random colony selection,
 whereas our experiments involved careful selection of colonies based on their unique morphological features, allowing for a
 more targeted approach. Regarding point (2): we have provided data on taxonomic overlap across different media at the genus,
 species, and ASV levels, now available in the SI as Tab. S2 and Tab. S3. Regarding point (3): as mentioned above, we included
 ASV-level data in our analyses, which are also presented in Fig. S5 and S6, as well as Tab S3 and S4. Please refer to the
 response to the following point for a detailed description of the new ASV data (rebuttal lines 230 – 252).

- The notion that a high-throughput of cultivation in different media yields many novel types of bacteria that can be grown is
 also not new. Typically even a combination of media remains selective, as also reported in this paper (LL 144-146). If the main
 thrust of this paper would be the recovery of key, not-yet-cultivated isolates from dental plaque, it would therefore be important
 (1) to determine the optimal combination of media for the growth of not-yet-cultivated microorganisms (what would be the
 minimum number of media types) e.g., through an UpSet plot, (2) to establish an approach that would allow to rapidly, at a
 high-throughput matching the laser-assisted printing, and reliably identify strains of taxa for which isolates are currently
 missing, or of strains of established taxa with distinctive, unknown properties (L. 250, how do the authors know that their
 isolates are fastidious, what is their definition), (3) to discuss why certain microbial groups are missed (possibly fastidious
 anaerobes, fragile cells – filamentous bacteria etc). The authors did not apply an anoxic atmosphere during bioprinting (L. 297)
 and it is to be expected that they missed fastidious anaerobes as a consequence. I would therefore refrain from sweeping
 generalizations such as in LL. 260ff - the last three decades have seen major improvements of cultivation methods (also high
 throughput), have yielded numerous really exciting and relevant isolates (published also in very high ranking journals).

**Author's Response:** Regarding point (1): We found that UpSet plots were difficult to interpret due to their complexity. For
 example, even after omitting the most robust medium (MSPS_029) to focus on how other media might complement it, the top
 amplicon sequence variants (ASVs) dataset (cut-off = 0.1%) still produced 63 rows in the UpSet plot. Despite these
 visualization challenges, the underlying data proved highly useful for the iterative optimization of media selection. Our strategy
 was to identify media that contributed the highest number of new ASVs when added stepwise. To better visualize different
 aspects of the ASV data, we designed Fig. S4 and S5 and Tab S3 and S4 accordingly. Shade plots in Fig. S5a and S5b
 (counterparts to the species-level representations in Fig. 3a and 3d) display presence/absence data to enhance contrast,
 following the approach of previous study (Huang *et al.*, 2023). These are complemented by Tab. S3, which contains detailed
 metadata including all data entries, taxonomic classifications, quality control information, and the 16S rRNA gene amplicon
 sequences for each ASV. In Fig. S5a and S5b, we also highlight the genera that were most difficult to recover, which may
 guide future media development. Our media evaluation began at a broader level, first assessing MSPS_029 alone, then in
 combination with other media used under anaerobic conditions, and finally with media used under both anaerobic and aerobic
 conditions. The resulting increases in recovery were plotted for major genera (Fig. S5c), underscoring the importance of all
 three components: the core medium, and media used in both anaerobic and aerobic settings. To refine this further, we
 constructed seven sequential UpSet plots. In each step, the medium with the best complementary ASV recovery was identified,
 removed, and replaced with the next best option. This stepwise approach is summarized in Fig. S5d, which integrates the
 quantitative and qualitative contributions of each medium and captures the cumulative information from the UpSet plots. To
 validate our strategy, we compared our ASV accumulation curve with 10,000 permutations of randomly selected media
 combinations using the full ASV set (Fig. S5e). These findings also relate to the reviewer's point (2). Finally, we computed an
 extensive set of univariate ASV diversity metrics, covering most standard ecological indices and compare it with data generated
 at species and genus levels (Tab. S4). Rarefaction curves were generated at the ASV, species, and genus levels for two
 comparisons: inoculum vs. MSPS_029, and inoculum vs. all media combined. Additionally, we included a rarefaction curve
 for cultured isolates at the genus level to enable direct comparison between molecular and microbiological data (Fig. S6). Major
 updates are in pages 7-8, lines 157-163, and page 9, lines 193-203.

Regarding point (2): due to the limited sample size and our current focus on isolating complete culturomes rather
 than targeting individual not-yet-cultivated microorganisms, we are still constantly striving to establish dedicated isolation or
 detection modules for such taxa using reference strains and isolates from this study. As these organisms often constitute a minor
 fraction of the microbial community, we acknowledge that enrichment-based pre-processing (Murugkar *et al.*, 2020) and
 targeted hybridization techniques or antibodies will be essential for their effective detection and isolation (Cross *et al.*, 2019).
 The discussion section has been updated to reflect this perspective. Fastidious microorganisms were identified based on
 information from Bergey's (Bergey's Manual of Systematic Bacteriology, 2005-2012). Few examples include "*Abiotrophia* is
 nutritionally fastidious"; "Although *Actinomyces* species are rather fastidious; "Noticeable is the growth of fastidious Gram-
 negative pleomorphic" about *Aggregatibacter*; "Isolation of a fastidious *Bergeyella* species", etc. Additionally, we included
 organisms in this group if they have been reported as difficult to culture in clinical microbiology settings or if they require
 specialized growth conditions that are typically not met by standard culture media (DOWNES *et al.*, 2001). Examples include
 diverse anaerobes "which are slow growing, fastidious and generally unreactive in biochemical tests. As a consequence,
 cultivation and identification of isolates are difficult and the taxonomy of the group remains indifferent". Finally, oral species
 that are yet to be named and cultured were, by definition, classified as fastidious (Escapa *et al.*, 2018). SI was updated (page
 27, lines 864-866).

Regarding point (3): we have included a supplementary table (Tab. S9), now cited in the discussion, listing taxa that
 were not recovered through cultivation. We provide a potential explanation for the cultivation failures and suggest strategies

that could be implemented in future iterations of our culturomics approach including references. The discussion has also been
 updated accordingly (page 14, lines 328-329).

- Just incorporating all isolates, instead of a selection, into a culture collection would not be meaningful, practicable and
 efficient. Most of the strains are largely uncharacterized at the time of isolation and hence require substantial additional
 experimental work, yet they often will be similar to the well-characterized strains in public collections and hence not of further
 interest. Regarding the concept of a complete collection of all isolated cultures as promoted in the current paper, it also has to
 be noted that institutional collections are often discontinued due to ever increasing costs and lack of sustainable funding.
 Contrary to the generalizing statement (L. 23), there is not dearth of robust culture collections (there are 859 registered culture
 collection in the World Data Center for Microorganisms). The authors may have intended to say that few isolates from the
 human microbiome are currently available, but such a statement would not be appropriate in general. Notably, the list of the
 isolates from oral microbiome they were able to obtain (see LL. 59ff) is rather long.

**Author's Response:** We agree that incorporating all isolates into culture collection is not feasible. Adhering to the culturomics
 concept promoted in our work, we aim at collecting all phenotypically different strains for given specimen, thus the isolates
 representing the same species that cannot be distinguished from each other, *e.g.*, by co-aggregation, antimicrobial resistance,
 and full 16S rRNA gene amplicon sequence profiles, are represented only by a single representative strain in our collection.
 These strains are then available as individual stocks but also as model defined complex stocks which can be used to study
 microbial ecology (Cheng *et al.*, 2022). We recognize the challenge of sustaining well-characterized, representative, and up-
 to-date oral microbiome collections tailored to major oral microbial taxa and relevant oral conditions (De Paoli, 2005; Ryan
 *et al.*, 2021). However, such resources are indispensable for ecological studies that emphasize the inclusion of a broader range of
 strains, including subspecies, serotypes, and other closely related variants. Our mission is to make these resources available to
 the research community, initially at the local level, and progressively expanding our reach. We have successfully established a
 dedicated Biobank BIT located at the Lower Saxony Center for Biomedical Engineering, Implant Research and Development.
 We are highly motivated to maintain and expand this resource for both local and broader scientific use. For example our
 collection is an important asset for SIIRI consortium (www.siiri-sfb.de/en/) which develops smart dental implants. Locally, we
 benefit from strong connections to national and international experts in biobanking and culture collections, including those at
 the German Collection of Microorganisms and Cell Cultures (DSMZ), the Hannover Unified Biobank (HUB) at Hannover
 Medical School (MHH), and the University of Veterinary Medicine Hannover (TiHo), who can support and guide us in the
 development and long-term sustainability of this initiative. We acknowledge the importance of the major existing collections
 of oral microorganisms and actively collaborate within this field. However, a careful evaluation of the World Data Center for
 Microorganisms (Fan *et al.*, 2024) confirmed that isolates from oral clinical biofilms, particularly those associated with peri-
 implant conditions, remain scarce, and corresponding culturomes have not been available until now. Nevertheless, we
 appreciate the reviewer's perspective and have adopted more tentative language in the manuscript to reflect this. Corrections
 were made accordingly (page 3, lines 42-44; page 13, lines 291-294).

What would be needed, are additional analyses that provide deeper, quantitative insights into the above questions so that the
 implications of the data become clear. Aside from the laser-assisted bioprinting itself, the methodological descriptions in some
 parts are incomplete or sometimes even remain unclear. Many of the rather generic statements and exemplary descriptions (see
 below) would not be needed.

**Author's Response:** We have addressed the subsequent comments of the reviewer.

One important indicator for the superiority of a novel technique for cultivation is the recovery of numerous phylogenetically
 novel microorganisms, particularly of higher taxa like families, orders or particularly classes. The authors retrieved 249 isolates
 from 14 classes and 124 species. In par. LL. 144-150 they provide only general, and partly only indirect, evidence for the
 success their approach. For instance, the title of the chapter L. 135f claims that nearly half of the original species richness was
 recovered. For this, good estimators of the number of species present (which is not the species richness *per se*) are required.

**Author's Response:** Our approach focused primarily on developing improved strategies for culturomics, rather than
 introducing novel culture conditions or protocols tailored to specific higher taxa. Such taxon-specific strategies have been
 addressed by other exemplary studies (Bor *et al.*, 2020; Cross *et al.*, 2018; Cross *et al.*, 2019; Murugkar *et al.*, 2020; Vartoukian
 *et al.*, 2016a) and can be integrated into the culturomics framework over time. In our study, species richness is defined as the
 number of species present in a given sample. Additionally, we computed an extensive set of univariate diversity metrics at the
 ASV, species, and genus levels, covering most standard ecological indices (Tab. S4). Rarefaction curves were also generated
 for each of these taxonomic levels (Fig. S6). Recognizing that subspecies-level resolution is essential for understanding
 microbial ecology and translating findings to clinical contexts, we included ASV-level analyses in our study. Notably, by using
 a single medium (MSPS_029), we recovered 43% of ASVs from the inocula that had reached at least 0.1% relative abundance
 in one or more samples (Fig. S5 and S6, Tab. S3 and S4). Eighteen prominent genera, each represented by at least five ASVs
 across all inocula, were initially classified as hard-to-print, as fewer than 25% of their ASVs were recovered using MSPS_029
 alone. However, the broader culturomics strategy increased ASV recovery to 73%, with only *Haemophilus*, *Lautropia*, and
 genera representing *Candidatus Saccharibacteria* still meeting the hard-to-print criterion under the same threshold (Fig. S5
 and S6, Tab. S3 and S4). Importantly, both anaerobic and aerobic culture conditions contributed to the success of the
 culturomics approach (Fig. S5c). Specifically, MSPS_087, MSPS_023, and MSPS_010 were found to complement MSPS_029

particularly well, as demonstrated in Fig. S5d and S5e. Results section was updated accordingly (pages 7-8, lines 157-163;
page 9, lines 193-203).

The authors neither provide rarefaction curves (which at a sufficiently high number of replicas should be saturated and hence
provide a reliable estimate), Chao non-parametric diversity estimators, or similar measures. As a note on the side, the authors
may want to consider using state-of-the-art Hill coefficients rather than the Shannon or Simpson diversity and Evenness indices.

**Author's Response:** The rarefaction curves and additional diversity estimators were provided for ASV, species and genus
levels (Fig. S6, Tab. S4) as well as isolates at genus level (Fig. S6g, page 12, lines 278-281). We also presented the ASVs
accumulation plot for culturomics media (Fig. S5e).

Which of the existing microbial lineages are systematically missed by which medium and in combination? Hierarchical
clustering based on Bray-Curtis similarity or NMDS do not explicitly provide this information. Fig. 4b demonstrates that
typically not-yet-cultivated bacteria like the Saccharibacteria (should read 'Candidatus Saccharibacteria', Abscondibacteria
(use quotation mark, has no standing in nomenclature) or Gracilibacteria (use quotation mark, has no standing in
nomenclature) were also missed by the present approach. Fig. 5 is particularly relevant for the understanding of these results.
At first sight it would seem that most isolates depicted in Fig. 5e are unnamed phylotypes, yet most of them were neither in
bioprints nor purified (hence not cultivated at all). For some lineages (e.g. Oscillatoriaphycidae) the entire information is
missing – how did they get into this graph then? These results have to be reported much more clearly.

**Author's Response:** Seven species from 'Candidatus *Saccharibacteria*' and one species from 'Candidatus *Abscondibacteria*'
were detected in the inocula but not recovered in the bioprint arrays. Additionally, three known species representing
'Candidatus *Gracilibacteria*' were not detected in any of the patients included in the study. Potential reasons for the failure to
recover these taxa, along with possible solutions, are summarized in Tab. S9. Detailed information regarding the isolated and
missed species and ASVs is provided in Tab. S2 and Tab S3. The large number of oral species absent in bioprints and isolates
could be at least partly attributed to their low prevalence and low sample size of culturomics in our study. Manuscript was
corrected accordingly (pages 7-8, lines 157-163, and page 9, lines 193-203)

A key issue is also the taxonomic level at which comparative work is done. Based on the reasoning of the authors that full 16S
rRNA gene sequences provide a higher resolution of phylogenetic differences, I would have expected that all comparisons
between microorganisms cultured and originally present would have been done on the sequence variant level. Yet, the authors
remained exclusively on the species level in their analyses. They state that overall 65% of the species were recovered. It can
be debated, whether this is to be interpreted as "closely resembling the clinical situation" (L. 160), particularly since on the
sequence variant level (applied by other studies), this value is expected to be even lower.

**Author's Response:** The analysis at ASV level was provided (pages 7-8, lines 157-163, and page 9, lines 193-203).
Culturomics recovered 73% of ASVs (cut-off = 0.1%) present in the original inoculum (Fig. S5).

Furthermore, the description of unnamed oral taxa and OTUs (LL. 165 – 169) remains enigmatic. It is therefore unclear, to
which extent the "culturomics based on laser bioprinting seems...appropriate...for the exploration of oral microbial dark
matter" (LL. 168-169).

**Author's Response:** Descriptions of unnamed oral taxa were retrieved from the Human Oral Microbiome Database, eHOMD,
(Escapa *et al.*, 2018), which currently includes 774 classified oral bacterial species (www.homd.org/). Of these, 58% are
officially named, 16% remain unnamed but have been cultivated, and 26% are known solely as uncultivated phylotypes.
eHOMD provides a provisional naming scheme for unnamed taxa based on 16S rRNA gene phylogeny, allowing oral strain
sequence data to be linked to a stable taxonomic framework. For species-level assignment, we applied a conservative and
stringent approach that relied primarily on eHOMD, supplemented by a few additional 16S rRNA gene databases. Where
species-level classification failed, operational taxonomic units (OTUs) were generated (see Supplementary Materials and
Methods for details). We acknowledge that the primary focus of this study is the development of improved culturomics methods
rather than the in-depth exploration of microbial "dark matter." Accordingly, we have adopted more cautious language and
clearly delineated the scope of the study in the revised manuscript (page 3, line 42-44; page 8, lines 180-183, page 13, lines
291-294).

For similar reasons, the implications of the isolates obtained cannot be fully evaluated as the authors report on their in situ
transcriptional activity only on the genus level (L. 34).

**Author's Response:** We agree that a comparison at a higher taxonomic resolution would be preferable. However, due to
limitations in taxonomic classification, genus level was the lowest consistently applicable level for our comparative analysis.
We found that transcriptional activity served as a relevant indicator, suggesting an important and active role of these taxa in
peri-implantitis biofilms. While we recognize the significance of subgenus-level diversity (added ASV-level analyses), we
believe that reproducing complex biofilms at the genus level would still represent a substantial achievement, especially
considering that many previous human microbiome studies have operated at even higher taxonomic resolutions. The manuscript
has been updated accordingly (page 12, lines 278-281).

As another complication, the authors seem to have identified part of their isolates just by MALDI-TOF (L. 233) but the fraction
of these isolates and how they are represented in the figures remains unclear.

**Author's Response:** We provided the requested details in Tab. S6.

I also noted that at least *S. mutans* and *S. anginosus* cannot be readily distinguished by FT-IR (Fig. 5c, d).

**Author's Response:** LD3 clearly separates *S. mutans* (red) from *S. anginosus* (violet) Fig. 5d.

Fig. 2 could be central for the understanding of the Ms. The three legends on the top and bottom left could be stacked on top
of each other for easier reading. Sectoral lines should be more distinct. The panel on the bottom right is designed like a legend
but they correspond to the circle depicted for bacteria and archaea, and in analogy, should be shown as a sector of a circle. In
addition, the abbreviations for genera are non-standard.

**Author's Response:** Fig. 2 was updated accordingly; however, phages and fungi were not integrated into the bacterial and
archaeal dendrogram, as they lack the 16S rRNA gene used as the marker for phylogenetic tree construction.

In Fig. 3a and d, relative abundances can almost not be determined – a different way of presentation should be chosen or the
panels left out.

**Author's Response:** We provided detailed information on relative abundances at species and ASV levels in Tab. S2 and Tab.
S3, respectively.

While the first oral strain of *Colibacter massiliensis* was isolated by the authors (L. 253), a first strain of this species (Marseille-
P2911) existed before. Knowing possible differences in the characteristics of the two hence would be of particular interest.

**Author's Response:** To clarify the relationship between the two *Colibacter massiliensis* strains from distinct habitats we
sequenced the complete genome for our SPS_974 strain and confirmed its taxonomy (Meier-Kolthoff and Göker, 2019).
Genome annotation and the subsequent comparison of metabolic reconstruction (Aziz *et al.*, 2008; Overbeek *et al.*, 2013)
revealed 13 functional features that were unique to each strain, most notably associated with CRISPR-associated proteins and
restriction-modification systems. Among metabolism-related features, cystathionine gamma-lyase (EC 4.4.1.1) was specific to
the oral strain, whereas adenylosuccinate synthetase (EC 6.3.4.4) and glutathionylspermidine synthase (EC 6.3.1.8) were found
only in the gut strain, suggesting possible niche specialization. Further large-scale genomic studies are needed to confirm these
findings, and the prospect of *Colibacter* transmission between the oral cavity and gut remains an intriguing avenue for future
research (Schmidt *et al.*, 2019). We have updated both the manuscript and supporting information accordingly, and a new Fig.
S10 is provided below for your review. The results section has been updated accordingly (page 12, lines 284-285).

At the same time, the authors isolated a representative of *Arachnia rubra* which – contrary to the statement in their manuscript
- is not the first oral isolate (Sk-1 was isolated from gingival sulcus samples in 2018).

**Author's Response:** We corrected the manuscript accordingly (page 12, line 285-287).

I commend the authors for implementing an approach that enables the exploitation of interactions using laser imprinting. Yet,
mentioning just one example (*P. pasteri*, LL. 172-176) does not allow the reader to assess, whether “High recovery of species
was attributed to microbial syntrophy” (L. 171f). A quantitative assessment is needed here.

**Author's Response:** We used more tentative language (pages 8-9 lines 185-192). A closer investigation of this phenomenon
led to the follow-up study with draft title "*Metatranscriptomics-Guided Discovery of Naphthoquinone Interactions in Implant-
Associated Biofilms Between Taxonomically Diverse Suppliers and Bacteroidia Recipients*" which is currently in preparation.

In addition, the relevance of the FISH-approach (LL. 205ff) does not become clear in this respect. Were more colonies tested
in this respect (than the one shown) to detect co-cultures? That might yield very important information. The statement in L.
477 of the SI is not valid when formulated in such a generic way. Dependence on helpers critically depends on cultivation
conditions. Often the dependence is not given, if growth factors, specific growth substrates or dynamics of substrate (e.g.,
removal of hydrogen of syntrophic anaerobes) can be provided during incubations.

**Author's Response:** FISH was applied as a proof of concept using a limited set of type strains to demonstrate the potential of
the approach. It was not used as a primary method for the isolation. This clarification has been added to the manuscript (page
9, line 205).

We agree that the dependence on microbial "helpers" is highly influenced by cultivation conditions. While this
relationship was not explored in detail in the present manuscript, our strategy included the use of culture conditions known to
be permissive for such interactions. To support this, we inferred potential key helper species (Fig. 1c) and appropriate culture
conditions, e.g., diverse rich media containing blood like MSPS_029, MSPS_074, and MSPS_151, using a custom-built
database (Tab. S1). This database is part of the Database for Oral Microbial Interaction Networks (DOMINO), which integrates
data on species, enzymes, metabolites, microbial interactions, and supporting literature (Dieckow *et al.*, 2024). Built on the
Neo4j platform, DOMINO connects to external databases such as the Human Oral Microbiome Database (eHOMD), the Human
Metabolome Database (HMDB), the Kyoto Encyclopedia of Genes and Genomes (KEGG), and the U.S. National Library of

Medicine's PubMed database. Interaction data were manually curated from literature sources listed in the Supplementary
 Information. A custom graph was used to visualize genus-level networks. The Supplementary Information has been updated
 accordingly (SI, page 17, lines 552–574), and DOMINO entries related to syntrophic interactions are now detailed in new Tab.
 S1. Manuscript was updated (page 6, lines 120-121).

The current design of the figures and tables make it often hard or even impossible to find the evidence for the general
 conclusions and statements that the authors make in the text. This renders it even more difficult for the unspecialized reader to
 follow the reasoning of the paper. Large parts of the manuscript proper and some parts of the figures are far too generic to
 understand the approach and the conclusions. The authors present and visualize numerous exemplary data (e.g. Fig. S4c) or
 provide non-specific, basic information that should be eliminated (Fig. 1a, Fig. 4d, Fig. S5, Fig. S6a,d, Fig. S7a,f). In particular,
 LL. 312-321 are not at all instructive for understanding the methods. LL. 177-185 are all very basic and standard and not needed
 here. Instead, the experimental strategy should be given here. As another example, LL. 507-510 in the SI are basic textbook
 descriptions and hence superfluous. The implications of Fig. 1e remain unclear; what does it demonstrate, except that printing
 of a portrait is possible? What are the colors? The statement that microorganisms can be identified by colony morphology is a
 sweeping generalization (L.186) that needs to be qualified: only under very selective growth conditions, this might be true in
 some cases. And yet, there are many examples where novel types of bacteria did not follow the “presumptive identification”
 criteria, yielding false positives or negatives. Also, Fig. 4a does not provide any proof for this statement. The par. LL. 210-225
 reads as if novel properties of well established pathogenic species had been detected, yet at least several were well known
 before. The authors need to more clearly distinguish novel findings from existing knowledge.

**Author's Response:**

Generic parts of figures were provided to depict the working principles of the applied methods and were addressed to broad
 readership of the journal and prefer to retain them for this purpose. We addressed the reviewer's comments with 9 new
 supplementary tables (Tab S1 – S8 and S10) and 3 new supplementary figures (Fig. S5, S6, and S10). We also polished the
 other visualizations. We have revised the materials and methods to improve its completeness and overall structure for clarity.

Figure 1e was introduced to demonstrate that virtually any 2D structure can be bioprinted. Interestingly, the pattern
 also bears some resemblance to the patchy microbial structures found on the tongue (Wilbert *et al.*, 2020), which we aim to
 replicate more accurately in future work. Achieving this, however, will require significant improvements in our printing
 resolution. Additionally, we believe that incorporating artistic elements can help engage a broader audience and contribute to
 the wider popularization of our work, and of oral culturomics as a whole. Given the novelty of integrating art into our scientific
 work, we hope that our bioprinted image might be considered for a journal cover or featured in an accompanying commentary.

For bioprinting complex multi-species microbial "live art," we used UTI Chromogenic Agar (MSPS_109), a medium
 designed to detect and differentiate bacteria commonly associated with urinary tract infections. This medium contains a mixture
 of chromogenic substrates that enable visual differentiation of several microbial taxa based on their enzymatic activity.
 Specifically, X- β -glucoside is cleaved by β -D-glucosidase produced by members of the *Enterobacteriaceae* family, resulting
 in blue-colored colonies, typically from *Enterobacter* and *Klebsiella* species but also unrelated *Enterococcus* sp.. Another
 chromogenic substrate, cleaved by β -galactosidase, stains *E. coli*, *Enterobacter* spp., and *Klebsiella* spp. colonies pink.
 Additionally, tryptophan deaminase activity, characteristic of *Proteus* spp., *Morganella* spp., and *Providencia* spp., produces
 light brown colonies. Organisms that do not interact with these chromogenic substrates retain their natural colony color.

In the bioprinting experiment shown in Fig. 1e, five bacterial strains were used as "inks," each forming colonies with
 distinct coloration observable under a phase contrast microscope. *Escherichia coli* strain SPS_395 produced pink colonies due
 to β -D-galactosidase activity. *Enterobacter aerogenes* strain SPS_563 formed dark blue to violet colonies, reflecting the
 combined activity of β -D-galactosidase and β -D-glucosidase. *Proteus mirabilis* strain SPS_780 generated light brown colonies
 as a result of tryptophan deaminase activity. *Enterococcus faecalis* strain SPS_743 exhibited light to medium blue colonies,
 owing to β -D-glucosidase activity. In contrast, *Staphylococcus aureus* strain SPS_462 formed light beige colonies, as it lacked
 activity for any of the aforementioned enzymes. The Supplementary Information have been updated accordingly.

Throughout the text, SI is only referred to as unspecifically and the reader is expected to search through the 32 pages of the SI
 text in trying to find the description or evidence the authors refer to. The authors claim that “bacteriophage-bacterium
 relationships were reproduced” (L. 118) but I could not find evidence for this statement, also after consulting the SI.

**Author's Response:** The table of contents of the Supplementary Information has been revised to improve navigation and clarity
 across sections. For details regarding bacteriophages, please refer to Fig. 1d (bottom left panel), Fig. S3 (row 5, last column),
 as well as the SI section titled "Strains and Basic Culture Conditions", specifically the subsection related to the class
 *Caudoviricetes* (SI, pages 5-6, lines 150-166).

Several sections of the Discussion reiterate the results. It would be preferable to discuss the implications of the findings in more
 detail. For better comparison, I would have also expected that the following ref. was considered, at least in the discussion
 section:

Khelaifia S, Virginie P, Belkacemi S, Tassery H, Terrer E, Aboudharam G. Culturing the Human Oral Microbiota, Updating
 Methodologies and Cultivation Techniques. *Microorganisms*. 2023 Mar 24;11(4):836. doi: 10.3390/microorganisms11040836

- **Author's Response:** The discussion has been expanded, mostly pages 14-15 and lines 328-346, and the relevant reference has
been included as no. 50.
- - Specific comments
- **Author's Response:** We have addressed all specific comments within the main manuscript.
- 496 L. 29 16S rRNA gene profiling – Corrected at multiple positions (“rRNA” added).
- 497 L. 38 be more precise: do you mean diverse biofilm-associated species or something else? Otherwise the meaning of the
498 sentence is not understood. – Corrected to “species”.
- 499 L. 40 Microbial strains cannot be personalized, a treatment is – Sentence removed.
- 500 L. 42 There is no need for culture collections (see above) but there is a need for particular isolates that are so far
underrepresented or missing in existing culture collections. – Corrected as recommended; the need for culture collections has
been further justified, we believe that the ability to study ecological mechanisms using culturomes, *i.e.*, strain collections
isolated from individual clinical samples, provides significant value and complements insights obtained from model
communities constructed with reference strains.
- 505 L. 88 16S rRNA gene profiling – Corrected.
- 506 L. 329 it would help if the authors had information to which extent the droplets containing cells are heated during the laser
pulse. This heating has at least been an issue with laser tweezers. – due to the laser pulses being extremely short and focused,
cell-impairing heating is negligible. We investigated this before by analyzing heat-shock protein HSP70 expression of printed
cells (Gruene *et al.*, 2011). See page 16, lines 378-379.
- 510 L. 125 highly diverse cell envelope structure – Corrected.
- 511 L. 143 “input to output cell counts ratio” unclear and not described in SI – The input cell number per droplet was estimated
using phase contrast and fluorescence microscopy to range between $2 \cdot 10^1$ and $2 \cdot 10^2$. The estimated cell output per microbial
colony was based on data from a model organism (Mashimo *et al.*, 2004), typically around $2 \cdot 10^9$ cells per colony. However,
to account for the miniaturized size of our bioprints, approximately 0.5 mm in radius compared to 5 mm in standard *E. coli*
colonies, we scaled the output down by a factor of 10^3 , based on the volume difference between spheres of those radii. This
results in an estimated output of $2 \cdot 10^6 / 2 \cdot 10^2$ cells per bioprinted colony. Therefore, the estimated fold increase in cell
number is approximately 10^4 . SI has been updated accordingly (page 23, lines 725-730).
- 518 L. 147 45% of the original richness – Corrected.
- LL. 196ff Statement unclear – Corrected. Page 10, line 222.
- 520 L. 252f first oral strains....which are.. – Corrected.
- 521 L. 421 Reference is incomplete – Corrected.
- 522 L.466 In the outermost ring, the media... - Corrected.
- 523 L.482 Why does medium 29 appear twice? This is not clear here. –Printing on medium MSPS_029 was performed in duplicate
to assess reproducibility. For the analysis, the resulting profiles were combined, and the figure legend has been updated
accordingly.
- 526 L. 487 .. by class (for those encompassing... - Corrected
- Supplementary Material
- 528 L. 39 How were the research groups / strains selected? – Updated. Healthy volunteers were recruited to provide dental plaque
samples for method development. The peri-implantitis cohort forms part of an ongoing clinical cross-sectional study conducted
at the Department of Prosthetic Dentistry and Biomedical Materials Science, Hannover Medical School, Germany, within the
framework of the ‘SIIRI Biofilm Implant Cohort (BIC)’. SIIRI (Safety Integrated and Infection Reactive Implants) is an
interdisciplinary research initiative aimed at recruiting hundreds of participants and longitudinally monitoring their peri-implant
microbiomes over a period of up to 12 years. The goal is to better understand the onset and progression of peri-implant diseases
and to develop strategies for early detection and prevention. Reference strains were selected to represent key members of the
oral biofilm community. Additionally, the collection was supplemented with hard-to-isolate and allochthonous species known
to play significant roles in specific clinical population. SI was updated (page 3, lines 54-55; page 12, lines 362-367).
- 537 L. 57 To which source does the identification number of cultivation media refer to? – Sentence updated.

- 538 L. 59 and throughout The names of the higher taxa (e.g., Actinomycetia) should be given in italics. Also, type strains should
be designated with a superscript "T" following taxonomy conventions. – Corrected.
- 540 L. 118, L. 153 “Note taxonomy update (Hitch et al., 2022).” “Taxonomy updated following (Nouioui et al., 2018).” – what is
541 this supposed to mean? – Removed.
- 542 L. 154, L. 203 which helper strains were used? – Updated.
- 543 L. 285 “The concept of the stamp was adapted for prints” – and how was this achieved? – Updated. SI pages 10-11, lines 320-
323.
- 545 L. 288 Whatman – Corrected.
- 546 L. 292 depending – Corrected.
- 547 L. 316 retrieved – Corrected.
- 548 L. 436 Heparinase is a lyase (and not a DNase) with eliminative cleavage of polysaccharides containing (1→4)-linked D-
549 glucuronate or L-iduronate residues and (1→4)- α -linked 2-sulfoamino-2-deoxy-6-sulfo-D-glucose residues to give
oligosaccharides with terminal 4-deoxy- α -D-gluc-4-enuronosyl groups at their non-reducing ends. How then was this enzyme
detected? – Activity was detected using the principles of the previous study (Zimmermann et al., 1990). The method relies on
the differential precipitation of intact heparin and heparinase-generated heparin fragments using protamine sulfate. Heparinase
activity is visualized as clear zones against a white background, indicating enzymatic degradation of heparin. This assay was
shown to be suitable for screening recombinant heparinase expression and for identifying *Flavobacterium heparinum* mutants
that constitutively express heparinase.
- 556 L. 444, L. 449 either explain principle of test reaction or provide reference – Updated. (Smith and Willett, 1968) added twice.
- 557 L. 467 either explain principle of test reaction or provide reference – Updated. (Whaley et al., 1982) added.
- 558 L. 490, 491 use strains (plural, four times) – Corrected.
- 559 L. 497 A 0.45 μ m pore size may allow cells with smaller diameters to cross the membrane. This cannot be excluded by this
approach. – Updated and reference added (Vartoukian et al., 2016b)
- 561 L. 514 either all in capital letters or not – Corrected.
- 562 L. 533 I did not find any Table 10 – Updated.
- 563 L. 552 16S rRNA gene-targeting – Corrected.
- 564 L. 569 primers – Corrected.
- 565 L. 573 (reference) missing – Corrected. (Escapa et al., 2018) added.
- 566 L. 580 microbial community composition – Corrected.
- 567 L. 581 PacBio – Corrected.
- 568 L. 585 The following parameters – Corrected.
- 569 L. 585ff put all information past “were performed” in parenthesis – Corrected.
- 570 L. 603 identified to the species level – Corrected.
- 571 L. 605 delete dash in front of “representative” – Corrected.
- 572 L. 609 Were tests with negative controls (no DNA template) performed to check for sequences produced from reagents alone?
To our experience, this can make substantial contributions to the sequence dataset. – Although negative controls were not
included, we did not process any low-input samples and used a low number of PCR cycles to minimize the risk of
contamination. We did not observe typical contaminants reported for DNA isolation reagents.
- 576 L. 621 nucleatum – Corrected.
- 577 L. 698 Cellular morphology, motility and – Corrected.
- 578 L. 701 hemin – Corrected.
- 579 L. 718 in the culture collection – Corrected.
- 580 L. 1069f give species names in italics – Corrected.

**Author's Response:** We have addressed all specific comments within the supplementary material.

**Reviewer #5 (Remarks to the Author):**

The manuscript by Qu *et al* describes the development of a laser-assisted microbial culturomics system and the deployment of
it for isolation of oral microbes. The authors describe the versatility of the system to print in different configurations including
co-cultures, in plates, on glass slides, and across membranes. Then the system was used to isolate clinical biofilms from dental
plaques, which showed the ability to recover most of the abundant species including some previously uncultured isolates. The
authors further deployed the system in combination with microscopy, spectroscopy and enzyme assays to improve isolation
performance including the development of algorithms that improved colony discrimination.

Overall, the technology described is interesting and will have general interest for the microbial cultivation field. The isolates
generated are of interest to those studying oral microbiome. While there is a lot of possible data in this paper, the description
of how they are generated and their organization is very poor. The paper is quite difficult to parse and much of the technical
details are either not available or hard to find. This paper should be re-written and reorganized to improve overall clarity of
what is done and how it was done.

**Author's Response:** We have extensively revised the manuscript and supplementary information, particularly the methodology
sections, to enhance clarity and include the previously missing details by the addition of 9 new supplementary tables.

Major concerns:

1) Details for different parts of the study are not well described. For example, Figure 1C shows syntrophic species interactions
that can support fastidious microbes, but it's not clear how that figure was generated and the underlying data for it. The
supplemental information only lists references, which is not very informative otherwise. Another example is Figure 6, how is
this figure generated? What RNAseq was performed?

**Author's Response:** The potential for metabolic interactions between oral species, where one species supplies essential factors
for another's growth, was inferred using a custom database (Fig. 1c). This database is part of the Database for Oral Microbial
Interaction Networks (DOMINO), which integrates information on species, enzymes, metabolites, interactions, and references
(Dieckow *et al.*, 2024). Built on the Neo4j platform, DOMINO connects to external databases such as the Human Oral
Microbiome Database (eHOMD), the Human Metabolome Database (HMDB), the Kyoto Encyclopedia of Genes and Genomes
(KEGG), and the U.S. National Library of Medicine's PubMed database. Interaction data were manually curated from literature.
A custom graph was used to visualize genus-level networks. The manuscript and SI have been updated accordingly (page 6,
lines 120-121; and SI, page 18, lines 552-574), and DOMINO entries related to syntrophic interactions are now detailed in
new Table S1. This analysis identified the species whose strains are potentially the most robust metabolic "helpers".

Figure 6 illustrates how well our strain collection represents microbial genera active in peri-implantitis biofilms from
our previous study (Grischke *et al.*, 2021). Given taxonomic limits of cultivation and sequencing, we compared isolates and
gene expression at the genus level. We prioritized *in vivo* transcriptional activity as a proxy for ecological and clinical relevance.
Of the top 20 most transcriptionally active genera, all but two were represented in our collection; the missing ones were covered
by printed reference strains, suggesting we may recover them with more samples. Metatranscriptomes were generated from
submucosal plaque and crevicular fluid collected using paper points and curettes, preserved in RNAprotect, and stored at
617 -80 °C. RNA was extracted, purified, depleted of rRNA, reverse-transcribed, and sequenced (50-cycle Illumina). Reads were
618 mapped to eHOMD reference genomes (Escapa *et al.*, 2018). Reads were aggregated at the genus level, ranked by mean relative
activity in biofilms, and the 50 most active genera were visualized using box plots. SI was updated accordingly (page 24, lines
748-756). For clarity, the data shown in Fig. 6 are also detailed in Tab. S7.

2) It is not clear how the isolation is performed following the printing. Are colonies picked by hand? If so, how many total
colonies were picked and what level of redundancy was found based on the most optimal colony differentiation strategy
(enzyme, spectra, nutrients)?

**Author's Response:** Yes, we performed manual colony transfer, selecting approximately 2,000 colonies in total. Overall,
colony morphology and fluorescence were effective differentiation strategies, similar to other culturomics approaches (Huang
*et al.*, 2023). However, strains exhibiting colony polymorphism, *e.g.*, certain members of the genera *Actinomyces*, *Lactobacillus*
*sensu lato*, *Prevotella*, *Shuttleworthia*, *Streptococcus*, and *Veillonella* or forming small, circular colonies without distinct
characteristics, *e.g.*, multiple very fastidious slow growing anaerobe species, led to approximately 5-fold or in extreme cases
10-fold higher redundancy of 10-40 strains but we acknowledge the importance of systematic quantification in the future.
Addressing this redundancies will require cost-effective targeted solutions, *e.g.*, Fourier-Transform Infrared Spectroscopy (FT-
IR), for rapid differentiation of polymorphic colonies. We also observed improved differentiation when antibiotic selection or
antibiotic testing was incorporated, either by controlled reduction of isolate diversity or classification methods. However, this
was not the case for *Lactobacillus sensu lato*, which appeared on multiple antibiotic media in varied colony forms, resulting in

redundancies across media. Alternative identification methods beyond colony morphology were introduced as proof-of-
principle examples; however, their practical application in high-throughput settings remains challenging and requires further
optimization. The results and discussion sections have been updated accordingly. Manuscript was updated accordingly (page
9, line 205, Tab. S5). Redundancy of our isolation approach could be estimated at genus level using rarefaction curve. Data
suggests that recovering new genera would require substantial effort, additional patient samples, or expanded culture conditions
(Fig. S6g, page 12, lines 278-281).

3) Genomic and taxonomic data associated with the isolation is missing. The specific strains isolated across the study is not
listed anywhere. A supplemental table of all isolates generated, and associated genomic info (e.g. 16S) is needed at the
minimum. The SI contains just a text dump of isolates which is not useful for most people. Are these strains available to the
larger research community?

**Author's Response:** We have added a supplemental table listing all representative isolates with their full or partial 16S rRNA
gene sequences and/or MALDI-based classification (Tab. S6). The isolates are currently available through a dedicated Biobank
BIT located in the Lower Saxony Center for Biomedical Engineering, Implant Research and Development and can be shared
upon reasonable request. We are also refining the Biobank BIT to develop well-characterized, representative, and up-to-date
oral microbiome collections tailored to major microbial taxa and relevant conditions.

4) Figure 2 is really difficult to interpret. There are so many different things being shown. The isolation media are listed as
numbers, but I am not able to find their correspondence anywhere. A table would have better summarized these results than
the circular plot. Also, much of the text is bunched up and it is just overall impossible to understand all the different elements.

**Author's Response:** We revised the figure to enhance clarity. Detailed information on the media is available in the SI (pages
12–16). Each medium can be identified by its MSPS number, which is provided in both Figure 2 and the SI.

5) It is not clear from the laser printing methodology how one can obtain clonal isolates directly from the biofilm. A single
droplet may contain dozens of different microbes. The manuscript is not very clear on how individual isolates are generated
from the spots. This is true for Figure 3 for example.

**Author's Response:** We manually transferred colonies under a phase-contrast microscope whenever possible, selecting based
on morphology using macro photography, phase-contrast microscopy, or occasionally confocal microscopy for IR-fluorescent
anaerobes. Transparent media enabled microscopy, while opaque media required imaging. Screening began with camera-based
selection, followed by microscopy. Colonies were grouped into morphotypes based on images processed using Adobe
Illustrator (CS6), and representative ones were manually passaged multiple times onto agar plates. Due to the limitations of
manual handling and fused colonies, some transfers included multiple strains, e.g. fluorescent colonies of *Prevotella* embedded
in or fused to colonies of other species. Resulting colonies were photographed, evaluated, purified, phenotyped and
taxonomically identified. The respective manuscript and SI sections were updated accordingly (page 10, lines 224-225 and SI,
page 20, lines 638–640). We have also revised the legend of Fig. 4 for improved clarity (page 26, lines 581-582). Fig. 3 presents
full 16S rRNA gene amplicon profiles for the complete arrays, reflecting the relative abundance of taxa within them.

6) The paper outlines different analysis steps that were done to study the biofilms including optical and biochemical. It is not
clear how different modalities truly contributed to isolating new strains. The manuscript simply shows some examples of plates,
but a quantitative analysis is needed.

**Author's Response:** Primary isolations were guided by microscopy, as in other culturomics studies. The additional techniques
were included to illustrate the broader approach and require further development to play a more central role. Enzyme assays,
for example, are currently basic and should be replaced with more robust hybridization-based methods. FT-IR shows strong
potential as a cost-effective tool but depends on building an extensive spectral database. Manual colony transfer also needs to
be replaced with at least a semi-automated approach. Strains isolated with a proof-of-principle methods are listed in Tab S5.
Manuscript was updated accordingly (page 9, line 205).

7) The manuscript stated that they used LDA to distinguish different microbes versus NN or RF, but it seems that this is tested
only on 5 type strains. Was this visualization modality really used for the isolation of the collection or just a proof of concept
on a few different strains? This is also true for Fig 6(a-d)

**Author's Response:** We agree with the reviewer. LDA was applied as a proof-of-concept using a limited set of type strains to
demonstrate the potential of the approach. It was not used as a primary method for the isolation but as an element of quality
control. This clarification had been already added to the manuscript. We are currently developing tailored isolation protocols
for specific taxa that combine FT-IR spectroscopy with selective media and AI-enhanced image analysis of macro photography,
with the aim of replacing or, in some cases, outperforming sequencing-based methods such as partial 16S rRNA gene amplicons
sequencing. Additional correction was made in page 5 line 90.

8) Overall, the supplemental figures are fuzzy and low resolution so it is hard to read all the data on there.

**Author's Response:** We have improved the overall quality of figures in SI.

**References**

- Aziz RK, Bartels D, Best AA, DeJongh M, Disz T, Edwards RA *et al.* (2008). The RAST Server: Rapid Annotations using Subsystems Technology.
*BMC Genomics* 9(1):75.
- Bergey's Manual of Systematic Bacteriology (2005-2012). 2nd ed.
- Bor B, Collins AJ, Murugkar PP, Balasubramanian S, To TT, Hendrickson EL *et al.* (2020). Insights Obtained by Culturing Saccharibacteria With
Their Bacterial Hosts. *Journal of dental research* 99(6):685-694.
- Cheng AG, Ho P-Y, Aranda-Díaz A, Jain S, Yu FB, Meng X *et al.* (2022). Design, construction, and in vivo augmentation of a complex gut
microbiome. *Cell* 185(19):3617-3636.e3619.
- Cross KL, Chirania P, Xiong W, Beall CJ, Elkins JG, Giannone RJ *et al.* (2018). Insights into the Evolution of Host Association through the Isolation
and Characterization of a Novel Human Periodontal Pathobiont, *Desulfobulbus oralis*. *mBio* 9(2).
- Cross KL, Campbell JH, Balachandran M, Campbell AG, Cooper CJ, Griffen A *et al.* (2019). Targeted isolation and cultivation of uncultivated
bacteria by reverse genomics. *Nature biotechnology* 37(11):1314-1321.
- Dar D, Dar N, Cai L, Newman DK (2021). Spatial transcriptomics of planktonic and sessile bacterial populations at single-cell resolution. *Science*
373(6556):eabi4882.
- De Paoli P (2005). Biobanking in microbiology: From sample collection to epidemiology, diagnosis and research. *FEMS Microbiology Reviews*
29(5):897-910.
- Dieckow S, Szafranski SP, Grischke J, Qu T, Doll-Nikutta K, Steglich M *et al.* (2024). Structure and composition of early biofilms formed on
dental implants are complex, diverse, subject-specific and dynamic. *npj Biofilms and Microbiomes* 10(1):155.
- DOWNES J, MUNSON MA, SPRATT DA, KONONEN E, TARKKA E, JOUSIMIES-SOMER H *et al.* (2001). Characterisation of Eubacterium-like strains
isolated from oral infections. *Journal of Medical Microbiology* 50(11):947-951.
- Escapa IF, Chen T, Huang Y, Gajare P, Dewhirst FE, Lemon KP (2018). New Insights into Human Nostril Microbiome from the Expanded Human
Oral Microbiome Database (eHOMD): a Resource for the Microbiome of the Human Aerodigestive Tract. *mSystems*
3(6):10.1128/msystems.00187-00118.
- Fan G, Sun Q, Sun Y, Liu D, Li S, Li M *et al.* (2024). GCM and gcType in 2024: comprehensive resources for microbial strains and genomic data.
*Nucleic Acids Research* 53(D1):D763-D771.
- Gloag ES, German GK, Stoodley P, Wozniak DJ (2018). Viscoelastic properties of *Pseudomonas aeruginosa* variant biofilms. *Scientific reports*
8(1):9691.
- Grischke J, Szafranski SP, Muthukumarasamy U, Haeussler S, Stiesch M (2021). Removable denture is a risk indicator for peri-implantitis and
facilitates expansion of specific periodontopathogens: a cross-sectional study. *BMC Oral Health* 21(1):173.
- Gruene M, Deiwick A, Koch L, Schlie S, Unger C, Hofmann N *et al.* (2011). Laser Printing of Stem Cells for Biofabrication of Scaffold-Free
Autologous Grafts. *Tissue Engineering Part C: Methods* 17(1):79-87.
- Huang Y, Sheth RU, Zhao S, Cohen LA, Dabaghi K, Moody T *et al.* (2023). High-throughput microbial culturomics using automation and machine
learning. *Nature biotechnology*.
- Mashimo K, Nagata Y, Kawata M, Iwasaki H, Yamamoto K (2004). Role of the RuvAB protein in avoiding spontaneous formation of deletion
mutations in the *Escherichia coli* K-12 endogenous *tonB* gene. *Biochemical and biophysical research communications* 323(1):197-203.
- Meier-Kolthoff JP, Göker M (2019). TYGS is an automated high-throughput platform for state-of-the-art genome-based taxonomy. *Nature*
*Communications* 10(1):2182.

Murugkar PP, Collins AJ, Chen T, Dewhirst FE (2020). Isolation and cultivation of candidate phyla radiation Saccharibacteria (TM7) bacteria in
coculture with bacterial hosts. *Journal of oral microbiology* 12(1):1814666.

Overbeek R, Olson R, Pusch GD, Olsen GJ, Davis JJ, Disz T *et al.* (2013). The SEED and the Rapid Annotation of microbial genomes using
Subsystems Technology (RAST). *Nucleic Acids Research* 42(D1):D206-D214.

Ryan MJ, Schloter M, Berg G, Kostic T, Kinkel LL, Eversole K *et al.* (2021). Development of Microbiome Biobanks – Challenges and
Opportunities. *Trends in Microbiology* 29(2):89-92.

Sarfatis A, Wang Y, Twumasi-Ankrah N, Moffitt JR (2025). Highly multiplexed spatial transcriptomics in bacteria. *Science* 387(6732):eadr0932.

Schmidt TSB, Hayward MR, Coelho LP, Li SS, Costea PI, Voigt AY *et al.* (2019). Extensive transmission of microbes along the gastrointestinal
tract. *eLife* 8(e42693).

Smith RF, Willett NP (1968). Rapid plate method for screening hyaluronidase and chondroitin sulfatase-producing microorganisms. *Applied*
*microbiology* 16(9):1434-1436.

Vartoukian SR, Adamowska A, Lawlor M, Moazzez R, Dewhirst FE, Wade WG (2016a). In Vitro Cultivation of 'Unculturable' Oral Bacteria,
Facilitated by Community Culture and Media Supplementation with Siderophores. *PLoS one* 11(1):e0146926.

Vartoukian SR, Moazzez RV, Paster BJ, Dewhirst FE, Wade WG (2016b). First Cultivation of Health-Associated *Tannerella* sp. HOT-286 (BU063).
*Journal of dental research* 95(11):1308-1313.

Wang S, Zhao Y, Breslawec AP, Liang T, Deng Z, Kuperman LL *et al.* (2023). Strategy to combat biofilms: a focus on biofilm dispersal enzymes.
*npj Biofilms and Microbiomes* 9(1):63.

Whaley DN, Dowell VR, Wanderlinder LM, Lombard GL (1982). Gelatin agar medium for detecting gelatinase production by anaerobic
bacteria. *Journal of clinical microbiology* 16(2):224-229.

Wilbert SA, Mark Welch JL, Borisy GG (2020). Spatial Ecology of the Human Tongue Dorsum Microbiome. *Cell Reports* 30(12):4003-
4015.e4003.

Zimmermann JJ, Langer R, Cooney CL (1990). Specific plate assay for bacterial heparinase. *Applied and environmental microbiology*
56(11):3593-3594.

20.05.2025, Hannover

Dear Reviewers,

Thank you very much for evaluating our revised manuscript *Laser-assisted microbial culturomics* (NCOMMS-24-31352A)
and your valuable feedback. We have thoroughly addressed your remaining comments and corrections, and the revised
manuscript includes updated figures to enhance clarity. Please find our detailed point-by-point responses below. We are looking
forward to hearing from you,

Sincerely,

Szymon Szafranski
on behalf of all co-authors

REVIEWER COMMENTS

**Reviewer #1 (Remarks to the Author):**

The authors have made substantial revisions to the manuscript and have adequately addressed previous concerns.

**Reviewer #2 (Remarks to the Author):**

The authors have satisfactorily addressed my comments and concerns; I would now suggest that this manuscript is suitable for
publication in Nature Communications.

**Reviewer #3 (Remarks to the Author):**

I co-reviewed this manuscript with one of the reviewers who provided the listed reports. This is part of the Nature
Communications initiative to facilitate training in peer review and to provide appropriate recognition for Early Career
Researchers who co-review manuscripts.

**Reviewer #4 (Remarks to the Author):**

Most missing information has been added to the manuscript and open questions have been clarified. However, I still find
figures 1 to 5 very complex.

**Author's Response:** The figures were revised to enhance clarity and completeness. In Figure 1c updates include the updated
taxon names of "*Candidatus Absconditabacteria*" and "*Candidatus Saccharimonadia*", as well as enhanced font size for better
readability. In Figure 2 updates include the updated taxon names of "*Candidatus Nanosynbacter*". The typo 'contast' in Figure
4 has been corrected; we regret that this error was overlooked during the previous revision.

Minor comments:

28 L. 121: These species were consequently have been employed... ; delete "have been"

**Author's Response:** "have been" was deleted (page 6, line 121)

LL. 354-364: extremely long, unstructured sentence that is hard to read and needs to be split into several, separate statements.

**Author's Response:** The fragment was revised to enhance clarity and completeness (pages 15 – 16, lines 355 – 370).

32 L. 170 and also L. 197, L. 328: there is no class 'Saccharibacteria' described so far. Only the phylum 'Candidatus
Saccharimonadota' (with *Candidatus* in italics and Saccharimonadota not) is currently recognized and hence should be used
here instead of the wrong 'Saccharibacteria'.

**Author's Response:** We used "*Candidatus Saccharimonadia*" as the class-level designation for oral members of the former
Candidate division TM7 throughout the manuscript (page 8, line 170; page 9, line 198; page 14, line 329). See
<https://lpsn.dsmz.de/class/saccharimonadia> as well as effective publication: Lemos *et al.*, Mol Ecol 2019; 28:4259-4271.

Table S9: Leptotrichia needs to be in italics

**Author's Response:** Corrected.

**Author's Comment:** In the Supplementary Information, the amplicon sequence variant (ASV) analysis was mistakenly placed
alongside the Sanger 16S sequencing section. In the revised version, it has been correctly relocated to the section on PacBio
16S sequencing (pages 22 – 23, lines 717 – 740).

**Reviewer #5 (Remarks to the Author):**

The revised manuscript is much improved. The reviewer has no further issues.